# On Measuring Fairness in Generative Models

**Christopher T. H. Teo**
christopher_teo@mymail.sutd.edu.sg

**Milad Abdollahzadeh**
milad_abdollahzadeh@sutd.sg

**Ngai-Man Cheung**[*]
ngaiman_cheung@sutd.edu.sg

Singapore University of Technology and Design (SUTD)

## Abstract

Recently, there has been increased interest in fair generative models. In this work, we conduct, for the first time, an in-depth study on **fairness measurement**, a critical component in gauging progress on fair generative models. We make three contributions. First, we conduct a study that reveals that the existing fairness measurement framework has considerable measurement errors, even when highly accurate sensitive attribute (SA) classifiers are used. These findings cast doubts on previously reported fairness improvements. Second, to address this issue, we propose CLassifier Error-Aware Measurement (CLEAM), a new framework which uses a statistical model to account for inaccuracies in SA classifiers. Our proposed CLEAM reduces measurement errors significantly, e.g., **4.98%→0.62%** for StyleGAN2 *w.r.t.* `Gender`. Additionally, CLEAM achieves this with minimal additional overhead. Third, we utilize CLEAM to measure fairness in important text-to-image generator and GANs, revealing considerable biases in these models that raise concerns about their applications. **Code and more resources:** `https://sutd-visual-computing-group.github.io/CLEAM/`.

## 1 Introduction

Fair generative models have been attracting significant attention recently [1, 2, 7–13]. In generative models [14–18], fairness is commonly defined as equal generative quality [11] or equal representation [1, 2, 7, 9, 12, 19, 20] *w.r.t.* some *Sensitive Attributes* (SA). In this work, we focus on the more widely utilized definition – *equal representation*. In this definition, as an example, a generative model is regarded as fair *w.r.t.* `Gender`, if it generates `Male` and `Female` samples with equal probability. This is an important research topic as such biases in generative models could impact their application efficacy, e.g., by introducing racial bias in face generation of suspects [21] or reducing accuracy when supplementing data for disease diagnosis [22].

**Fairness measurement for generative models.** Recognizing the importance of fair generative models, several methods have been proposed to mitigate biases in generative models [1, 2, 7, 9, 12]. However, *in our work, we focus mainly on the accurate fairness measurement of deep generative models i.e. assessing and quantifying the bias of generative models.* This is a critical topic, as accurate measurements are essential to reliably gauge the progress of bias mitigation techniques. The general fairness measurement framework is shown in Fig. 1 (See Sec. 2 for details). This framework is utilized in existing works to assess their proposed fair generators. Central to the fairness measurement framework is a *SA classifier*, which classifies the generated samples *w.r.t.* a SA, in order to estimate the bias of the generator. For example, if eight out of ten generated face images are classified as `Male`

---

[*]Corresponding Author

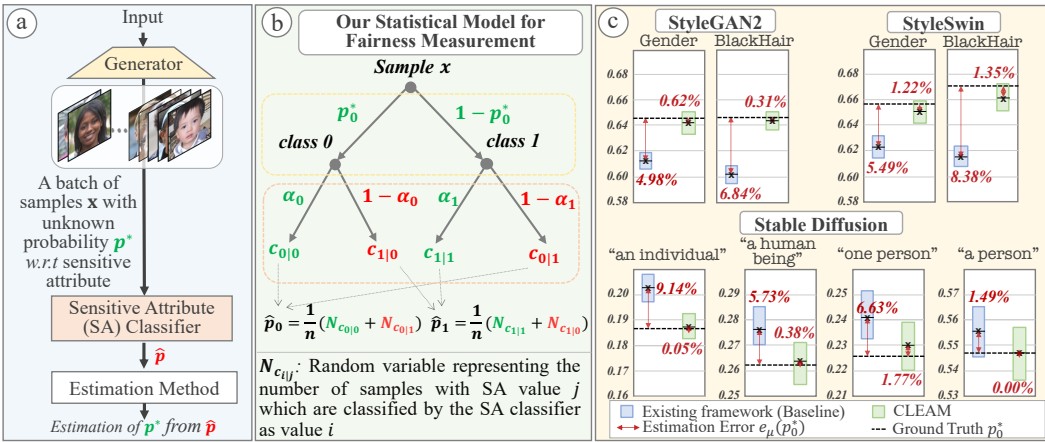

Figure 1: ⓐ **General framework for measuring fairness in generative models.** Generated samples with unknown ground-truth (GT) probability $p^*$ *w.r.t.* sensitive attribute (SA) are fed into a SA classifier to obtain $\hat{p}$. Existing framework (Baseline) uses the classifier output $\hat{p}$ as estimation of $p^*$. In contrast, our proposed CLEAM includes an improved estimation that accounts for inaccuracies in the SA classifier (see Alg. 1). ⓑ **Our statistical model for fairness measurement.** This model accounts for inaccuracies in the SA classifier and is the base of our proposed CLEAM (see Sec. 4.1). ⓒ **Improvements with CLEAM.** CLEAM improves upon Baseline [1, 2] by reducing the relative error in estimating the GT $p_0^*$ for SOTA GANs: StyleGAN2 [3] and StyleSwin [4], and Stable Diffusion Model [5]. First row displays the Baseline and CLEAM estimates for each GAN, using ResNet-18 as the SA classifier for `Gender` and `BlackHair`. The Baseline incurs significant fairness measurement errors (*e.g.* 4.98%), even when utilizing a highly accurate ResNet-18 ($\approx$97% accuracy). Meanwhile, CLEAM reduces the error significantly in all setups, *e.g.* in the first panel, the error is reduced: 4.98% $\rightarrow$ 0.62%. Similarly, in the second row, CLEAM reduces measurement error significantly in the Stable Diffusion Model [5], using CLIP [6] as the SA classifier for `Gender`, *e.g.* first panel: 9.14% $\rightarrow$ 0.05% (Detailed evaluation in Tab. 1 and Tab. 2). **Best viewed in color.**

by the SA classifier, then the generator is deemed biased at $0.8$ towards `Male` (further discussion in Sec. 2). We follow previous works [1, 2, 12] and focus on binary SA due to dataset limitations.

**Research gap.** In this paper, we study a critical research gap in fairness measurement. Existing works assume that when SA classifiers are highly accurate, measurement errors should be insignificant. As a result, the effect of errors in SA classifiers has not been studied. However, our study reveals that *even with highly accurate SA classifiers, considerable fairness measurement errors could still occur*. This finding raises concerns about potential errors in previous works' results, which are measured using existing framework. Note that the SA classifier is *indispensable* in fairness measurement as it enables automated measurement of generated samples.

**Our contributions.** We make three contributions to fairness measurement for generative models. *As our first contribution*, we analyze the accuracy of fairness measurement on generated samples, which previous works [1, 2, 7, 9, 12] have been unable to carry out due to the unavailability of proper datasets. We overcome this challenge by proposing new datasets of *generated samples* with manual labeling *w.r.t.* various SAs. The datasets include generated samples from Stable Diffusion Model (SDM) [5] —a popular text-to-image generator— as well as two State-of-The-Art (SOTA) GANs (StyleGAN2 [3] and StyleSwin [4]) *w.r.t.* different SAs. Our new datasets are then utilized in our work to evaluate the accuracy of the existing fairness measurement framework. Our results reveal that the accuracy of the existing fairness measurement framework is not adequate, due to the lack of consideration for the SA classifier inaccuracies. More importantly, we found that *even in setups where the accuracy of the SA classifier is high, the error in fairness measurement could still be significant*. Our finding raises concerns about the accuracy of previous works' results [1, 2, 12], especially since some of their reported improvements are smaller than the margin of measurement errors that we observe in our study when evaluated under the same setup; further discussion in Sec. 3.

To address this issue, *as our second (major) contribution*, we propose CLassifier Error-Aware Measurement (CLEAM), a new more accurate fairness measurement framework based on our developed statistical model for SA classification (further details on the statistical model in Sec. 4.1).

Specifically, CLEAM utilizes this statistical model to account for the classifier's inaccuracies during SA classification and outputs a more accurate fairness measurement. We then evaluate the accuracy of CLEAM and validate its improvement over existing fairness measurement framework. We further conduct a series of different ablation studies to validate performance of CLEAM. We remark that CLEAM is not a new fairness metric, but an improved fairness measurement framework that could achieve better accuracy in bias estimation when used with various fairness metrics for generative models.

*As our third contribution*, we apply CLEAM as an accurate framework to reliably measure biases in popular generative models. Our study reveals that SOTA GANs have considerable biases *w.r.t.* several SA. Furthermore, we observe an intriguing property in Stable Diffusion Model: slight differences in semantically similar prompts could result in markedly different biases for SDM. These results prompt careful consideration on the implication of biases in generative models. **Our contributions are:**

- We conduct a study to reveal that even highly-accurate SA classifiers could still incur significant fairness measurement errors when using existing framework.

- To enable evaluation of fairness measurement frameworks, we propose new datasets based on generated samples from StyleGAN, StyleSwin and SDM, with manual labeling *w.r.t.* SA.

- We propose a statistically driven fairness measurement framework, CLEAM, which accounts for the SA classifier inaccuracies to output a more accurate bias estimate.

- Using CLEAM, we reveal considerable biases in several important generative models, prompting careful consideration when applying them to different applications.

## 2   Fairness Measurement Framework

Fig.1(a) illustrates the fairness measurement framework for generative models as in [1, 2, 7, 9, 12]. Assume that with some input *e.g.* noise vector for a GAN or text prompt for SDM, a generative model synthesizes a sample $\mathbf{x}$. Generally, as the generator does not label synthesized samples, the ground truth (GT) class probability of these samples *w.r.t.* a SA (denoted by $\boldsymbol{p}^*$) is unknown. Thus, an SA classifier $C_\mathbf{u}$ is utilized to estimate $\boldsymbol{p}^*$. Specifically, for each sample $\mathbf{x} \in \{\mathbf{x}\}$, $C_\mathbf{u}(\mathbf{x})$ is the argmax classification for the respective SA. In existing works, the expected value of the SA classifier output over a batch of samples, $\hat{\boldsymbol{p}} = \mathbb{E}_\mathbf{x}[C_\mathbf{u}(\mathbf{x})]$ (or the average of $\hat{\boldsymbol{p}}$ over multiple batches of samples), is used as an estimation of $\boldsymbol{p}^*$. This estimate may then be used in some fairness metric $f$ to report the fairness value for the generator, *e.g.* fairness discrepancy metric between $\hat{\boldsymbol{p}}$ and a uniform distribution $\bar{\boldsymbol{p}}$ [1, 20](see Supp A.3 for details on how to calculate $f$). Note that *the general assumption behind the existing framework is that with a reasonably accurate SA classifier, $\hat{\boldsymbol{p}}$ could be an accurate estimation of $\boldsymbol{p}^*$* [1, 9]. In the next section, we will present a deeper analysis on the effects of an inaccurate SA classifier on fairness measurement. Our findings suggest that there could be a large discrepancy between $\hat{\boldsymbol{p}}$ and $\boldsymbol{p}^*$, even for highly accurate SA classifiers, indicative of significant fairness measurement errors in the current measurement framework.

One may argue that conditional GANs (cGANs) [23, 24] may be used to generate samples conditioned on the SA, thereby eliminating the need for an SA classifier. However, cGANs are not considered in previous works due to several limitations. These include the limited availability of large *labeled* training datasets, the unreliability of sample quality and labels [25], and the exponentially increasing conditional terms, per SA. Similarly, for SDM, Bianchi *et al.* [26] found that utilizing well-crafted prompts to mitigate biases is ineffective due to the presence of existing biases in its training dataset. Furthermore in Sec. 6, utilizing CLEAM, we will discuss that even subtle prompt changes (while maintaining the semantics) result in drastically different SA biases. See Supp G for further comparison between [26] and our findings.

## 3   A Closer Look at Fairness Measurement

In this section, we take a closer look at the existing fairness measurement framework. In particular, we examine its performance in estimating $\boldsymbol{p}^*$ of the samples generated by SOTA GANs and SDM, a task previously unstudied due to the lack of a labeled generated dataset. We do so by designing an experiment to demonstrate these errors while evaluating biases in popular image generators. Following previous works, our main focus is on binary SA which takes values in $\{0, 1\}$. Note that,

we assume that the accuracy of the SA classifier $C_{\mathbf{u}}$ is known and is characterized by $\boldsymbol{\alpha} = \{\alpha_0, \alpha_1\}$, where $\alpha_i$ is the probability of correctly classifying label $i$. For example, for Gender attribute, $\alpha_0$ and $\alpha_1$ are the probability of correctly classifying Female, and Male classes, respectively. In practice, $C_{\mathbf{u}}$ is trained on standard training procedures (more details in the Supp F) and $\boldsymbol{\alpha}$ can be measured during the validation stage of $C_{\mathbf{u}}$ and be considered a constant when the validation dataset is large enough. Additionally, $\boldsymbol{p}^*$ can be assumed to be a constant vector, given that the samples generated can be considered to come from an infinite population, as theoretically there is no limit to the number of samples from a generative model like GAN or SDM.

**New dataset by labeling generators output.** The major limitation of evaluating the existing fairness measurement framework is the unavailability of $\boldsymbol{p}^*$. *To pave the way for an accurate evaluation, we create a new dataset by manually labeling the samples generated by GANs and SDM.* More specifically, we utilize the official publicly released pre-trained StyleGAN2 [3] and StyleSwin [4] on CelebA-HQ [27] for sample generation. Then, we randomly sample from these GANs and utilize Amazon Mechanical Turks to hand-label the samples *w.r.t.* Gender and BlackHair, resulting in ≈9K samples for each GAN; see Supp H for more details and examples. Next, we follow a similar labeling process *w.r.t.* Gender, but with a SDM [5] pre-trained on LAION-5B[28]. Here, we input prompts using best practices [26, 29–31], beginning with a scene description ("A photo with the face of"), followed by four indefinite (gender-neutral) pronouns or nouns [32, 33] – {"an individual", "a human being", "one person", "a person"} to collect ≈2k high-quality samples. We refer to this new dataset as Generated Dataset (**GenData**), which includes generated images from three models with corresponding SA labels: GenData-StyleGAN2, GenData-StyleSwin, GenData-SDM. We remark that these labeled datasets only provide a strong approximation of $\boldsymbol{p}^*$ for each generator, however as the datasets are reasonably large, we find this approximation sufficient and simply refer to it as the GT $\boldsymbol{p}^*$. Then utilizing this GT $\boldsymbol{p}^*$, we compare it against the estimated baseline ($\hat{\boldsymbol{p}}$). One interesting observation revealed by GenData is that all three generators exhibit a considerable amount of bias (see Tab.1 and 2); more detail in Sec. 6. Note that for a fair generator we have $p_0^* = p_1^* = 0.5$, and measuring the $p_0^*$ and $p_1^*$ is a good proxy for measuring fairness.

**Experimental setup.** Here, we follow Choi *et al.* [1] as the *Baseline* for measuring fairness. In particular, to calculate each $\hat{\boldsymbol{p}}$ value for a generator, a corresponding batch of $n = 400$ samples is randomly drawn from GenData and passed into $C_{\mathbf{u}}$ for SA classification. We repeat this for $s = 30$ batches and report the mean results denoted by $\mu_{\text{Base}}$ and the 95% confidence interval denoted by $\rho_{\text{Base}}$. For a comprehensive analysis of the GANs, we repeat the experiment using four different SA classifiers: Resnet-18, ResNet-34 [34], MobileNetv2 [35], and VGG-16 [36]. Then, to evaluate the SDM, we utilize CLIP [6] to explore the utilization of pre-trained models for zero-shot SA classification; more details on the CLIP SA classifier in Supp. E. As CLIP does not have a validation dataset, to measure $\boldsymbol{\alpha}$ for CLIP, we utilize CelebA-HQ, a dataset with a similar domain to our application. We found this to be a very accurate approximation; see Supp D.7 for validation results. Note that for SDM, a separate $\hat{\boldsymbol{p}}$ is measured for each text prompt as SDM's output images are conditioned on the input text prompt. As seen in Tab. 1 and 2, all classifiers demonstrate reasonably high average accuracy $\in [84\%, 98.7\%]$. Note that as we focus on binary SA (*e.g.* Gender:{Male, Female}), both $\boldsymbol{p}^*$ and $\hat{\boldsymbol{p}}$ have two components *i.e.* $\boldsymbol{p}^* = \{p_0^*, p_1^*\}$, and $\hat{\boldsymbol{p}} = \{\hat{p}_0, \hat{p}_1\}$. After computing the $\mu_{\text{Base}}$ and $\rho_{\text{Base}}$, we calculate *normalized $L_1$ point error $e_\mu$*, and *interval max error $e_\rho$ w.r.t.* the $p_0^*$ (GT) to evaluate the measurement accuracy of the baseline method:

$$e_{\mu_{\text{Base}}} = \tfrac{1}{p_0^*}|p_0^* - \mu_{\text{Base}}| \quad ; \quad e_{\rho_{\text{Base}}} = \tfrac{1}{p_0^*}\max\{|\min(\rho_{\text{Base}}) - p_0^*|, |\max(\rho_{\text{Base}}) - p_0^*|\} \quad (1)$$

**Based on our results in Tab. 1**, for GANs, we observe that despite the use of reasonably accurate SA classifiers, there are significant estimation errors in the existing fairness measurement framework, *i.e.* $e_{\mu_{\text{Base}}} \in [4.98\%, 17.13\%]$. In particular, looking at the SA classifier with the highest average accuracy of $\approx 97\%$ (ResNet-18 on Gender), we observe significant discrepancies between GT $p_0^*$ and $\mu_{\text{Base}}$, with $e_{\mu_{\text{Base}}} = 4.98\%$. These errors generally worsen as accuracy marginally degrades, *e.g.* MobileNetv2 with accuracy $\approx 96\%$ results in $e_{\mu_{\text{Base}}} = 5.45\%$. These considerably large errors contradict prior assumptions – that for a reasonably accurate SA classifier, we can assume $e_{\mu_{\text{Base}}}$ to be fairly negligible. Similarly, our results in Tab. 2 for the SDM, show large $e_{\mu_{\text{Base}}} \in [1.49\%, 9.14\%]$, even though the classifier is very accurate. We discuss the reason for this in more detail in Sec. 5.1.

*Overall, these results are concerning as they cast doubt on the accuracy of prior reported results.* For example, imp-weighting [1] which uses the same ResNet-18 source code as our experiment, reports a 2.35% relative improvement in fairness against its baseline *w.r.t.* Gender, which falls within the

range of our experiments smallest relative error, $e_{\mu_{\text{Base}}}$=4.98%. Similarly, Teo *et al.* [2] and Um *et al.* [12] report a relative improvement in fairness of 0.32% and 0.75%, compared to imp-weighting [1]. These findings suggest that some prior results may be affected due to oversight of SA classifier's inaccuracies; see Supp. A.4 for more details on how to calculate these measurements.

**Remark:** In this section, we provide the keystone for the evaluation of measurement accuracy in the current framework by introducing a labeled dataset based on generated samples. These evaluation results raise concerns about the accuracy of existing framework as considerable error rates were observed even when using accurate SA classifiers, an issue previously seen to be negligible.

## 4    Mitigating Error in Fairness Measurements

The previous section exposes the inaccuracies in the existing fairness measurement framework. Following that, in this section, we first develop a statistical model for the erroneous output of the SA classifier, $\hat{p}$, to help draw a more systematic relationship between the inaccuracy of the SA classifier and error in fairness estimation. Then, with this statistical model, we propose CLEAM – a new measurement framework that reduces error in the measured $\hat{p}$ by accounting for the SA classifier inaccuracies to output a more accurate statistical approximation of $p^*$.

### 4.1    Proposed Statistical Model for Fairness Measurements

As shown in Fig.1(a), to measure the fairness of the generator, we feed $n$ generated samples to the SA classifier $C_{\mathbf{u}}$. The output of the SA classifier ($\hat{p}$) is in fact a random variable that aims to approximate the $p^*$. Here, we propose a statistical model to derive the distribution of $\hat{p}$.

As Fig.1(b) demonstrates in our running example of a binary SA, each generated sample is from *class 0* with probability $p_0^*$, or from *class 1* with probability $p_1^*$. Then, generated sample from *class i* where $i \in \{0, 1\}$, will be classified correctly with the probability of $\alpha_i$, and wrongly with the probability of $\alpha_i' = 1 - \alpha_i$. Thus, for each sample, there are four mutually exclusive possible events denoted by $\mathbf{c}$, with the corresponding probability vector $\mathbf{p}$:

$$\mathbf{c}^T = \begin{bmatrix} c_{0|0} & c_{1|0} & c_{1|1} & c_{0|1} \end{bmatrix} \quad , \quad \mathbf{p}^T = \begin{bmatrix} p_0^*\alpha_0 & p_0^*\alpha_0' & p_1^*\alpha_1 & p_1^*\alpha_1' \end{bmatrix} \tag{2}$$

where $c_{i|j}$ denotes the event of assigning label $i$ to a sample with GT label $j$. Given that this process is performed independently for each of the $n$ generated images, the probability of the counts for each output $\mathbf{c}^T$ in Eqn. 2 (denoted by $\mathbf{N_c}$) can be modeled by a multinomial distribution, *i.e.* $\mathbf{N_c} \sim Multi(n, \mathbf{p})$ [37–39]. Note that $\mathbf{N_c}$ models the *joint probability distribution* of these outputs, *i.e.* $\mathbf{N_c} \sim \mathbb{P}(N_{c_{0|0}}, N_{c_{1|0}}, N_{c_{1|1}}, N_{c_{0|1}})$ where, $N_{c_{i|j}}$ is the random variable of the count for event $c_{i|j}$ after classifying $n$ generated images. Since $\mathbf{p}$ is not near the boundary of the parameter space, and as we utilize a large $n$, based on the central limit theorem, $Multi(n, \mathbf{p})$ can be approximated by a multivariate Gaussian distribution, $\mathbf{N_c} \sim \mathcal{N}(\boldsymbol{\mu}, \boldsymbol{\Sigma})$, with $\boldsymbol{\mu} = n\mathbf{p}$ and $\boldsymbol{\Sigma} = n\mathbf{M}$ [40, 39], where $\mathbf{M}$ is defined as:

$$\mathbf{M} = diag(\mathbf{p}) - \mathbf{pp}^T \tag{3}$$

$diag(\mathbf{p})$ denotes a square diagonal matrix corresponding to vector $\mathbf{p}$ (see Supp A.1 for expanded form). The *marginal distribution* of this multivariate Gaussian distribution gives us a univariate (one-dimensional) Gaussian distribution for the count of each output $\mathbf{c}^T$ in Eqn. 2. For example, the distribution of the count for event $c_{0|0}$, denoted by $N_{c_{0|0}}$, can be modeled as $N_{c_{0|0}} \sim \mathcal{N}(\boldsymbol{\mu}_1, \boldsymbol{\Sigma}_{11})$.

Lastly, we find the total percentage of data points labeled as class $i$ when labeling $n$ generated images using the normalized sum of the related random variables, *i.e.* $\hat{p}_i = \frac{1}{n}\sum_j N_{c_{i|j}}$. For our binary example, $\hat{p}_i$ can be calculated by summing random variables with Gaussian distribution, which results in another Gaussian distribution [41], *i.e.* , $\hat{p}_0 \sim \mathcal{N}(\tilde{\mu}_{\hat{p}_0}, \tilde{\sigma}_{\hat{p}_0}^2)$, where:

$$\tilde{\mu}_{\hat{p}_0} = \frac{1}{n}(\boldsymbol{\mu}_1 + \boldsymbol{\mu}_4) = p_0^*\alpha_0 + p_1^*\alpha_1' \tag{4}$$

$$\tilde{\sigma}_{\hat{p}_0}^2 = \frac{1}{n^2}(\boldsymbol{\Sigma}_{11} + \boldsymbol{\Sigma}_{44} + 2\boldsymbol{\Sigma}_{14}) = \frac{1}{n}[(p_0^*\alpha_0 - (p_0^*\alpha_0)^2) + (p_1^*\alpha_1' - (p_1^*\alpha_1')^2)] + \frac{2}{n}p_0^*p_1^*\alpha_0\alpha_1' \tag{5}$$

Similarly $\hat{p}_1 \sim \mathcal{N}(\tilde{\mu}_{\hat{p}_1}, \tilde{\sigma}_{\hat{p}_1}^2)$ with $\tilde{\mu}_{\hat{p}_1} = (\boldsymbol{\mu}_2 + \boldsymbol{\mu}_3)/n$, and $\tilde{\sigma}_{\hat{p}_1}^2 = (\boldsymbol{\Sigma}_{22} + \boldsymbol{\Sigma}_{33} + 2\boldsymbol{\Sigma}_{23})/n^2$ which is aligned with the fact that $\hat{p}_1 = 1 - \hat{p}_0$.

---

**Algorithm 1:** Computing point and interval estimates using CLEAM.

---

**Require:** accuracy of SA classifier, $\boldsymbol{\alpha}$.

1  Compute SA classifier output $\hat{\boldsymbol{p}} : \{\hat{\boldsymbol{p}}^1, \ldots, \hat{\boldsymbol{p}}^s\}$ for $s$ batches of generated data.
2  Compute sample mean $\ddot{\mu}_{\hat{p}}$ and sample variance $\ddot{\sigma}_{\hat{p}}^2$ using (6) and (7).
3  Use (8) to compute point estimate $\mu_{\text{CLEAM}}$.
4  Use (10) to compute interval estimate $\rho_{\text{CLEAM}}$.

---

**Remark:** In this section, considering the probability tree diagram in Fig.1(b), we propose a joint distribution for the possible events of classification ($N_{c_{i|j}}$), and use it to compute the marginal distribution of each event, and finally the distribution of the SA classifier outputs ($\hat{p}_0$, and $\hat{p}_1$). Note that considering Eqn. 4, 5, only with a perfect classifier ($\alpha_i = 1$, *i.e.* acc= 100%) the $\tilde{\mu}_{\hat{p}_0}$ converges to $p_0^*$. However, training a perfect SA classifier is not practical *e.g.* due to the lack of an appropriate dataset and task hardness [42, 43]. As a result, in the following, we will propose CLEAM which instead utilizes this statistical model to mitigate the error of the SA classifier.

## 4.2  CLEAM for Accurate Fairness Measurement

In this section, we propose a new estimation method in fairness measurement that considers the inaccuracy of the SA classifier. For this, we use the statistical model, introduced in Sec 4.1, to compute a more accurate estimation of $\boldsymbol{p}^*$. Specifically, we first propose a Point Estimate (PE) by approximating the *maximum likelihood value* of $\boldsymbol{p}^*$. Then, we use the *confidence interval* for the observed data ($\hat{\boldsymbol{p}}$) to propose an Interval Estimate (IE) for $\boldsymbol{p}^*$.

**Point Estimate (PE) for $\boldsymbol{p}^*$.** Suppose that we have access to $s$ samples of $\hat{\boldsymbol{p}}$ denoted by $\{\hat{\boldsymbol{p}}^1, \ldots, \hat{\boldsymbol{p}}^s\}$, *i.e.* SA classification results on $s$ batches of generated data. We can then use the proposed statistical model to approximate the $\boldsymbol{p}^*$. In the previous section, we demonstrate that we can model $\hat{p}_j^i$ using a Gaussian distribution. Considering this, first, we use the available samples to calculate sample-based statistics including the mean and variance of the $\hat{p}_j$ samples:

$$\ddot{\mu}_{\hat{p}_j} = \frac{1}{s} \sum_{i=1}^s \hat{p}_j^i \tag{6}$$

$$\ddot{\sigma}_{\hat{p}_j}^2 = \frac{1}{s-1} \sum_{i=1}^s (\hat{p}_j^i - \ddot{\mu}_{\hat{p}_j})^2 \tag{7}$$

For a Gaussian distribution, the Maximum Likelihood Estimate (MLE) of the population mean is its sample mean $\ddot{\mu}_{\hat{p}_j}$ [44]. Given that $s$ is large enough (*e.g.* $s > 30$), we can assume that $\ddot{\mu}_{\hat{p}_j}$ is a good approximation of the population mean [45], and equate it to the statistical population mean $\tilde{\mu}_{\hat{p}_j}$ in Eqn. 4 (see Supp A.2 for derivation). With that, we get the *maximum likelihood approximation of $\boldsymbol{p}^*$*, which we call the *CLEAM's point estimate, $\mu_{CLEAM}$*:

$$\mu_{\text{CLEAM}}(p_0^*) = (\ddot{\mu}_{\hat{p}_0} - \alpha_1')/(\alpha_0 - \alpha_1') \quad , \quad \mu_{\text{CLEAM}}(p_1^*) = 1 - \mu_{\text{CLEAM}}(p_0^*) \tag{8}$$

Notice that $\mu_{\text{CLEAM}}$ accounts for the inaccuracy of the SA classifier.

**Interval Estimate (IE) for $\boldsymbol{p}^*$.** In the previous part, we propose a PE for $\boldsymbol{p}^*$ using the statistical model, and sample-based mean $\ddot{\mu}_{\hat{p}_0}$. However, as we use only $s$ samples of $\hat{\boldsymbol{p}}$, $\ddot{\mu}_{\hat{p}_0}$ may not capture the exact value of the population mean. This adds some degree of inaccuracy into $\mu_{\text{CLEAM}}$. In fact, in our framework, $\ddot{\mu}_{\hat{p}_0}$ equals $\tilde{\mu}_{\hat{p}_0}$ when $s \to \infty$. However, increasing each unit of $s$ significantly increases the computational complexity, as each $\hat{\boldsymbol{p}}$ requires $n$ generated samples. To address this, we recall that $\hat{p}_0$ follows a Gaussian distribution and instead utilize frequentist statistics [41] to propose a 95% confidence interval (CI) for $\boldsymbol{p}^*$. To do this, first we derive the CI for $\tilde{\mu}_{\hat{p}_0}$:

$$\ddot{\mu}_{\hat{p}_0} - 1.96 \frac{\ddot{\sigma}_{\hat{p}_0}}{\sqrt{s}} \leq \tilde{\mu}_{\hat{p}_0} \leq \ddot{\mu}_{\hat{p}_0} + 1.96 \frac{\ddot{\sigma}_{\hat{p}_0}}{\sqrt{s}} \tag{9}$$

Then, applying Eqn.4 to Eqn.9 gives the lower and upper bounds of the approximated 95% CI for $p_0^*$:

$$\mathcal{L}(p_0^*), \mathcal{U}(p_0^*) = (\ddot{\mu}_{\hat{p}_0} \mp 1.96(\ddot{\sigma}_{\hat{p}_0}/\sqrt{s}) - \alpha_1')/(\alpha_0 - \alpha_1') \tag{10}$$

This gives us the interval estimate of CLEAM, $\rho_{\text{CLEAM}} = [\mathcal{L}(p_0^*), \mathcal{U}(p_0^*)]$, a range of values that we can be approximately 95% confident to contain $p_0^*$. The range of possible values for $p_1^*$ can be simply derived considering $p_1^* = 1 - p_0^*$. The overall procedure of CLEAM is summarized in Alg. 1. Now, with the IE, we can provide statistical significance to the reported fairness improvements.

Table 1: Comparing the *point estimates* and *interval estimates* of Baseline [1], Diversity [46] and our CLEAM in estimating $p^*$ of StyleGAN2 [3] and StyleSwin [4] with the proposed GenData datasets. We utilize SA classifiers Resnet-18/34 (R18, R34)[34], MobileNetv2 (MN2)[35] and VGG-16 (V16)[36], with different accuracies $\alpha$, to classify samples *w.r.t.* attributes `Gender` and `BlackHair`. The $p_0^*$ value of each GAN *w.r.t.* SA is determined by manually hand-labeling the generated data. We repeat this for 5 experimental runs and report the mean error rate, per Eqn. 1. See Supp D.1 for the standard deviation of PE and IE.

| | | | Point Estimate | | | | | | Interval Estimate | | | | |
|---|---|---|---|---|---|---|---|---|---|---|---|---|---|
| | | | Baseline | | Diversity | | CLEAM (Ours) | | Baseline | | Diversity | | CLEAM (Ours) | |
| $\alpha = \{\alpha_0, \alpha_1\}$ | Avg. $\alpha$ | $\mu_{\text{Base}}$ | $e_\mu(\downarrow)$ | $\mu_{\text{Div}}$ | $e_\mu(\downarrow)$ | $\mu_{\text{CLEAM}}$ | $e_\mu(\downarrow)$ | $\rho_{\text{Base}}$ | $e_\rho(\downarrow)$ | $\rho_{\text{Div}}$ | $e_\rho(\downarrow)$ | $\rho_{\text{CLEAM}}$ | $e_\rho(\downarrow)$ |
| (A) StyleGAN2 | | | | | | | | | | | | | |
| Gender with GT class probability $p_0^*$=0.642 | | | | | | | | | | | | | |
| R18 {0.947, 0.983} | 0.97 | 0.610 | 4.98% | — | — | 0.638 | **0.62%** | [0.602, 0.618] | 6.23% | — | — | [0.629, 0.646] | **2.02%** |
| R34 {0.932, 0.976]} | 0.95 | 0.596 | 7.17% | — | — | 0.634 | **1.25%** | [0.589, 0.599] | 8.26% | — | — | [0.628, 0.638] | **2.18%** |
| MN2 {0.938, 0.975} | 0.96 | 0.607 | 5.45% | — | — | 0.637 | **0.78%** | [0.602, 0.612] | 6.23% | — | — | [0.632, 0.643] | **1.56%** |
| V16 {0.801, 0.919} | 0.86 | 0.532 | 17.13% | 0.550 | 14.30% | 0.636 | **0.93%** | [0.526, 0.538] | 18.06% | [0.536 , 0.564] | 16.51% | [0.628, 0.644] | **2.18%** |
| Average Error: | | | 8.68% | | 14.30% | | **0.90%** | | 9.70% | | 16.51% | | **1.99%** |
| BlackHair with GT class probability $p_0^*$=0.643 | | | | | | | | | | | | | |
| R18 {0.869, 0.885} | 0.88 | 0.599 | 6.84% | — | — | 0.641 | **0.31%** | [0.591, 0.607] | 8.08% | — | — | [0.631, 0.652] | **1.40%** |
| R34 {0.834, 0.916} | 0.88 | 0.566 | 11.98% | — | — | 0.644 | **0.16%** | [0.561, 0.572] | 12.75% | — | — | [0.637, 0.651] | **1.24%** |
| MN2 {0.839, 0.881} | 0.86 | 0.579 | 9.95% | — | — | 0.639 | **0.62%** | [0.574, 0.584] | 10.73% | — | — | [0.632, 0.647] | **1.71%** |
| V16 {0.851, 0.836} | 0.84 | 0.603 | 6.22% | 0.582 | 9.49% | 0.640 | **0.47%** | [0.597, 0.608] | 7.15% | [0.568, 0.596] | 11.66% | [0.632, 0.648] | **1.71%** |
| Average Error: | | | 8.75% | | 9.49% | | **0.39%** | | 9.68% | | 11.66% | | **1.52%** |
| (B) StyleSwin | | | | | | | | | | | | | |
| Gender with GT class probability $p_0^*$=0.656 | | | | | | | | | | | | | |
| R18 {0.947, 0.983} | 0.97 | 0.620 | 5.49% | — | — | 0.648 | **1.22%** | [0.612,0.629] | 6.70% | — | — | [0.639,0.658] | **2.59%** |
| R34 {0.932, 0.976} | 0.95 | 0.610 | 7.01% | — | — | 0.649 | **1.07%** | [0.605,0.615] | 7.77% | — | — | [0.643,0.654] | **1.98%** |
| MN2 {0.938, 0.975} | 0.96 | 0.623 | 5.03% | — | — | 0.655 | **0.15%** | [0.618, 0.629] | 5.79% | — | — | [0.649,0.661] | **1.07%** |
| V16 {0.801, 0.919} | 0.86 | 0.555 | 15.39% | 0.562 | 14.33% | 0.668 | **1.83%** | [0.549,0.560] | 16.31% | [0.548,0.576] | 16.46% | [0.660,0.675] | **2.90%** |
| Average Error: | | | 8.23% | | 14.33% | | **1.07%** | | 9.14% | | 16.46% | | **2.14%** |
| BlackHair with GT class probability $p_0^*$=0.668 | | | | | | | | | | | | | |
| R18 {0.869, 0.885} | 0.88 | 0.612 | 8.38% | — | — | 0.659 | **1.35%** | [0.605,0.620] | 9.43% | — | — | [0.649,0.670] | **2.84%** |
| R34 {0.834, 0.916} | 0.88 | 0.581 | 13.02% | — | — | 0.662 | **0.90%** | [0.576,0.586] | 13.77% | — | — | [0.656,0.669] | **1.80%** |
| MN2 {0.839, 0.881} | 0.86 | 0.596 | 10.78% | — | — | 0.659 | **1.35%** | [0.591,0.600] | 11.50% | — | — | [0.652,0.666] | **2.40%** |
| V16 {0.851, 0.836} | 0.84 | 0.625 | 6.44% | 0.608 | 8.98% | 0.677 | **1.35%** | [0.620,0.630] | 7.19% | [0.590,0.626] | 11.68% | [0.670,0.684] | **2.40%** |
| Average Error: | | | 9.66% | | 8.98% | | **1.24%** | | 10.47% | | 11.68% | | **2.36%** |

Table 2: Comparing the *point estimates* and *interval estimates* of Baseline and CLEAM in estimating the $p^*$ of the Stable Diffusion Model [5] with the GenData-SDM dataset. We use prompt input starting with "A photo with the face of" and ending with synonymous (Gender neutral) prompts. We utilized CLIP as the classifier for `Gender`, to obtain $\hat{p}$.

| | | Point Estimate | | | | Interval Estimate | | | |
|---|---|---|---|---|---|---|---|---|---|
| | | Baseline | | CLEAM (Ours) | | Baseline | | CLEAM (Ours) | |
| Prompt | GT $p_0^*$ | $\mu_{\text{Base}}$ | $e_\mu(\downarrow)$ | $\mu_{\text{CLEAM}}$ | $e_\mu(\downarrow)$ | $\rho_{\text{Base}}$ | $e_\rho(\downarrow)$ | $\rho_{\text{CLEAM}}$ | $e_\rho(\downarrow)$ |
| $\alpha$=[0.998,0.975], Avg. $\alpha$=0.987, CLIP −Gender | | | | | | | | | |
| "A photo with the face of an individual" | 0.186 | 0.203 | 9.14% | 0.187 | **0.05%** | [ 0.198 , 0.208 ] | 11.83% | [ 0.182 , 0.192 ] | **3.23%** |
| "A photo with the face of a human being" | 0.262 | 0.277 | 5.73% | 0.263 | **0.38%** | [ 0.270 , 0.285 ] | 8.78% | [ 0.255 , 0.271 ] | **3.44%** |
| "A photo with the face of one person" | 0.226 | 0.241 | 6.63% | 0.230 | **1.77%** | [ 0.232 , 0.251 ] | 11.06% | [ 0.220 , 0.239 ] | **5.75%** |
| "A photo with the face of a person" | 0.548 | 0.556 | 1.49% | 0.548 | **0.00%** | [ 0.545 , 0.566 ] | 3.28% | [ 0.537 , 0.558 ] | **2.01%** |
| Average Error | | | 5.75% | | **0.44%** | | 8.74% | | **3.61%** |

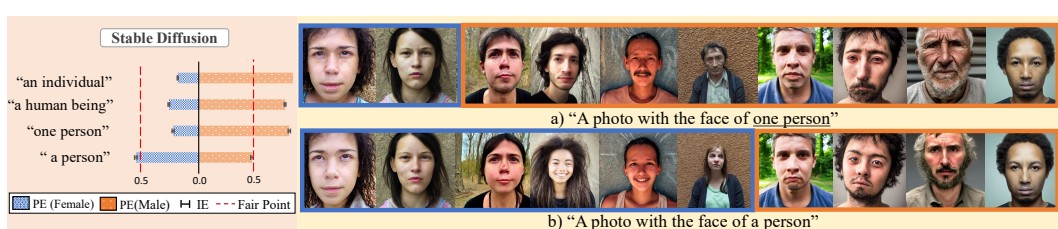

Figure 2: **LHS:** Applying CLEAM to assess `Gender` bias in SDM[5] with CLIP [6]. Here, we utilize synonymous neutral prompts, prefixed with "A photo with the face of __" as the input. We found that subtle changes to the prompts resulted in significant changes in the bias. **RHS:** Illustrating the shift in `Gender` bias. Using the same random seeds (per column), we generate 10 samples from two prompts. Note how in some columns, `Gender` changes while retaining general features *e.g.* pose.

# 5 Experiments

In this section, we first evaluate fairness measurement accuracy of CLEAM on both GANs and SDM (Sec.5.1) with our proposed GenData dataset. Then we evaluate CLEAM's robustness through some ablation studies (Sec. 5.2). To the best of our knowledge, there is no similar literature for improving fairness measurements in generative models. Therefore, we compare **CLEAM** with the two most related works: a) the **Baseline** used in previous works [1, 2, 7, 9, 12] b) **Diversity** [46] which computes disparity within a dataset via an intra-dataset pairwise similarity algorithm. We remark that, as discussed by Keswani *et al.* [46] Diversity is model-specific using VGG-16 [36]; see Supp. D.2 for more details. Finally, unless specified, we repeat the experiments with $s = 30$ batches of images from the generators with batch size $n = 400$. For a fair comparison, all three algorithms use the exact same inputs. However, while Baseline and Diversity ignore the SA classifier inaccuracies, CLEAM makes good use of it to rectify the measurement error. As mentioned in Sec. 4.2, for CLEAM, we utilize $\boldsymbol{\alpha}$ measured on real samples, which we found to be a good approximation of the $\boldsymbol{\alpha}$ measured on generated samples (see Supp. D.7 for results). We repeat each experiment 5 times and report the mean value for each test point for both PE and IE. See Supp D.1 for the standard deviation.

## 5.1 Evaluating CLEAM's Performance

**CLEAM for fairness measurement of SOTA GANs – StyleGAN2 and StyleSwin.** For a fair comparison, we first compute $s$ samples of $\hat{\boldsymbol{p}}$, one for each batch of $n$ images. For Baseline, we use the mean $\hat{\boldsymbol{p}}$ value as the PE (denoted by $\mu_{\text{Base}}$), and the $95\%$ confidence interval as IE ($\rho_{\text{Base}}$). With the same $s$ samples of $\hat{\boldsymbol{p}}$, we apply Alg. 1 to obtain $\mu_{\text{CLEAM}}$ and $\rho_{\text{CLEAM}}$. For Diversity, following the original source code [46], a controlled dataset with fair representation is randomly selected from a held-out dataset of CelebA-HQ [27]. Then, we use a VGG-16 [36] feature extractor and compute Diversity, $\delta$. With $\delta$ we find $\hat{p}_0 = (\delta + 1)/2$ and subsequently $\mu_{\text{Div}}$ and $\rho_{\text{Div}}$ from the mean and $95\%$ CI (see Supp D.2 for more details on diversity). We then compute $e_{\mu_{\text{CLEAM}}}$, $e_{\mu_{\text{Div}}}$, $e_{\rho_{\text{CLEAM}}}$ and $e_{\rho_{\text{Div}}}$ with Eqn 1, by replacing the Baseline estimates with CLEAM and Diversity.

As discussed, our results in Tab.1 show that the baseline experiences significantly large errors of $4.98\% \leq e_{\mu_{\text{Base}}} \leq 17.13\%$, due to a lack of consideration for the inaccuracies of the SA classifier. We note that this problem is prevalent throughout the different SA classifier architectures, even with higher capacity classifiers *e.g.* ResNet-34. Diversity, a method similarly unaware of the inaccuracies of the SA classifier, presents a similar issue with $8.98\% \leq e_{\mu_{\text{Div}}} \leq 14.33\%$ In contrast, CLEAM dramatically reduces the error for all classifier architectures. Specifically, CLEAM reduces the average point estimate error from $e_{\mu_{\text{Base}}} \geq 8.23\%$ to $e_{\mu_{\text{CLEAM}}} \leq 1.24\%$, in both StyleGAN2 and StyleSwin. The IE presents similar results, where in most cases $\rho_{\text{CLEAM}}$ bounds the GT value of $\boldsymbol{p^*}$.

**CLEAM for fairness measurement of SDM.** We evaluate CLEAM in estimating the bias of the SDM *w.r.t.* `Gender`, based on the synonymous (gender-neutral) prompts introduced in Sec. 3. Recall that here we utilize CLIP as the zero-shot SA classifier. Our results in Tab 2, as discussed, show that utilizing the baseline results in considerable error ($1.49\% \leq e_{\mu_{\text{Base}}} \leq 9.14\%$) for all prompts, even though the SA classifier's average accuracy was high, $\approx 98.7\%$ (visual results in Fig.2). A closer look at the theoretical model's Eqn. 4 reveals that this is due to the larger inaccuracies observed in the biased class ($\alpha_1'$) coupled with the large bias seen in $p_1^*$, which results in $\mu_{\text{Base}}$ deviating from $p_0^*$. In contrast, CLEAM accounts for these inaccuracies and significantly minimizes the error to $e_{\mu_{\text{CLEAM}}} \leq 1.77\%$. Moreover, CLEAM's IE is able to consistently bound the GT value of $p_0^*$.

## 5.2 Ablation Studies and Analysis

Here, we perform the ablation studies and compare CLEAM with classifier correction methods. ***We remark that detailed results of these experiments are provided in the Supp due to space limitations.***

**CLEAM for measuring varying degrees of bias.** As we cannot control the bias in trained generative models, to simulate different degrees of bias, we evaluate CLEAM with a *pseudo-generator*. Our results show that CLEAM is effective at different biases ($p_0^* \in [0.5, 0.9]$) reducing the average error from $2.80\% \leq e_{\mu_{\text{Base}}} \leq 6.93\%$ to $e_{\mu_{\text{CLEAM}}} \leq 0.75\%$ on CelebA [47] *w.r.t.* {`Gender,BlackHair`}, and AFHQ [48] *w.r.t.* `Cat/Dog`. See Supp D.3 and D.4 for full experimental results.

**CLEAM vs Classifier Correction Methods [49]**. CLEAM generally accounts for the classifier's inaccuracies, without targeting any particular cause of inaccuracies, for the purpose of rectifying the fairness measurements. This objective is unlike traditional classifier correction methods as it does not aim to improve the actual classifier's accuracy. However, considering that classifier correction methods may improve the fairness measurements by directly rectifying the classifier inaccuracies, we compare its performance against CLEAM. As an example, we utilize the Black Box Shift Estimation / Correction (BBSE / BBSC) [49] which considers the label shift problem and aims to correct the classifier output by detecting the distribution shift. Our results, based on Sec. 5.1 setup, show that while BBSE does improve on the fairness measurements of the baseline *i.e.* $4.20\% \leq e_{\mu_{\text{BBSE}}} \leq 3.38\%$, these results are far inferior to CLEAM's results seen in Tab. 1. In contrast, BBSC demonstrates no improvements in fairness measurements. See Supp D.8 for full experimental results. We postulate that this is likely due to the strong assumption of label shift made by both methods.

**Effect of batch-size.** Utilizing experimental setup in Sec. 5.1 for batch size $n \in [100, 600]$, our results in Fig. 3 show that $n$=400 is an ideal batch size, balancing computational cost and measurement accuracy. See Supp F for full experimental details and results.

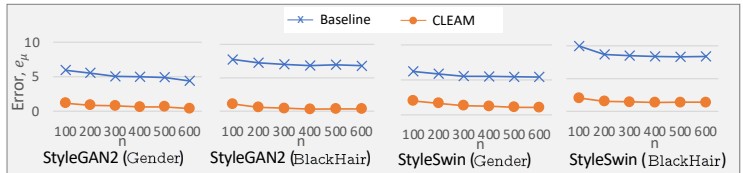

Figure 3: Comparing the point error $e_{\mu}$ for Baseline and CLEAM when evaluating the bias of GenData-CelebA with ResNet-18, while varying sample size, $n$.

## 6 Applying CLEAM: Bias in Current SOTA Generative Models

In this section, we leverage the improved reliability of CLEAM to study biases in the popular generative models. Firstly, with the rise in popularity of text-to-image generators [5, 50–52], we revisit our results when passing different prompts, with synonymous neutral meanings to an SDM, and take a closer look at how subtle prompt changes can impact bias *w.r.t.* Gender. Furthermore, we further investigate if similar results would occur in other SA, Smiling. Secondly, with the shift in popularity from convolution to transformer-based architectures [53–55], due to its better sample quality, we determine whether the learned bias would also change. For this, we compare StylesSwin (transformer) and StyleGAN2 (convolution), which are both based on the same architecture backbone.

Our results, on SDM, demonstrate that the use of different synonymous neutral prompts [32, 33] results in different degrees of bias *w.r.t.* both Gender and Smiling attributes. For example in Fig. 2, a semantically insignificant prompt change from "one person" to "a person" results in a significant shift in Gender bias. Then, in Fig. 4a, we observe that while the SDM *w.r.t.* our prompts appear to be heavily biased to not-Smiling, having "person" in the prompt appears to significantly reduce this bias. This suggests that for SDM, even semantically similar neutral prompts [32, 33] could result in different degrees of bias, thereby demonstrating certain instability in SDM. Next, our results in Fig. 4b compare the bias in StyleGAN2, StylesSwin, and the training CelebA-HQ dataset over an extended number of SAs. Overall, we found that while StyleSwin produces better quality samples [4], the same biases still remain statistically unchanged between the two architectures *i.e.* their IE overlap. Interestingly, our results also found that both the GANs were less biased than the training dataset itself.

## 7 Discussion

**Conclusion.** In this work, we address the limitations of the existing fairness measurement framework. Since generated samples are typically unlabeled, we first introduce a new labeled dataset based on three state-of-the-art generative models for our studies. Our findings suggest that the existing framework, which ignores classification inaccuracies, suffers from significant measurement errors, even when the SA classifier is very accurate. To rectify this, we propose CLEAM, which considers these inaccuracies in its statistical model and outputs a more accurate fairness measurement. Overall, CLEAM demonstrates improved accuracy over extensive experimentation, including both real

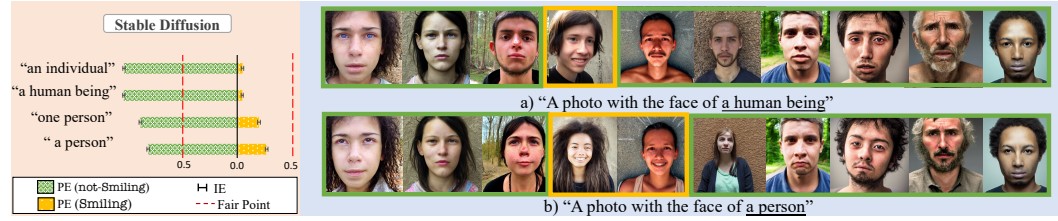

(a) Evaluating bias of SDM [5] *w.r.t.* SA `Smiling` using CLEAM. **LHS:** Bias value with different text prompts as input to SDM. CLEAM identifies that having "person" in the prompt results in more `Smiling` samples generated. **RHS:** Visual comparison of the samples generated for two different but semantically similar text prompts. Samples in the same column are generated using the same random seed. Notice how the SA (`smiling`) of the samples changes with a slight difference in the prompts.

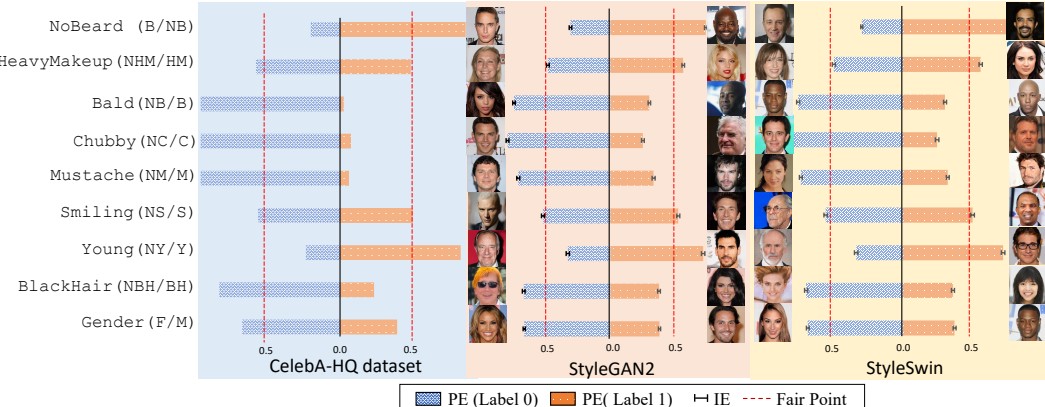

(b) CLEAM on StyleGAN2 [3] and StyleSwin [4], both pre-trained on CelebA-HQ but are based on different architecture variants. We utilize a ResNet-18 and compare the bias *w.r.t.* various SAs.

Figure 4: Applying CLEAM to further assess the bias in popular generative models.

generators and controlled setups. Moreover, by applying CLEAM to popular generative models, we uncover significant biases that raise efficacy concerns about these models' real-world application.

**Broader Impact.** Given that generative models are becoming more widely integrated into our everyday society *e.g.* text-to-image generation, it is important that we have reliable means to measure fairness in generative models, thereby allowing us to prevent these biases from proliferating into new technologies. CLEAM provides a step in this direction by allowing for more accurate evaluation. We remark that our work *does not introduce any social harm* but instead improves on the already existing measurement framework.

**Limitations.** Despite the effectiveness of the proposed method along various generative models, our work addresses only one facet of the problems in the existing fairness measurement and there is still room for further improvement. For instance, it may be beneficial to consider SA to be non-binary *e.g.* when hair color is not necessary fully black (grey). Additionally, existing datasets used to train classifiers are commonly human-annotated, which may itself contain certain notions of bias. See Supp. I for further discussion.

# Acknowledgements

This research is supported by the National Research Foundation, Singapore under its AI Singapore Programmes (AISG Award No.: AISG2-TC-2022-007) and SUTD project PIE-SGP-AI-2018-01. This research work is also supported by the Agency for Science, Technology and Research (A*STAR) under its MTC Programmatic Funds (Grant No. M23L7b0021). This material is based on the research/work support in part by the Changi General Hospital and Singapore University of Technology and Design, under the HealthTech Innovation Fund (HTIF Award No. CGH-SUTD-2021-004). We thank anonymous reviewers for their insightful feedback and discussion.

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
