# On Measuring Fairness in Generative Models
# —Supplementary Material—

**Christopher T. H. Teo**
christopher_teo@mymail.sutd.edu.sg

**Milad Abdollahzadeh**
milad_abdollahzadeh@sutd.sg

**Ngai-Man Cheung**[*]
ngaiman_cheung@sutd.edu.sg

Singapore University of Technology and Design (SUTD)

This supplementary provides additional experiments as well as details that are required to reproduce our results. These were not included in the main paper due to space limitations. The supplementary is arranged as follows:

- **Section A**: Details on Modelling
  - **Section A.1** Details of Theoretical Modelling
  - **Section A.2** Additional Details on CLEAM Algorithm
  - **Section A.3** Details on Fairness Metric
  - **Section A.4** Details of Significance of the Baseline Errors
- **Section B**: Deeper Analysis on Error in Fairness Measurement
- **Section C**: Validating Statistical Model for Classifier Output
  - **Section C.1** Validation of Sample-Based Estimate vs Model-Based Estimate
  - **Section C.2** Goodness-of-Fit Test: $\hat{p}$ from the Real GANs with Our Theoretical Model
- **Section D**: Additional Experimental Results
  - **Section D.1** Experimental Results with Standard Deviation
  - **Section D.2** Experimental Setup for Diversity
  - **Section D.3** Measuring Varying Degrees of Bias (`Gender` and `BlackHair`)
  - **Section D.4** Measuring Varying Degrees of Bias with Additional Sensitive Attributes (`Young` and `Attractive`)
  - **Section D.5** Measuring Varying Degrees of Bias with Additional Sensitive Attribute Classifiers (MobileNetV2)
  - **Section D.6** Measuring SOTA GANs and Diffusion Models with Additional Classifier (CLIP)
  - **Section D.7** Comparing Classifiers Accuracy on Validation Dataset vs Generated Dataset
  - **Section D.8** Comparing CLEAM with Classifier Correction Techniques (BBSE/BBSC)
  - **Section D.9** Applying CLEAM to Re-evaluate Bias Mitigation Algorithms
- **Section E:** Details on Applying CLIP as a SA Classifier
- **Section F:** Ablation Study: Details of Hyper-Parameter Settings and Selection
- **Section G:** Related work
- **Section H:** Details of the GenData: A New Dataset of Labeled Generated Images
- **Section I:** Limitations and Considerations

---

[*]Corresponding Author

37th Conference on Neural Information Processing Systems (NeurIPS 2023).

# A Details on Modelling

## A.1 Details of Theoretical Modelling

In Sec 4.1 of main paper, we have proposed a statistical model for the sensitive attribute classifier output which is then used in CLEAM to rectify current measurement method. In this section, we give more details of this model which is not included in the main paper due to lack of space.

Recall that in main paper, we mentioned that there are four possible mutually exclusive outputs $\mathbf{c}$ for each sample with corresponding probability vector $\mathbf{p}$:

$$\mathbf{c} = \begin{bmatrix} c_{0|0} \\ c_{1|0} \\ c_{1|1} \\ c_{0|1} \end{bmatrix}; \quad \mathbf{p} = \begin{bmatrix} p_0^* \alpha_0 \\ p_0^* \alpha_0' \\ p_1^* \alpha_1 \\ p_1^* \alpha_1' \end{bmatrix}$$

where $c_{i|j}$ denotes the event of assigning label $i$ to a sample with GT label $j$. Then, we mentioned that the count for each ouptut can be modeled as a multinomial distribution, $\mathbf{N_c} \sim Multi(n, \mathbf{p})$. Note that $\mathbf{N_c} = [N_{c_{0|0}}, N_{c_{1|0}}, N_{c_{1|1}}, N_{c_{0|1}}]^T$ is the random vector of counts for individual outputs of $\mathbf{c}$. $N_{c_{i|j}}$ is the random variable of the count for event $c_{i|j}$ after classifying $n$ generated images. First, we consider following assumptions:

1. **Classifiers are reasonably accurate.** We state that, given the advancement in classifiers architecture, and the assumption that the sensitive attribute classifier is trained with proper training procedures, it is a reasonable assumption that it achieves reasonable accuracy and hence, $\alpha_0 \neq 0$ and $\alpha_1 \neq 0$. Similarly, we assume that it is highly unlikely to have a perfect classifier and as such $\alpha_0' = 1 - \alpha_0 \neq 0$ and $\alpha_1' = 1 - \alpha_1 \neq 0$.

2. **Generators are not completely biased.** Given that a generator is trained on a reliable dataset with the availability of all classes of a given sensitive attribute, coupled with the advancement in generator's architecture, it is a fair assumption that the generator would learn some representation of each class in the sensitive attribute and not be completely biased, as such $p_0^* \neq 0$ and $p_1^* \neq 0$.

Based on this assumptions, $\mathbf{p}$ is not near the boundary of the parameter space, and we can conclude that $0 < \mathbf{p} < 1$. Therefore, we can approximate the multinomial distribution as a Gaussian, $\mathbf{N_c} \sim \mathcal{N}(\boldsymbol{\mu}, \boldsymbol{\Sigma})$, with $\boldsymbol{\mu} = n\mathbf{p}$ and $\boldsymbol{\Sigma} = n\mathbf{M}$ [1], where

$$\boldsymbol{M} = \begin{bmatrix} p_0^* \alpha_0 & 0 & 0 & 0 \\ 0 & p_0^* \alpha_0' & 0 & 0 \\ 0 & 0 & p_1^* \alpha_1 & 0 \\ 0 & 0 & 0 & p_1^* \alpha_1' \end{bmatrix} - \begin{bmatrix} (p_0^* \alpha_0)^2 & (p_0^*)^2 \alpha_0 \alpha_0' & p_0^* p_1^* \alpha_0 \alpha_1 & p_0^* p_1^* \alpha_0 \alpha_1' \\ (p_0^*)^2 \alpha_0 \alpha_0' & (p_0^* \alpha_0')^2 & p_0^* p_1^* \alpha_0' \alpha_1 & p_0^* p_1^* \alpha_0' \alpha_1' \\ p_0^* p_1^* \alpha_0 \alpha_1 & p_0^* p_1^* \alpha_0' \alpha_1 & (p_1^* \alpha_1)^2 & (p_1^*)^2 \alpha_1 \alpha_1' \\ p_0^* p_1^* \alpha_0 \alpha_1' & p_0^* p_1^* \alpha_0' \alpha_1' & (p_1^*)^2 \alpha_1 \alpha_1' & (p_1^* \alpha_1')^2 \end{bmatrix}$$

and therefore:

$$\boldsymbol{\mu} = \begin{bmatrix} \boldsymbol{\mu}_1 \\ \boldsymbol{\mu}_2 \\ \boldsymbol{\mu}_3 \\ \boldsymbol{\mu}_4 \end{bmatrix} = n \begin{bmatrix} p_0^* \alpha_0 \\ p_0^* \alpha_0' \\ p_1^* \alpha_1 \\ p_1^* \alpha_1' \end{bmatrix}$$

$$\boldsymbol{\Sigma} = \begin{bmatrix} \boldsymbol{\Sigma}_{11} & \boldsymbol{\Sigma}_{12} & \boldsymbol{\Sigma}_{13} & \boldsymbol{\Sigma}_{14} \\ \boldsymbol{\Sigma}_{21} & \boldsymbol{\Sigma}_{22} & \boldsymbol{\Sigma}_{23} & \boldsymbol{\Sigma}_{24} \\ \boldsymbol{\Sigma}_{31} & \boldsymbol{\Sigma}_{32} & \boldsymbol{\Sigma}_{33} & \boldsymbol{\Sigma}_{34} \\ \boldsymbol{\Sigma}_{41} & \boldsymbol{\Sigma}_{42} & \boldsymbol{\Sigma}_{43} & \boldsymbol{\Sigma}_{44} \end{bmatrix} = n \begin{bmatrix} p_0^* \alpha_0 - (p_0^* \alpha_0)^2 & (p_0^*)^2 \alpha_0 \alpha_0' & p_0^* p_1^* \alpha_0 \alpha_1 & p_0^* p_1^* \alpha_0 \alpha_1' \\ (p_0^*)^2 \alpha_0 \alpha_0' & p_0^* \alpha_0' - (p_0^* \alpha_0')^2 & p_0^* p_1^* \alpha_0' \alpha_1 & p_0^* p_1^* \alpha_0' \alpha_1' \\ p_0^* p_1^* \alpha_0 \alpha_1 & p_0^* p_1^* \alpha_0' \alpha_1 & p_1^* \alpha_1 - (p_1^* \alpha_1)^2 & (p_1^*)^2 \alpha_1 \alpha_1' \\ p_0^* p_1^* \alpha_0 \alpha_1' & p_0^* p_1^* \alpha_0' \alpha_1' & (p_1^*)^2 \alpha_1 \alpha_1' & p_1^* \alpha_1' - (p_1^* \alpha_1')^2 \end{bmatrix}$$

With this, we note that the *marginal distribution* of this multivariate Gaussian distribution gives us a univariate (one-dimensional) Gaussian distribution for the count of each output in $\mathbf{c}^T$. For example, the distribution of the count for event $c_{0|0}$, denoted by $N_{c_{0|0}}$, can be modeled as $N_{c_{0|0}} \sim \mathcal{N}(\boldsymbol{\mu}_1, \boldsymbol{\Sigma}_{11})$. Then, we find the total rate of data points labeled as class $i$ when labeling $n$ generated images using the normalized sum of the related random variables *i.e.* $\hat{p}_i = \frac{1}{n} \sum_j N_{c_{i|j}}$. More specifically:

$$\hat{p}_0 = \frac{1}{n}(\mathbf{N_{c_{0|0}}} + \mathbf{N_{c_{0|1}}}) \sim \mathcal{N}(\tilde{\mu}_{\hat{p}_0}, \tilde{\sigma}_{\hat{p}_0}^2)$$

$$\tilde{\mu}_{\hat{p}_0} = (p_0^* \alpha_0 + p_1^* \alpha_1')$$

$$\tilde{\sigma}_{\hat{p}_0}^2 = \frac{1}{n}((p_0^* \alpha_0 - (p_0^* \alpha_0)^2) + (p_1^* \alpha_1' - (p_1^* \alpha_1')^2) + 2p_0^* p_1^* \alpha_0 \alpha_1')$$

$$\hat{p}_1 = \frac{1}{n}(\mathbf{N}_{\mathbf{c}_{1|0}} + \mathbf{N}_{\mathbf{c}_{1|1}}) \sim \mathcal{N}(\tilde{\mu}_{\hat{p}_1}, \tilde{\sigma}_{\hat{p}_1}^2)$$

$$\tilde{\mu}_{\hat{p}_1} = (p_0^*\alpha_0' + p_1^*\alpha_1)$$

$$\tilde{\sigma}_{\hat{p}_1}^2 = \frac{1}{n}((p_0^*\alpha_0' - (p_0^*\alpha_0')^2) + (p_1^*\alpha_1 - (p_1^*\alpha_1)^2) + 2p_0^*p_1^*\alpha_0'\alpha_1)$$

**Remark:** In Sec 4.1 of the main paper, considering the probability tree diagram in Fig.1(b) (main paper), we proposed a distribution for the possible events of classification ($c_{i|j}$), and used it to compute distribution of each event, and finally the distribution of the output of the sensitive attribute classifier ($\hat{p}_0$, and $\hat{p}_1$). Here, we provide more information on the necessary assumptions and the expanded forms of the equations. In the following Sec. A.2, we will similarly provide more information on proposed CLEAM, presented in Sec. 4.2 of the main paper, which utilizes this statistical model to mitigate the sensitive attribute classifier's error.

## A.2 Additional Details on CLEAM Algorithm

**MLE value of Population Mean.** In this section, first, we discuss the maximum likelihood estimate (MLE) of the population mean for a Gaussian distribution. Given a Gaussian distribution with the population mean $\tilde{\mu}_{\hat{p}_0}$ and standard deviation $\tilde{\sigma}_{\hat{p}_0}$, we can first find the joint probability distribution from the product of each probabilistic outcome (we introduce the natural log as a monotonic function, for ease of calculation). Then, to find the MLE of $\tilde{\mu}_{\hat{p}_0}$, we take the partial derivative of this joint distribution *w.r.t.* $\tilde{\mu}_{\hat{p}_0}$, and solve for its maximum value. This maximum value is equal to the sample mean, $\ddot{\mu}_{\hat{p}_0}$, as detailed below:

$$\frac{\partial}{\partial\tilde{\mu}_{\hat{p}_0}} \prod_{i=1}^{s} ln(\frac{1}{\tilde{\sigma}_{\hat{p}_0}\sqrt{2\pi}} e^{\frac{-(\hat{p}_0^i - \tilde{\mu}_{\hat{p}_0})^2}{2\tilde{\sigma}_{\hat{p}_0}^2}}) = 0$$

$$\frac{1}{\tilde{\sigma}_{\hat{p}_0}^2} \sum_{i=0}^{s}(\hat{p}_0^i - \tilde{\mu}_{\hat{p}_0}) = 0$$

$$\tilde{\mu}_{\hat{p}_0} = \frac{1}{s} \sum_{i}^{s} \hat{p}_0^i = \ddot{\mu}_{\hat{p}_0}$$

**Point Estimate of CLEAM.** From this, given that $s$ is sufficiently large, we utilize the sample mean as the maximum likelihood approximate of the population mean. As the population mean was modeled in Sec. A.1, we can equate the sample mean to the expanded theoretical model:

$$\ddot{\mu}_{\hat{p}_0} = \tilde{\mu}_{\hat{p}_0} = p_0^*\alpha_0 + (1 - p_0^*)\alpha_1'$$

Now given that the classifier's accuracy $\boldsymbol{\alpha} = [\alpha_0, \alpha_1]$ and the sample mean $\ddot{\mu}_{\hat{p}_0}$ can be measured, we are able to solve for the maximum likelihood point estimate of $p_0^*$, which we denoted with $\mu_{\text{CLEAM}}$ as follows:

$$\mu_{\text{CLEAM}} = \frac{\ddot{\mu}_{\hat{p}_0} - \alpha_1'}{\alpha_0 - \alpha_1'} = \frac{\ddot{\mu}_{\hat{p}_0} - 1 + \alpha_1}{\alpha_0 - 1 + \alpha_1}$$

Note that we compute $\mu_{\text{CLEAM}}$ *w.r.t.* $p_0^*$ *i.e.* $\mu_{\text{CLEAM}}(p_0^*)$ through-out this paper for ease of discussion, however as $p_1^* = 1 - p_0^*$, a similar $\mu_{\text{CLEAM}}$ *w.r.t.* $p_1^*$ *i.e.* $\mu_{\text{CLEAM}}(p_1^*)$ can be found with $\mu_{\text{CLEAM}}(p_1^*) = 1 - \mu_{\text{CLEAM}}(p_0^*)$.

**Interval Estimate of CLEAM.** We acknowledge that there exist other statistically probable solutions for $\boldsymbol{p}^*$ that could output the $s$ $\hat{\boldsymbol{p}}$ samples, other than the Maximum likelihood point estimate of $\boldsymbol{p}^*$. We thus propose the following approximation for the $95\%$ confidence interval of $\boldsymbol{p}^*$. Recall the notations $\ddot{\mu}_{\hat{p}_0}$ and $\ddot{\sigma}_{\hat{p}_0}$ are the sample mean and standard deviation respectively:

$$\ddot{\mu}_{\hat{p}_0} = \frac{1}{s} \sum_{i}^{s} \hat{p}_0^i \quad ; \quad \ddot{\sigma}_{\hat{p}_0} = \sqrt{\frac{\sum_{i=1}^{s}(\hat{p}_0^i - \ddot{\mu}_{\hat{p}_0})^2}{s - 1}}$$

Since $\hat{\boldsymbol{p}}$ follows a Gaussian distribution, we can propose the following equation:

$$Pr(-z_{\frac{\delta}{2}} \leq \frac{\ddot{\mu}_{\hat{p}_0} - \tilde{\mu}_{\hat{p}_0}}{\frac{\ddot{\sigma}_{\hat{p}_0}}{\sqrt{s}}} \leq \mathbf{z}_{\frac{\delta}{2}}) = 1 - \delta$$

Solving for $\tilde{\mu}_{\hat{p}}$, we get:

$$Pr(\ddot{\mu}_{\hat{p}_0} + z_{\frac{\delta}{2}}(\frac{\ddot{\sigma}_{\hat{p}_0}}{\sqrt{s}}) \geq \tilde{\mu}_{\hat{p}_0} \geq \ddot{\mu}_{\hat{p}_0} - z_{\frac{\alpha}{2}}\frac{\ddot{\sigma}_{\hat{p}_0}}{\sqrt{s}}) = 1 - \delta$$

Then, given that $\tilde{\mu}_{\hat{p}} = p_0^*\alpha_0 + p_1^*\alpha_1' = p_0^*(\alpha_0 - \alpha_1') + \alpha_1'$ we formulate the following:

$$Pr(\frac{\ddot{\mu}_{\hat{p}_0} + z_{\frac{\delta}{2}}(\frac{\ddot{\sigma}_{\hat{p}_0}}{\sqrt{s}}) - \alpha_1'}{\alpha_0 - \alpha_1'} \geq p_0^* \geq \frac{\ddot{\mu}_{\hat{p}_0} - z_{\frac{\alpha}{2}}\frac{\ddot{\sigma}_{\hat{p}_0}}{\sqrt{s}} - \alpha_1'}{\alpha_0 - \alpha_1'}) = 1 - \delta \tag{1}$$

As such when $\delta = 0.05$, we can determine that the $95\%$ approximated confidence interval of $p_0^*$ is :

$$\rho_{\text{CLEAM}}(p_0^*) = [\mathcal{L}(p_0^*), \mathcal{U}(p_0^*)] = [\frac{\ddot{\mu}_{\hat{p}_0} - 1.96(\frac{\ddot{\sigma}_{\hat{p}_0}}{\sqrt{s}}) - \alpha_1'}{\alpha_0 - \alpha_1'} \quad , \quad \frac{\ddot{\mu}_{\hat{p}_0} + 1.96\frac{\ddot{\sigma}_{\hat{p}_0}}{\sqrt{s}} - \alpha_1'}{\alpha_0 - \alpha_1'}]$$

**Extending the point estimate to a multiple label setup.** We remark that in current literature, fairness of generative models has been studied for binary sensitive attributes mainly due to the lack of availability of a large labeled dataset needed for systematic experimentation. As a result, CLEAM similarly focuses on binary SA to address a common flaw in the evaluation process of the many proposed State-of-the-Art methods.

Assuming that the constraint of the dataset is addressed, our same CLEAM approach can be easily extended to a multi-label setup. For example, given a 3 label sensitive attribute where $p_j^*$ is the probability of generating a sample with label $j$ and $\alpha_{i|j}$ denotes the probability ("accuracy") of the SA classifier in classifying a sample with GT label $j$ as $i$ for $i,j \in \{0,1,2\}$. Fig. 1 shows our statistical model for this setting. We can then similarly solve for the point estimate by solving the matrix:

$$\begin{bmatrix} \alpha_{0|0} & \alpha_{0|1} & \alpha_{0|2} \\ \alpha_{1|0} & \alpha_{1|1} & \alpha_{1|2} \\ \alpha_{2|0} & \alpha_{2|1} & \alpha_{2|2} \end{bmatrix} \begin{bmatrix} p_0^* \\ p_1^* \\ p_2^* \end{bmatrix} = \begin{bmatrix} \ddot{\mu}_{\hat{p}_0} \\ \ddot{\mu}_{\hat{p}_1} \\ \ddot{\mu}_{\hat{p}_2} \end{bmatrix} \tag{2}$$

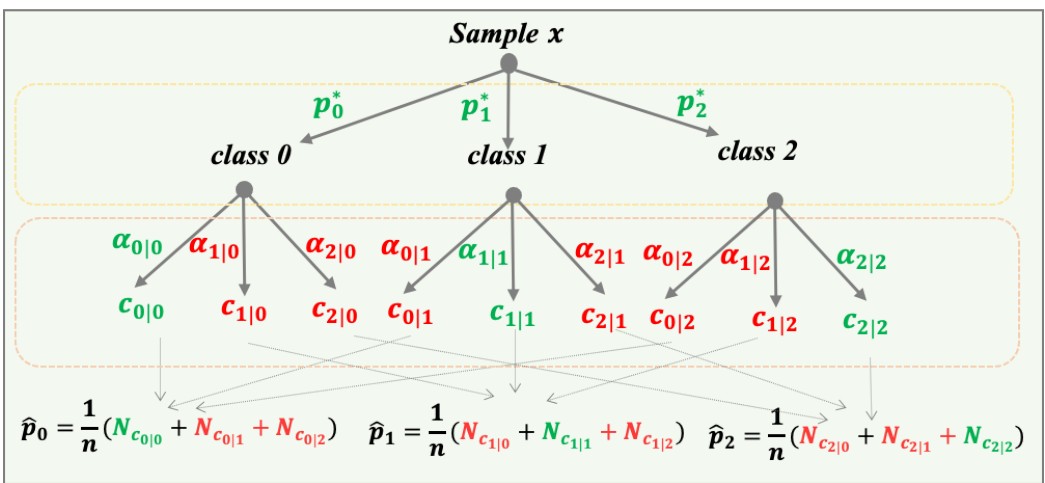

Figure 1: Our statistical model for fairness measurement when considering a multi-label SA. For illustration purposes, we utilize 3 labels for a given SA. Note that, **our same approach can be applied to other multi-label settings.** This statistical model accounts for inaccuracies in the SA classifier and is the base of our proposed CLEAM (see Sec. 4.1). Here, $p_j^*$ is the ground truth probability of a generator outputting a sample with label $j$ and $\alpha_{i|j}$ denotes the probability ("accuracy") of the SA classifier classifying a sample with GT label $j$ as $i$ for $i, j \in \{0, 1, 2\}$.

### A.3 Details on Fairness Metric

Fairness in generative models is defined as *Equal Representation* meaning that the generator is supposed to generate an equal number of samples for each element of an attribute, e.g., an equal number

of generated `Male` and `Female` samples when the sensitive attribute is `Gender`. Therefore, the expected distribution for a fair generator is usually a uniform distribution denoted by $\bar{p}$. Considering this, the fairness discrepancy (FD) metric [2] measures the L2 norm between $\bar{p}$ and the estimated class probability of the generator by each measurement method, *i.e.* $\mu_{\dagger}$, where $\dagger \in \{$`Base`, `CLEAM`, `Div`$\}$, as follows:

$$f = |\bar{p} - \mu_{\dagger}|_2 \tag{3}$$

Note that for a fair generator the fairness discrepancy $f$ would be zero, which also indicates zero bias.

## A.4 Details of Significance of the Baseline Errors

In the main manuscript (Sec. 3 of the main paper), we discussed that the relative improvement of the previous fair generative models could be small, e.g. Teo *et al.* [3] and Um *et al.* [4] report a relative improvement in the fairness of 0.32% and 0.75%, compared to imp-weighting [2], and they fall within the range of our experiment's smallest relative error, $e_{\mu_{Base}}$=4.98%. Here, we provide more detail on how we calculate the relative improvements in the main manuscript. Specifically, we calculate the relative change of the proposed work against the previous work with the following:

$$Relaitve\ Improvement = \frac{|(\hat{p}_0\ of\ previous\ work) - (\hat{p}_0\ of\ proposed\ work)|}{(\hat{p}_0\ of\ previous\ work)} \tag{4}$$

Notice that this is similar to $e_{\mu_{Base}}$ of Eqn. 1 in the main paper. For example, Teo *et al.* (Tab. 1 90_10 and perc=0.1 settings) [3] reports that fairTL measures a $f = 0.105$ which is compared against the previous work's (Choi *et al.* [2]) $f = 0.107$. Utilizing Eqn. 3, we find that this is equivalent to $p_0^* = 0.4257$ or $0.5743$, and $0.4243$ or $0.5757$, respectively. We remark that here we report two values per $f$, as the FD metric is a symmetric metric. Then applying Eqn. 4, and taking the maximum of the values, we find the relative improvement to be $0.32\%$, at best. Note that as we mentioned in the main paper for this setup the baseline measurement framework results in $4.98\%$ error rate (with the best performing sensitive attribute classifier), meaning that it may not be reliable for gauging the improvement.

# B  Deeper Analysis on Error in Fairness Measurement

In the main paper Sec.3, we discussed that there could be considerable error in the fairness measurement, $\hat{p}$, even though the sensitive attribute classifier's accuracy is considerably high. In addition, we further develop on this and discuss two additional factors that could result in a variation of errors. *We remark that in the main manuscript, we report diversity only using VGG-16, as specified by Keswani et al. [5]. Further discussion in Sec. D.2*

**Accuracy of Individual Classes ($\alpha = \{\alpha_0, \alpha_1\}$) Impacts the Degree of Error**. Notice that in some cases even though the sensitive attribute classifier may have a very similar average accuracy, different degrees of errors could exist for the two different classifiers *e.g.* R18 and R34 in Tab. 1. This is because *the fairness measurement error is not only dependent on the average accuracy but on the individual class accuracy i.e.* $\alpha_0$ and $\alpha_1$. More specifically, given that there is a larger error in $\alpha_0$ for R34 and the bias exists in $p_0^* = 0.643$, this results in a compounded effect and hence a larger error of $e_{\mu_{Base}}$=11.98% is observed as compared to R18 $e_{\mu_{Base}}$=6.84%.

**Uniform Inaccuracies at Unbiased Test-Point** ($p_0^* = p_1^* = 0.5$). In our extended experiments in Sec. D.3 for a Pseudo-generator, we discuss that for some sensitive attribute classifiers *e.g.* ResNet-18 for `Gender` and `BlackHair`, the Baseline performs better than CLEAM at the unbiased test-point *i.e.* $p_0^* = 0.5$. This is just due to the `Gender` and `Blackhair` setups having a specific combination of (i) the Pseudo-Generator producing almost perfectly unbiased data with $\boldsymbol{p^*} = [0.5, 0.5]$, (ii) sensitive attribute classifier with almost perfectly uniform inaccuracies $\alpha_0' \approx \alpha_1'$, thereby leading to *uniform misclassification* and hence the false impression of better accuracy by the baseline method, at $\boldsymbol{p^*} = [0.5, 0.5]$ (See Tab. 2 for extracted table) . To further illustrate this, notice how the ResNet-18 trained on `Cat/Dog` did not demonstrate this better performance in the Baseline due to its non-uniform $\boldsymbol{\alpha}$. Nevertheless, we note this situation whereby the Baseline outperform CLEAM is specific to the test-point $p_0^* = 0.5$ and does not impact the overall effectiveness of CLEAM. Furthermore, CLEAM still demonstrates outstanding results with low error for both the PE and IE at $p_0^* = 0.5$.

To further demonstrate these effects, we repeat this same experiment, but with sensitive attributes `Young` and `Attractive` from the CelebA dataset. As seen in Tab. 3, both `Young` or `Attractive` have similar average accuracy, $\alpha_{Avg} = \frac{\alpha_0 + \alpha_1}{2}$ of 0.801 and 0.794 but a different $skew_{\boldsymbol{\alpha}} = |\alpha_0 - \alpha_1|$ of 0.103 and 0.027. As such, we are able to investigate the effects that $skew_{\boldsymbol{\alpha}}$ has on both CLEAM and Baseline. We did not include Diversity in this study, due to its poor performance on harder sensitive attribute, as discussed in Sec. D.2. ***From our results in Tab. 3***, we observe that as the $skew_{\boldsymbol{\alpha}}$ increases from sensitive attribute `Attractive` to `Young`, the error becomes much more significant in the baseline method. The average $e_{\mu_{Base}}$ increases from 12.69% to 17.63%. Furthermore, unlike `Gender` and `Blackhair`, who have relatively negligible skew, `Young` and `Attractive` observes a significantly larger error at $\boldsymbol{p^*} = [0.5, 0.5]$.

Table 1: **Extracted from Tab. 11 for ease of viewing.** Comparing the *point estimates* and *interval estimates* of Baseline [2] and our proposed CLEAM measurement framework in estimating $\boldsymbol{p^*}$ of the GenData datasets sampled from (A) StyleGAN2 [6]. The $p_0^*$ value for each GAN with a certain attribute is determined by manually hand-labeling the generated data. We utilize two different classifiers Resnet-18/34 (R18, R34)[7] with different accuracy $\boldsymbol{\alpha}$ to obtain $\hat{p}$ by classifying samples *w.r.t.* `BlackHair`. For calculating each $\hat{p}$, we utilize $n = 400$ samples and evaluate for a batch size of $s = 30$. We repeat this for 5 experimental runs and report the mean error rate, per Eqn. 1 of the main paper.

| Classifier | $\alpha = \{\alpha_0, \alpha_1\}$ | Avg. $\alpha$ | Point Estimate | | | | | | Interval Estimate | | | | | |
| | | | **Baseline** | | **Diversity** | | **CLEAM (Ours)** | | **Baseline** | | **Diversity** | | **CLEAM (Ours)** | |
| | | | $\mu_{\text{Base}}$ | $e_\mu(\downarrow)$ | $\mu_{\text{Div}}$ | $e_\mu(\downarrow)$ | $\mu_{\text{CLEAM}}$ | $e_\mu(\downarrow)$ | $\rho_{\text{Base}}$ | $e_\rho(\downarrow)$ | $\rho_{\text{Div}}$ | $e_\rho(\downarrow)$ | $\rho_{\text{CLEAM}}$ | $e_\rho(\downarrow)$ |
| **(A) StyleGAN2** | | | | | | | | | | | | | | |
| BlackHair with GT class probability $p_0^*$=0.643 | | | | | | | | | | | | | | |
| R18 | {0.869, 0.885} | 0.88 | 0.599 | 6.84% | — | — | 0.641 | **0.31%** | [0.591, 0.607] | 8.08% | — | — | [0.631, 0.652] | **1.40%** |
| R34 | {0.834, 0.916} | 0.88 | 0.566 | 11.98% | — | — | 0.644 | **0.16%** | [0.561, 0.572] | 12.75% | — | — | [0.637, 0.651] | **1.24%** |

Table 2: **Extracted from Tab. 11 for ease of viewing.** Comparing the *point estimates* and *interval estimate* of Baseline [2], Diversity [5] and proposed CLEAM measurement frameworks in estimating different $p^*$ of a *pseudo-generator*, based on the CelebA [8] and AFHQ [9] dataset. The $\hat{p}$ is computed with a ResNet-18 sensitive attribute classifier and the error rate is reported using Eqn. 1 of the main paper. We repeat this for `Gender`, `BlackHair` and `Cat/Dog` attributes.

| | Point Estimate | | | | | | Interval Estimate | | | | | |
|---|---|---|---|---|---|---|---|---|---|---|---|---|
| GT | Baseline | | Diversity | | CLEAM (Ours) | | Baseline | | Diversity | | CLEAM (Ours) | |
| | $\mu_{\text{Base}}$ | $e_\mu(\downarrow)$ | $\mu_{\text{Div}}$ | $e_\mu(\downarrow)$ | $\mu_{\text{CLEAM}}$ | $e_\mu(\downarrow)$ | $\rho_{\text{Base}}$ | $e_\rho(\downarrow)$ | $\rho_{\text{Div}}$ | $e_\rho(\downarrow)$ | $\rho_{\text{CLEAM}}$ | $e_\rho(\downarrow)$ |
| | $\alpha$=[0.976,0.979], `Gender` (CelebA) | | | | | | | | | | | |
| $p_0^*$=0.5 | 0.501 | **0.20**% | 0.481 | 3.80% | 0.502 | 0.40% | [0.495 , 0.507 ] | **1.40**% | [0.473 , 0.490 ] | 5.40% | [0.497, 0.508] | 1.60% |
| | $\alpha$=[0.881,0.887], `BlackHair` (CelebA) | | | | | | | | | | | |
| $p_0^*$=0.5 | 0.500 | **0.00**% | 0.521 | 4.20% | 0.504 | 0.8% | [ 0.495 , 0.505 ] | **1.00**% | [0.506 , 0.536 ] | 7.20% | [0.497, 0.511] | 2.20% |
| | $\alpha$=[0.953,0.0.990], `Cat/Dog` (AFHQ) | | | | | | | | | | | |
| $p_0^*$=0.5 | 0.486 | 2.80% | 0.469 | 6.20% | 0.505 | **1.00**% | [ 0.480 , 0.493 ] | 4.00% | [ 0.458, 0.480 ] | 8.40% | [ 0.498 , 0.511 ] | **2.20**% |

Table 3: **Duplicate of Tab. 12 for ease of viewing.** Comparing **point estimate** and **interval estimate** of Baseline [2], and proposed CLEAM measurement framework on a pseudo-generator with sensitive attribute {`Young`, `Attractive`}

| | Point Estimate | | | | | | Interval Estimate | | | | | |
|---|---|---|---|---|---|---|---|---|---|---|---|---|
| GT | Baseline | | Diversity | | CLEAM (Ours) | | Baseline | | Diversity | | CLEAM (Ours) | |
| | $\mu_{\text{Base}}$ | $e_\mu(\downarrow)$ | $\mu_{\text{Div}}$ | $e_\mu(\downarrow)$ | $\mu_{\text{CLEAM}}$ | $e_\mu(\downarrow)$ | $\rho_{\text{Base}}$ | $e_\rho(\downarrow)$ | $\rho_{\text{Div}}$ | $e_\rho(\downarrow)$ | $\rho_{\text{CLEAM}}$ | $e_\rho(\downarrow)$ |
| | $\alpha$=[0.749,0.852], `Young` | | | | | | | | | | | |
| $p_0^* = 0.9$ | 0.690 | 23.33% | — | — | 0.905 | **0.56**% | [0.684,0.695] | 24.00% | — | — | [0.890,0.920] | **2.22**% |
| $p_0^* = 0.8$ | 0.630 | 21.25% | — | — | 0.804 | **0.50**% | [0.625,0.635] | 21.88% | — | — | [0.795,0.813] | **1.63**% |
| $p_0^* = 0.7$ | 0.570 | 18.57% | — | — | 0.698 | **0.29**% | [0.565,0.575] | 19.29% | — | — | [0.690,0.706] | **1.43**% |
| $p_0^* = 0.6$ | 0.510 | 15.00% | — | — | 0.595 | **0.83**% | [0.505,0.515] | 15.83% | — | — | [0.590,0.600] | **1.67**% |
| $p_0^* = 0.5$ | 0.450 | 10.0% | — | — | 0.506 | **1.20**% | [0.445,0.455] | 11.00% | — | — | [0.502,0.510] | **2.00**% |
| Avg Error | | 17.63% | | —% | | **0.68**% | | 18.40% | | —% | | **1.79**% |
| | $\alpha$=[0.780,0.807], `Attractive` | | | | | | | | | | | |
| $p_0^* = 0.9$ | 0.730 | 18.89% | — | — | 0.908 | **0.89**% | [0.724,0.736] | 19.56% | — | — | [0.900,0.916] | **1.78**% |
| $p_0^* = 0.8$ | 0.670 | 16.25% | — | — | 0.804 | **0.50**% | [0.665,0.675] | 16.88% | — | — | [0.795,0.813] | **1.63**% |
| $p_0^* = 0.7$ | 0.600 | 14.29% | — | — | 0.696 | **0.57**% | [0.594,0.606] | 15.14% | — | — | [0.690,0.712] | **1.71**% |
| $p_0^* = 0.6$ | 0.540 | 10.00% | — | — | 0.592 | **1.33**% | [0.534,0.546] | 11.00% | — | — | [0.580,0.604] | **3.33**% |
| $p_0^* = 0.5$ | 0.480 | 4.00% | — | — | 0.493 | **1.40**% | [0.475,0.485] | 5.00% | — | — | [0.487,0.499] | **2.60**% |
| Avg Error | | 12.69% | | —% | | **0.94**% | | 13.52% | | —% | | **2.22**% |

# C  Validating Statistical Model for Classifier Output

## C.1  Validation of Sample-Based Estimate vs Model-Based Estimate

As described in the main paper, we utilize the sample-based estimate, $\ddot{\mu}_{\hat{p}_0}$, $\ddot{\sigma}^2_{\hat{p}_0}$ as an approximate for the model-based estimate $\tilde{\mu}_{\hat{p}_0}$, $\tilde{\sigma}^2_{\hat{p}_0}$. As discussed in Sec. A.2, $\ddot{\mu}_{\hat{p}_0}$ allows us to find the maximum likelihood approximate of $p^*$.

**To validate this approximation,** we utilize a ResNet-18 trained on `Gender` and `BlackHair` to compute $\hat{p}$. Then with the samples from the pseudo-generators with different $p^*$ (following Sec. D.3) we computed $\hat{p}$ with a batch-size of $s = 30$ and sample size $n = 400$. Finally, we calculate the sample-based estimates as given in Eqn. 6, 7 of the main paper. As the GT $p^*$ and classifier's accuracy $\alpha$ is known, we also calculate the model-based estimates as given in Eqn. 4, 5 of the main manuscript and compare it against the sample-based estimates.

**Our results** in Tab. 4 shows that both the sample and theoretical means and standard deviations are close approximate to one another. Thus, we can utilise the sample statistics as a close approximation in our proposed method, CLEAM. Additional results for different values of batch-sizes ($s$) and sample-sizes ($n$) are tabulated in Tab. 5, 6 and 7. Notice that a reduction in $s$ and $n$ values contributed to increased errors between the sample-based and model-based estimates. While making $s$ very large ($s = 200$), results in the sample based estimate almost a perfectly approximating the model based estimates.

Table 4: **Comparing sample-based estimates ($\ddot{\mu}_{\hat{p}_0}$, $\ddot{\sigma}_{\hat{p}_0}$) against model-based estimates ($\tilde{\mu}_{\hat{p}_0}$, $\tilde{\sigma}_{\hat{p}_0}$).** The results show that sample-based estimates are close to model-based estimates. Furthermore, note the discrepancy between $p_0^*$ and $\ddot{\mu}_{\hat{p}_0}$, and that between $p_0^*$ and $\tilde{\mu}_{\hat{p}_0}$, highlighting the issue of using $\hat{p}_0$ directly to estimate $p_0^*$ and the need to compensate for the sensitive attribute classifier error as we discussed. We utilize a $s = 30$ and $n = 400$.

| GT | Sampled-based estimates | | Model-based estimates | |
|---|---|---|---|---|
| | $\ddot{\mu}_{\hat{p}_0}$ | $\sqrt{\ddot{\sigma}^2_{\hat{p}_0}}$ | $\tilde{\mu}_{\hat{p}_0}$ | $\sqrt{\tilde{\sigma}^2_{\hat{p}_0}}$ |
| Gender, $\alpha$=[0.976,0.979] | | | | |
| $p_0^* = 0.9$ | 0.881 | 0.0101 | 0.881 | 0.0106 |
| $p_0^* = 0.8$ | 0.781 | 0.0133 | 0.785 | 0.0135 |
| $p_0^* = 0.7$ | 0.692 | 0.0149 | 0.690 | 0.0152 |
| $p_0^* = 0.6$ | 0.590 | 0.0165 | 0.594 | 0.0162 |
| $p_0^* = 0.5$ | 0.503 | 0.0164 | 0.499 | 0.0164 |
| $\alpha$=[0.881,0.887], Black-Hair | | | | |
| $p_0^* = 0.9$ | 0.802 | 0.0130 | 0.804 | 0.0139 |
| $p_0^* = 0.8$ | 0.723 | 0.0151 | 0.727 | 0.0162 |
| $p_0^* = 0.7$ | 0.653 | 0.0169 | 0.650 | 0.0177 |
| $p_0^* = 0.6$ | 0.580 | 0.0180 | 0.574 | 0.0186 |
| $p_0^* = 0.5$ | 0.502 | 0.0180 | 0.497 | 0.0189 |

Table 5: We repeat the same experiment as Tab.4 with $s = 20$ and $n = 400$ samples.

| GT | Sampled-based estimates | | Model-based estimates | |
|---|---|---|---|---|
| | $\ddot{\mu}_{\hat{p}_0}$ | $\sqrt{\ddot{\sigma}^2_{\hat{p}_0}}$ | $\tilde{\mu}_{\hat{p}_0}$ | $\sqrt{\tilde{\sigma}^2_{\hat{p}_0}}$ |
| Gender, $\boldsymbol{\alpha}$=[0.976,0.979] | | | | |
| $p_0^* = 0.9$ | 0.855 | 0.0201 | 0.881 | 0.0106 |
| $p_0^* = 0.8$ | 0.774 | 0.0211 | 0.785 | 0.0135 |
| $p_0^* = 0.7$ | 0.672 | 0.0219 | 0.690 | 0.0152 |
| $p_0^* = 0.6$ | 0.580 | 0.0181 | 0.594 | 0.0162 |
| $p_0^* = 0.5$ | 0.510 | 0.0230 | 0.499 | 0.0164 |
| $\boldsymbol{\alpha}$=[0.881,0.887], Black-Hair | | | | |
| $p_0^* = 0.9$ | 0.768 | 0.180 | 0.804 | 0.0139 |
| $p_0^* = 0.8$ | 0.712 | 0.210 | 0.727 | 0.0162 |
| $p_0^* = 0.7$ | 0.658 | 0.190 | 0.650 | 0.0177 |
| $p_0^* = 0.6$ | 0.554 | 0.230 | 0.574 | 0.0186 |
| $p_0^* = 0.5$ | 0.508 | 0.242 | 0.497 | 0.0189 |

Table 6: We repeat the same experiment as per Tab.4 with $s = 30$ and $n = 200$ samples.

| GT | Sampled-based estimates | | Model-based estimates | |
|---|---|---|---|---|
| | $\ddot{\mu}_{\hat{p}_0}$ | $\sqrt{\ddot{\sigma}^2_{\hat{p}_0}}$ | $\tilde{\mu}_{\hat{p}_0}$ | $\sqrt{\tilde{\sigma}^2_{\hat{p}_0}}$ |
| Gender, $\boldsymbol{\alpha}$=[0.976,0.979] | | | | |
| $p_0^* = 0.9$ | 0.860 | 0.0232 | 0.881 | 0.0149 |
| $p_0^* = 0.8$ | 0.780 | 0.0286 | 0.785 | 0.0191 |
| $p_0^* = 0.7$ | 0.710 | 0.0294 | 0.690 | 0.0215 |
| $p_0^* = 0.6$ | 0.578 | 0.0380 | 0.594 | 0.0228 |
| $p_0^* = 0.5$ | 0.520 | 0.0321 | 0.499 | 0.0233 |
| $\boldsymbol{\alpha}$=[0.881,0.887], Black-Hair | | | | |
| $p_0^* = 0.9$ | 0.742 | 0.0312 | 0.804 | 0.0197 |
| $p_0^* = 0.8$ | 0.740 | 0.0332 | 0.727 | 0.0229 |
| $p_0^* = 0.7$ | 0.610 | 0.0291 | 0.650 | 0.0250 |
| $p_0^* = 0.6$ | 0.582 | 0.350 | 0.574 | 0.0262 |
| $p_0^* = 0.5$ | 0.542 | 0.388 | 0.497 | 0.0267 |

Table 7: We repeat the same experiment as per Tab.4 with $s = 200$ and $n = 400$ samples.

| GT | Sampled-based estimates | | Model-based estimates | |
|---|---|---|---|---|
| | $\ddot{\mu}_{\hat{p}_0}$ | $\sqrt{\ddot{\sigma}^2_{\hat{p}_0}}$ | $\tilde{\mu}_{\hat{p}_0}$ | $\sqrt{\tilde{\sigma}^2_{\hat{p}_0}}$ |
| Gender, $\boldsymbol{\alpha}$=[0.976,0.979] | | | | |
| $p_0^* = 0.9$ | 0.881 | 0.0104 | 0.881 | 0.0106 |
| $p_0^* = 0.8$ | 0.784 | 0.0133 | 0.785 | 0.0135 |
| $p_0^* = 0.7$ | 0.690 | 0.0153 | 0.690 | 0.0152 |
| $p_0^* = 0.6$ | 0.594 | 0.0160 | 0.594 | 0.0162 |
| $p_0^* = 0.5$ | 0.500 | 0.0164 | 0.499 | 0.0164 |
| $\boldsymbol{\alpha}$=[0.881,0.887], Black-Hair | | | | |
| $p_0^* = 0.9$ | 0.804 | 0.0137 | 0.804 | 0.0139 |
| $p_0^* = 0.8$ | 0.726 | 0.0160 | 0.727 | 0.0162 |
| $p_0^* = 0.7$ | 0.650 | 0.0179 | 0.650 | 0.0177 |
| $p_0^* = 0.6$ | 0.573 | 0.0185 | 0.574 | 0.0186 |
| $p_0^* = 0.5$ | 0.498 | 0.0191 | 0.497 | 0.0189 |

## C.2 Goodness-of-Fit Test: $\hat{p}$ from the Real GANs with Our Theoretical Model

In order to make sure that our proposed theoretical model in Eqn. 4 and Eqn. 5 of the main paper, is also a good representation of the $\hat{p}$ distribution when using a generator, we perform a goodness of fit test between the proposed model for the distribution of $\hat{p}$ and sample data generated by a GAN.

Table 8: *Validating goodness-of-fit of the proposed theoretical model against generated samples.* A KS-test [10] is conducted between the samples distribution of $\hat{p}$ - measured from GenData with a ResNet-18, and the theoretical distribution of $\hat{p}$. We utilize $s$=30, $n$=400 with $D_{crit}$=0.24. Since $\eta < D_{crit}$, all of the $\hat{p}$ are statistically similar to the theoretical Gaussian at 95% confidence. This is further observed by the sample-based mean ($\ddot{\mu}$) $\approx$ model-based mean ($\tilde{\mu}$).

| Model Type | Sensitive Attribute | $\eta$ | $\tilde{\mu}$ | $\ddot{\mu}$ |
|---|---|---|---|---|
| StyleGAN2 | Gender | 0.1048 | 0.610 | 0.609 |
| | Blackhair | 0.1065 | 0.601 | 0.601 |
| StyleSwin | Gender | 0.1509 | 0.628 | 0.629 |
| | Blackhair | 0.1079 | 0.619 | 0.614 |

To do this, we first obtain $s = 30$ values of $\hat{p}$ from framework shown in Fig. 1 of the main paper, and use StyleGAN2 [6] and StyleSwin [11] as the generative model. Then using ResNet-18 with known $\alpha$ and GAN's GT $p^*$, as discussed in Sec. 4.1 of the main paper, we form the theoretical model's Gaussian distribution, $\mathcal{N}(\tilde{\mu}_{\hat{p}_0}, \tilde{\sigma}^2_{\hat{p}_0})$.

Now with both our model distribution and the GAN samples, we utilise the Kolmogorov-Smirnov goodness of fit test (K-S test) to determine if the samples distribution is statistically similar to the proposed Gaussian model. We thus propose the following hypothesis test for the samples $\hat{p}^i_j, i \in \{1, \cdots, s\}$:

$\mathbf{H_0}$ : *the samples $\hat{p}^i_j$ belong to the modelled distribution.*

$\mathbf{H_1}$ : *at least one of the samples $\hat{p}^i_j$ does not match the modelled distribution.*

The K-S test then measures a D-statistic ($\eta$) and compares it against a $D_{crit}$ for a given $s$. As we use $s = 30$, and a significance level $\delta = 0.05$ in our setup, we have $D_{crit} = 0.24$. As seen from Tab. 8, all of the measured $\eta$ values are below $D_{crit}$, thus we cannot reject the null hypothesis at a 95% confidence with the K-S test. Therefore, we conclude that the distribution of the obtained samples from the framework (by GANs as generator) are statistically similar to the proposed Gaussian distribution. As a result, we can utilise CLEAM to approximate the $p^*$ range in the presence of a real GAN as the generator.

We further perform a Quantile-Quantile(QQ) analysis to provide a more visual representation. In particular, we plot the Quantile-Quantile(QQ) plot between the $\hat{p}$ samples (produced for the data generated by the GAN) and proposed model. As seen in Fig. 2, the $\hat{p}$ samples from GAN correlate tightly with the standardised line (in red), a line indicating a perfect correlation between theoretical and sample quantiles. This analysis supports our claim that the $\hat{p}$ samples from a real generator (GAN) follow the distribution estimated by the proposed model.

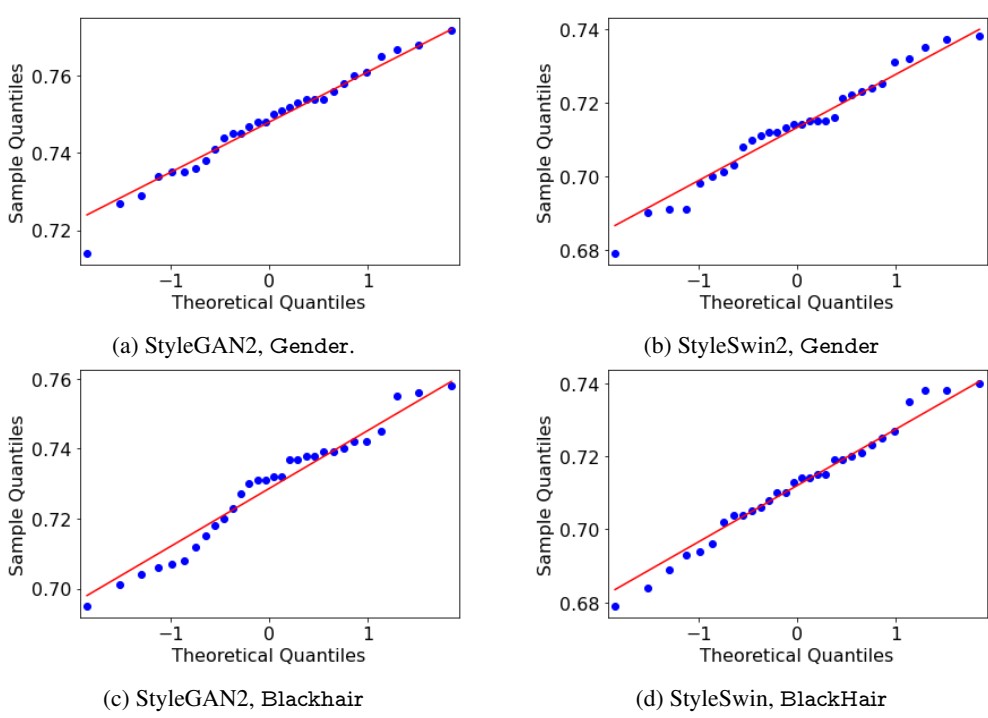

Figure 2: Quartile-Quartile(QQ) plot between $s = 30$ $\hat{\boldsymbol{p}}$ samples calculated for StyleGAN2 [6] and StyleSwin [11] generators and proposed theoretical model for $\hat{\boldsymbol{p}}$

# D Additional Experiments

## D.1 Experimental Results with Standard Deviation

In the main manuscript, we did not include the error bars of our experiments due to space constraints. Hence, in this section, we provide the full tables for Tab. 1 and 2 of the main manuscript with the standard deviation over 5 runs. Note that generally, the standard deviation at each test point is relatively small and hence can be considered as negligible. This is likely due to the large $s$ and $n$ utilized. As a result, we can utilize the mean results (as seen in the main manuscript) to compare CLEAM against Diversity and the Baseline.

Table 9: Comparing the *point estimates* and *interval estimates* of Baseline [2], Diversity [5] and our proposed CLEAM measurement framework in estimating $p^*$ of the GenData datasets sampled from (A) StyleGAN2 [6] and (B) StyleSwin [11]. The $p_0^*$ value for each GAN with a certain sensitive attribute is determined by manually hand-labeling the generated data. We utilize four different sensitive attribute classifier Resnet-18/34 (R18, R34)[7], MobileNetv2 (MN2)[12] and VGG-16 (V16)[13], with different accuracy $\alpha$, to classify attributes Gender and BlackHair, to obtain $\hat{p}$. Each $\hat{p}$ utilizes $n = 400$ samples and is evaluated for a batch size of $s = 30$. We repeat this for 5 experimental runs and report the mean error rate, per Eqn. 1 of the main manuscript.

| | Point Estimate | | | | | | Interval Estimate | | | | | |
| --- | --- | --- | --- | --- | --- | --- | --- | --- | --- | --- | --- | --- |
| | Baseline | | Diversity | | CLEAM (Ours) | | Baseline | | Diversity | | CLEAM (Ours) | |
| | $\mu_{Base}$ | $e_\mu(\downarrow)$ | $\mu_{Div}$ | $e_\mu(\downarrow)$ | $\mu_{CLEAM}$ | $e_\mu(\downarrow)$ | $\rho_{Base}$ | $e_\rho(\downarrow)$ | $\rho_{Div}$ | $e_\rho(\downarrow)$ | $\rho_{CLEAM}$ | $e_\rho(\downarrow)$ |
| (A) StyleGAN2 | | | | | | | | | | | | |
| Gender with GT class probability $p_0^*$=0.642 | | | | | | | | | | | | |
| R18 | 0.610 ± 0.004 | 4.98% | — | — | 0.638 ± 0.006 | **0.62%** | [0.602± 0.004, 0.618± 0.004] | 6.23% | — | — | [0.629 ± 0.006, 0.646± 0.006] | **2.02%** |
| R34 | 0.596± 0.003 | 7.17% | — | — | 0.634± 0.002 | **1.25%** | [0.589± 0.003, 0.599± 0.003] | 8.26% | — | — | [0.628± 0.002, 0.638± 0.002] | **2.18%** |
| MN2 | 0.607 ± 0.003 | 5.45% | — | — | 0.637 ± 0.002 | **0.78%** | [0.602± 0.003, 0.612 ± 0.003] | 6.23% | — | — | [0.632 ± 0.002, 0.643 ± 0.002] | **1.56%** |
| V16 | 0.532 ± 0.007 | 17.13% | 0.550 ± 0.011 | 14.3% | 0.636 ± 0.007 | **0.93%** | [0.526 ± 0.007, 0.538 ± 0.007] | 18.06% | [0.536 ± 0.011 , 0.564 ± 0.011] | 16.51% | [0.628 ± 0.007, 0.644 ± 0.007] | **2.18%** |
| Avg Error | | 8.68% | | 14.30% | | **0.90%** | | 9.70% | | 16.51% | | **1.99%** |
| BlackHair with GT class probability $p_0^*$=0.643 | | | | | | | | | | | | |
| R18 | 0.599 ± 0.006 | 6.84% | — | — | 0.641 ± 0.004 | **0.31%** | [0.591 ± 0.006, 0.607 ± 0.005] | 8.08% | — | — | [0.631 ± 0.004, 0.652 ± 0.003] | **1.40%** |
| R34 | 0.566 ± 0.007 | 11.98% | — | — | 0.644 ± 0.008 | **0.16%** | [0.561 ± 0.007, 0.572 ± 0.006] | 12.75% | — | — | [0.637 ± 0.009, 0.651 ± 0.008] | **1.24%** |
| MN2 | 0.579 ± 0.007 | 9.95% | — | — | 0.639 ± 0.007 | **0.62%** | [0.574 ± 0.008, 0.584 ± 0.008] | 10.73% | — | — | [0.632 ± 0.007, 0.647 ± 0.007] | **1.71%** |
| V16 | 0.603 ± 0.004 | 6.22% | 0.582 ± 0.011 | 9.49% | 0.640 ± 0.005 | **0.47%** | [0.597 ± 0.004, 0.608 ± 0.003] | 7.15% | [0.568 ± 0.010, 0.596 ± 0.011] | 11.66% | [0.632 ± 0.004, 0.648 ± 0.005] | **1.71%** |
| Avg Error | | 8.75% | | 9.49% | | **0.39%** | | 9.68% | | 11.66% | | **1.52%** |
| (B) StyleSwin | | | | | | | | | | | | |
| Gender with GT class probability $p_0^*$=0.656 | | | | | | | | | | | | |
| R18 | 0.620 ± 0.005 | 5.49% | — | — | 0.648 ± 0.004 | **1.22%** | [0.612 ± 0.004,0.629 ± 0.005] | 6.70% | — | — | [0.639 ± 0.005,0.658 ± 0.005] | **2.59%** |
| R34 | 0.610 ± 0.002 | 7.01% | — | — | 0.649 ± 0.005 | **1.07%** | [0.605 ± 0.003,0.615 ± 0.002] | 7.77% | — | — | [0.643 ± 0.006,0.654 ± 0.002] | **1.98%** |
| MN2 | 0.623 ± 0.008 | 5.03% | — | — | 0.655 ± 0.005 | **0.15%** | [0.618 ± 0.007,0.629± 0.007] | 5.79% | — | — | [0.649 ± 0.006,0.661 ± 0.006] | **1.07%** |
| V16 | 0.555 ± 0.004 | 15.39% | 0.562 ± 0.015 | 14.33% | 0.668 ± 0.006 | **1.83%** | [0.549 ± 0.004,0.560 ± 0.004] | 16.31% | [0.548 ± 0.014,0.576 ± 0.014] | 16.46% | [0.660 ± 0.007,0.675 ± 0.007] | **2.90%** |
| Avg Error | | 8.23% | | 14.33% | | **1.07%** | | 9.14% | | 16.46% | | **2.14%** |
| BlackHair with GT class probability $p_0^*$=0.668 | | | | | | | | | | | | |
| R18 | 0.612 ± 0.005 | 8.38% | — | — | 0.659 ± 0.006 | **1.35%** | [0.605 ± 0.005,0.620 ± 0.006] | 9.43% | — | — | [0.649 ± 0.004,0.670 ± 0.004] | **2.84%** |
| R34 | 0.581 ± 0.006 | 13.02% | — | — | 0.662 ± 0.006 | **0.90%** | [0.576 ± 0.005,0.586 ± 0.006] | 13.77% | — | — | [0.656 ± 0.005,0.669 ± 0.005] | **1.80%** |
| MN2 | 0.596 ± 0.006 | 10.78% | — | — | 0.659 ± 0.005 | **1.35%** | [0.591 ± 0.006,0.600 ± 0.007] | 11.50% | — | — | [0.652 ± 0.005,0.666± 0.005] | **2.40%** |
| V16 | 0.625 ± 0.006 | 6.44% | 0.608 ± 0.014 | 8.98% | 0.677 ± 0.005 | **1.35%** | [0.620 ± 0.005,0.630 ± 0.006] | 7.19% | [0.590 ± 0.012,0.626 ± 0.013] | 11.68% | [0.670 ± 0.005,0.684 ± 0.006] | **2.40%** |
| Avg Error | | 9.66% | | 8.98% | | **1.24%** | | 10.47% | | 11.68% | | **2.36%** |

Table 10: Comparing the *point estimates* and *interval estimates* of Baseline and CLEAM in estimating the $p^*$ of the Stable Diffusion Model [14] with the GenData-SDM dataset. We use prompt input starting with "A photo of with the face of" and ending with synonymous (Gender neutral) prompts. We utilized CLIP as the sensitive attribute classifier for Gender, to obtain $\hat{p}$.

| | | Point Estimate | | | | Interval Estimate | | | |
| --- | --- | --- | --- | --- | --- | --- | --- | --- | --- |
| Prompt | GT | Baseline | | CLEAM (Ours) | | Baseline | | CLEAM (Ours) | |
| | | $\mu_{Base}$ | $e_\mu(\downarrow)$ | $\mu_{CLEAM}$ | $e_\mu(\downarrow)$ | $\rho_{Base}$ | $e_\rho(\downarrow)$ | $\rho_{CLEAM}$ | $e_\rho(\downarrow)$ |
| $\alpha$=[0.998,0.975], Avg. $\alpha$=0.987, CLIP−Gender | | | | | | | | | |
| "A photo with the face of an individual" | 0.186 | 0.203 ± 0.011 | 9.14% | 0.187 ± 0.11 | **0.05%** | [ 0.198 ± 0.10 , 0.208 ± 0.10 ] | 11.83% | [ 0.182 ± 0.10 , 0.192 ± 0.10 ] | **3.23%** |
| "A photo with the face of a human being" | 0.262 | 0.277 ± 0.10 | 5.73% | 0.263 ± 0.10 | **0.38%** | [ 0.270 ± 0.10 , 0.285 ± 0.10 ] | 8.78% | [ 0.255 ± 0.10 , 0.271 ± 0.10 ] | **3.44%** |
| "A photo with the face of one person" | 0.226 | 0.241 ± 0.009 | 6.63% | 0.230 ± 0.08 | **1.77%** | [ 0.232 ± 0.10 , 0.251 ± 0.10 ] | 11.06% | [ 0.220 ± 0.09 , 0.239 ± 0.09 ] | **5.75%** |
| "A photo with the face of a person" | 0.548 | 0.556 ± 0.12 | 1.49% | 0.548 ± 0.11 | **0.00%** | [ 0.545 ± 0.11 , 0.566 ± 0.11 ] | 3.28% | [ 0.537 ± 0.11 , 0.558 ± 0.11 ] | **2.01%** |
| Average Error | | | 5.75% | | **0.44%** | | 8.74% | | **3.61%** |

## D.2   Experimental Setup for Diversity[5]

In this section, we describe our experimental setup for Diversity [5], as utilized in the main paper. Recall that as discussed by Kewsani *et al.* [5] a VGG-16 [13] model pre-trained on ImageNet [15] is utilized as a feature extractor. Then, this feature extractor is applied to both the unknown (generator's data) and the controlled dataset. Finally, the unknown sample's features are compared against the controlled one's via a similarity algorithm to compute diversity, $\delta$.

From our results in Fig. 3a (LHS) based on the pseudo-generator's setup (discussed in more details in Sec. D.3), we recognize that the original implementation with VGG-16 trained on ImageNet works well on the `Gender` sensitive attribute. This is seen by the close approximation made by the proxy diversity score when compared against the GT diversity score evaluated with Eqn. 5, as per [5].

$$GT\ Diversity = p_0^* - p_1^* \tag{5}$$

However, when evaluated on the harder `BlackHair` sensitive attribute, our results in Fig. 3a (RHS) observed significant error between the GT Diversity scores and the proxy Diversity scores. This error was especially prevalent in the larger biases *e.g.* $p_0^* = 0.9$. We theorized that this was due to the differences between the domains of the feature extractor and the generated/controlled images *i.e.* ImageNet versus CelebA/CelebA-HQ.

To verify this, we fine-tune the VGG-16 model on the CelebA dataset with the respective sensitive attribute. Then we removed the last fully connected layer of the classifier model, and utilise the 4096 feature vector for the diversity measurement, as per [5]. Our results in Fig. 3b demonstrate significant improvement on both `Gender` and `BlackHair`, based on the new improved VGG-16 model implementation. This thereby verifies our intuition that there exists a mismatch of domains in the VGG-16 pretrained on ImageNet when utilized with CelebA samples.

However, upon further experimentation, we recognize certain limitations still exist in the Diversity measure when used on more ambiguous and harder sensitive attribute *e.g.* `Young` and `Attractive`. Similar to before, we fine-tuned the sensitive attribute classifier (feature extractor) which achieved accuracies of $78.44\%$ and $84.41\%$ for `Young` and `Attractive`, respectively. However even with this re-implementation, the diversity persistent to perform poorly, as seen in Fig. 4.

Regardless, given the improvement seen on the `BlackHair` sensitive attribute, we utilized our improved VGG-16 feature extractor in the main paper, in place of the pre-trained VGG-16 (ImageNet).

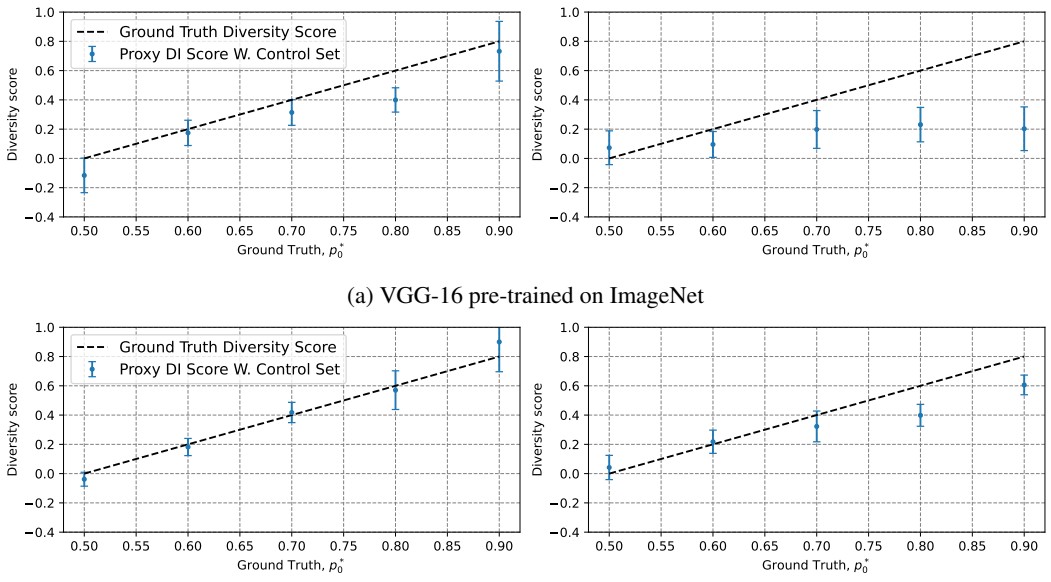

(a) VGG-16 pre-trained on ImageNet

(b) VGG-16 pre-trained on ImageNet then fine-tuned on CelebA

Figure 3: **Improvement in Diversity by fine-tuning the VGG-16, as a feature extractor**: **(a)** Diversity implementation by [5] with VGG-16 pre-trained on ImageNet as the feature extractor testing on the pseudo-generator's with $p_0^* = \{0.9, 0.8, 0.7, 0.6, 0.5\}$ for sensitive attribute Gender(Left) and BlackHair(Right). **(b)** We re-implemented VGG-16 and furter fine-tune it with CelebA as the feature extractor. We observed improvement in predicting the GT $p^*$

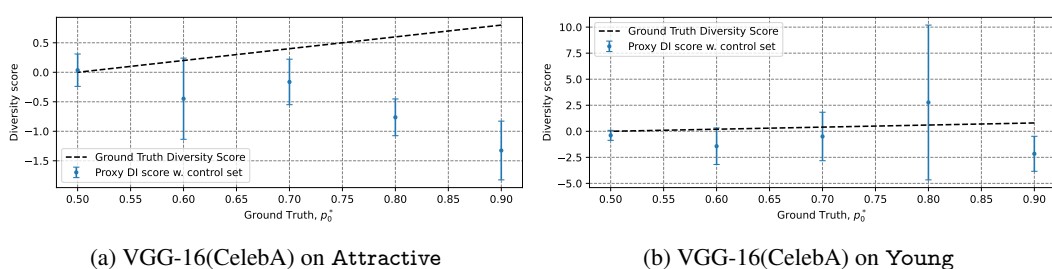

(a) VGG-16(CelebA) on Attractive

(b) VGG-16(CelebA) on Young

Figure 4: **Limitations Of Diversity algorithm.** Our implementation of VGG-16 fine-tuned on CelebA *w.r.t.* sensitive attribute Attractive and Young. VGG-16 Classifier achieved an accuracy of $78.44\%$ and $84.1\%$ for sensitive attribute Attractive and Young. However, the same VGG-16 performs poorly on the diversity metric, demonstrating the limitations of the diversity framework.

## D.3 Measuring Varying Degrees of Bias

**CLEAM for Measuring Varying Degrees of Bias.** In previous experiments, we show the performance of different methods in measuring the fairness of generators and evaluating bias mitigation techniques. Another interesting analysis would be to see how these methods fare with different bias, *i.e.* different $p^*$ values. A challenge of this analysis is that we cannot control the training dynamics of either the GANs nor the Stable Diffusion Model to obtain an exact value of $p^*$. Thus, we introduce a new setup and use a *pseudo-generator* instead of real GANs.

In this setup, we utilize the CelebA [8] and the AFHQ [9] dataset to construct different modified datasets that follow different values of $p^*$ *w.r.t.* the sensitive attribute *e.g.* BlackHair attribute, when $p^* = \{0.9, 0.1\}$, the modified dataset contains 4880 BlackHair and 542 Non-BlackHair samples. Then, a pseudo-generator with bias $p^*$ works by random sampling from the corresponding datasets. Note that the samples in the modified dataset are unseen to the sensitive attribute classifier. For our experiment, we use different GT values, $p^* = \{p_0^*, p_1^*\}$, where $p_0^* \in \{0.9, 0.8, 0.7, 0.6, 0.5\}$, and $p_1^* = 1 - p_0^*$. For a pseudo-generator, to calculate each value of $\hat{p}$, a batch of $n$ samples is randomly drawn from the corresponding dataset and fed into the $C_\mathbf{u}$ for classification. We utilize a ResNet-18 to evaluate our pseudo-generator. **The results in Tab. 11** for $p_0^*$ demonstrate that CLEAM is effective for different degrees of bias, reducing the average error ($e_\mu$) of the Baseline from 1.43%→0.27% and 6.23%→0.49% for Gender and BlackHair on celebA respectively, and 3.52%→0.75% for Cat/Dog on AFHQ. Additionally, note how measurement error in Baseline and Diversity increases by increasing the data bias, while CLEAM remains consistently low. See Sec. D.4 and D.5 for analysis with more attributes and classifiers.

Table 11: Comparing the *point estimates* and *interval estimate* of Baseline [2], Diversity [5] and CLEAM in estimating different $p^*$ of a *pseudo-generator*, based on CelebA [8] and AFHQ [9], for sensitive attribute Gender, BlackHair and Cat/Dog. The $\hat{p}$ is computed with a ResNet-18 and the error rate is reported per Eqn.1 of the main manuscript

| GT | Point Estimate | | | | | | Interval Estimate | | | | | |
|---|---|---|---|---|---|---|---|---|---|---|---|---|
| | Baseline | | Diversity | | CLEAM (Ours) | | Baseline | | Diversity | | CLEAM (Ours) | |
| | $\mu_{\text{Base}}$ | $e_\mu(\downarrow)$ | $\mu_{\text{Div}}$ | $e_\mu(\downarrow)$ | $\mu_{\text{CLEAM}}$ | $e_\mu(\downarrow)$ | $\rho_{\text{Base}}$ | $e_\rho(\downarrow)$ | $\rho_{\text{Div}}$ | $e_\rho(\downarrow)$ | $\rho_{\text{CLEAM}}$ | $e_\rho(\downarrow)$ |
| $\alpha$=[0.976,0.979], Gender (CelebA) | | | | | | | | | | | | |
| $p_0^*$=0.9 | 0.880 | 2.22% | 0.950 | 5.55% | 0.899 | **0.11**% | [0.876 , 0.884 ] | 2.67% | [0.913 , 0.986 ] | 9.56% | [0.895, 0.904] | **0.56**% |
| $p_0^*$=0.8 | 0.783 | 2.10% | 0.785 | 1.88% | 0.798 | **0.25**% | [ 0.778 , 0.788 ] | 2.75% | [0.762 , 0.809 ] | 4.75% | [0.794,0.803] | **0.75**% |
| $p_0^*$=0.7 | 0.691 | 1.29% | 0.709 | 1.29% | 0.701 | **0.14**% | [ 0.687 , 0.695 ] | 1.86% | [0.696 , 0.722 ] | 3.14% | [0.697, 0.707] | **0.10**% |
| $p_0^*$=0.6 | 0.592 | 1.33% | 0.591 | 1.50% | 0.597 | **0.50**% | [0.586 , 0.598 ] | 2.33% | [0.581 , 0.612 ] | 3.17% | [0.591,0.603] | **1.50**% |
| $p_0^*$=0.5 | 0.501 | **0.20**% | 0.481 | 3.80% | 0.502 | 0.40% | [0.495 , 0.507 ] | **1.40**% | [0.473 , 0.490 ] | 5.40% | [0.497, 0.508] | 1.60% |
| Average Error: | | 1.43% | | 2.80% | | **0.27**% | | 2.20% | | 5.20% | | **0.90**% |
| $\alpha$=[0.881,0.887], BlackHair (CelebA) | | | | | | | | | | | | |
| $p_0^*$=0.9 | 0.803 | 10.77% | 0.803 | 10.77% | 0.899 | **0.11**% | [ 0.800 , 0.806 ] | 11.11% | [0.791, 0.815] | 12.11% | [0.893, 0.905] | **0.78**% |
| $p_0^*$=0.8 | 0.723 | 9.63% | 0.699 | 12.63% | 0.796 | **0.50**% | [0.719 , 0.727 ] | 10.13% | [0.686 , 0.713 ] | 14.25% | [0.790, 0.803] | **1.25**% |
| $p_0^*$=0.7 | 0.654 | 6.57% | 0.661 | 5.57% | 0.705 | **0.71**% | [ 0.648 , 0.660 ] | 7.43% | [ 0.643 , 0.68 ] | 8.14% | [0.698, 0.712] | **1.71**% |
| $p_0^*$=0.6 | 0.575 | 4.17% | 0.609 | 1.50% | 0.602 | **0.33**% | [0.564 , 0.586 ] | 6.00% | [0.604 , 0.614 ] | 2.30% | [0.599, 0.606] | **1.00**% |
| $p_0^*$=0.5 | 0.500 | **0.00**% | 0.521 | 4.20% | 0.504 | 0.8% | [0.495 , 0.505 ] | **1.00**% | [0.506 , 0.536 ] | 7.20% | [0.497, 0.511] | 2.20% |
| Average Error: | | 6.23% | | 6.93% | | **0.49**% | | 7.13% | | 8.80% | | **1.39**% |
| $\alpha$=[0.953,0.0.990], Cat/Dog (AFHQ) | | | | | | | | | | | | |
| $p_0^*$=0.9 | 0.862 | 4.44% | 0.855 | 5.00% | 0.903 | **0.33**% | [ 0.859 , 0.865 ] | 4.56% | [ 0.844 , 0.866 ] | 6.22% | [ 0.900 , 0.907 ] | **0.78**% |
| $p_0^*$=0.8 | 0.766 | 4.25% | 0.774 | 3.25% | 0.802 | **0.25**% | [ 0.762 , 0.771 ] | 4.75% | [ 0.765 , 0.784 ] | 4.38% | [ 0.797, 0.807 ] | **0.88**% |
| $p_0^*$=0.7 | 0.677 | 3.29% | 0.670 | 4.29% | 0.707 | **1.00**% | [ 0.672 , 0.682 ] | 4.00% | [ 0.655 , 0.686 ] | 6.43% | [ 0.701 , 0.712 ] | **1.71**% |
| $p_0^*$=0.6 | 0.583 | 2.83% | 0.551 | 8.17% | 0.607 | **1.17**% | [ 0.578 , 0.588 ] | 3.67% | [ 0.540 , 0.562 ] | 10.00% | [ 0.602 , 0.613 ] | **2.17**% |
| $p_0^*$=0.5 | 0.486 | 2.80% | 0.469 | 6.20% | 0.505 | **1.00**% | [ 0.480 , 0.493 ] | 4.00% | [ 0.458 , 0.480 ] | 8.40% | [ 0.498 , 0.511 ] | **2.20**% |
| Average Error: | | 3.52% | | 5.38% | | **0.75**% | | 4.20% | | 7.09% | | **1.55**% |

## D.4 Measuring Varying Degrees of Bias with Additional Sensitive Attributes

In Sec. D.3, we demonstrate CLEAM's ability to improve accuracy in approximating $p^*$ for the sensitive attributes Gender and BlackHair. In this section, we extend the experiment on CelebA dataset but with harder (lower $\alpha$) sensitive attributes *i.e.* Young, and Attractive. We did not include Diversity in this study, due to its poor performance on harder sensitive attribute, as discussed in D.2.

**From our results in Tab. 12**, both Young and Attractive classifiers have relatively large errors ($e_{\mu_{Base}}$) in the baseline *i.e.* on average 17.63% and 12.69%, respectively. Then utilizing CLEAM, even with the harder sensitive attributes, we are able to significantly reduce these errors to 0.68% and 0.94%. See Sec. B for more details regarding the effect that the different degrees of inaccuracies in $\alpha$ have on the Baseline error.

Table 12: Comparing **point estimate** and **interval estimate** of Baseline [2], and proposed CLEAM measurement framework on a pseudo-generator with sensitive attribute {Young, Attractive}

| GT | Point Estimate | | | | | | Interval Estimate | | | | | |
|---|---|---|---|---|---|---|---|---|---|---|---|---|
| | Baseline | | Diversity | | CLEAM (Ours) | | Baseline | | Diversity | | CLEAM (Ours) | |
| | $\mu_{\text{Base}}$ | $e_\mu(\downarrow)$ | $\mu_{\text{Div}}$ | $e_\mu(\downarrow)$ | $\mu_{\text{CLEAM}}$ | $e_\mu(\downarrow)$ | $\rho_{\text{Base}}$ | $e_\rho(\downarrow)$ | $\rho_{\text{Div}}$ | $e_\rho(\downarrow)$ | $\rho_{\text{CLEAM}}$ | $e_\rho(\downarrow)$ |
| $\alpha$=[0.749,0.852], Young | | | | | | | | | | | | |
| $p_0^*=0.9$ 0.690 | 23.33% | — | — | 0.905 | **0.56**% | [0.684,0.695] | 24.00% | — | — | [0.890,0.920] | **2.22**% | |
| $p_0^*=0.8$ 0.630 | 21.25% | — | — | 0.804 | **0.50**% | [0.625,0.635] | 21.88% | — | — | [0.795,0.813] | **1.63**% | |
| $p_0^*=0.7$ 0.570 | 18.57% | — | — | 0.698 | **0.29**% | [0.565,0.575] | 19.29% | — | — | [0.690,0.706] | **1.43**% | |
| $p_0^*=0.6$ 0.510 | 15.00% | — | — | 0.595 | **0.83**% | [0.505,0.515] | 15.83% | — | — | [0.590,0.600] | **1.67**% | |
| $p_0^*=0.5$ 0.450 | 10.0% | — | — | 0.506 | **1.20**% | [0.445,0.455] | 11.00% | — | — | [0.502,0.510] | **2.00**% | |
| Avg Error | 17.63% | | —% | | | **0.68**% | 18.40% | | —% | | | **1.79**% |
| $\alpha$=[0.780,0.807], Attractive | | | | | | | | | | | | |
| $p_0^*=0.9$ 0.730 | 18.89% | — | — | 0.908 | **0.89**% | [0.724,0.736] | 19.56% | — | — | [0.900,0.916] | **1.78**% | |
| $p_0^*=0.8$ 0.670 | 16.25% | — | — | 0.804 | **0.50**% | [0.665,0.675] | 16.88% | — | — | [0.795,0.813] | **1.63**% | |
| $p_0^*=0.7$ 0.600 | 14.29% | — | — | 0.696 | **0.57**% | [0.594,0.606] | 15.14% | — | — | [0.690,0.712] | **1.71**% | |
| $p_0^*=0.6$ 0.540 | 10.00% | — | — | 0.592 | **1.33**% | [0.534,0.546] | 11.00% | — | — | [0.580,0.604] | **3.33**% | |
| $p_0^*=0.5$ 0.480 | 4.00% | — | — | 0.493 | **1.40**% | [0.475,0.485] | 5.00% | — | — | [0.487,0.499] | **2.60**% | |
| Avg Error | 12.69% | | —% | | | **0.94**% | 13.52% | | —% | | | **2.22**% |

## D.5 Measuring Varying Degrees of Bias with Additional Sensitive Attribute Classifier

In this section, we validate CLEAM's versatility with different sensitive attribute classifier architectures. In our setup, we utilise MobileNetV2 [12] as in [16]. Then similar to Sec. D.3, we utilize a pseudo-generator with known GT $p^*$ for `Gender` and `BlackHair` sensitive attribute from the CelebA [8] dataset, and `Cat/Dog` from the AFHQ [9] dataset, to evaluate CLEAM's effectiveness at determining bias.

As seen in our results in Tab.13, MobileNetV2 achieved reasonably high average accuracy $\in$[0.889,0.983]. Then, when evaluating $p^*$ of the pseudo-generator we observed similar behavior to ResNet-18 discussed in sec. D.3. In particular, we observed a significantly large $e_{\mu_{Base}}$ (for the baseline) of $1.42\%$, $9.74\%$ and $2.81\%$ for the `Gender`, `BlackHair` and `Cat/Dog` sensitive attribute, respectively. Whereas, CLEAM reported an $e_{\mu_{CLEAM}}$ of $0.13\%$, $0.7\%$ and $0.62\%$, respectively. The same trend can be observed in the IE. We thus demonstrate CLEAM's versatility and ability to be deployed as a post-processing method (without retraining), on models of varying architecture.

Table 13: Comparing the *point estimates* and *interval estimate* of Baseline [2], Diversity [5] and proposed CLEAM measurement frameworks in estimating different $p^*$ of a *pseudo-generator*, based on the CelebA [8] and AFHQ [9] dataset. The $\hat{p}$ is computed with a MobileNetV2[12] classifier and the error rate is reported using Eqn. 1 of the main manuscript. We repeat this on `Gender` (CelebA), `BlackHair` (CelebA) and `Cat/Dog`(AFHQ) sensitive attribute.

| | Point Estimate | | | | | | Interval Estimate | | | | | |
|---|---|---|---|---|---|---|---|---|---|---|---|---|
| GT | Baseline | | Diversity | | CLEAM (Ours) | | Baseline | | Diversity | | CLEAM (Ours) | |
| | $\mu_{\text{Base}}$ | $e_\mu(\downarrow)$ | $\mu_{\text{Div}}$ | $e_\mu(\downarrow)$ | $\mu_{\text{CLEAM}}$ | $e_\mu(\downarrow)$ | $\rho_{\text{Base}}$ | $e_\rho(\downarrow)$ | $\rho_{\text{Div}}$ | $e_\rho(\downarrow)$ | $\rho_{\text{CLEAM}}$ | $e_\rho(\downarrow)$ |
| $\alpha$=[0.980,0.986], `Gender` (CelebA) | | | | | | | | | | | | |
| $p_0^*=0.9$ | 0.882 | 2.00% | 0.950 | 5.55% | 0.899 | **0.11**% | [0.879,0.885] | 2.33% | [0.913,0.986] | 9.56% | [0.895,0.902] | **0.56**% |
| $p_0^*=0.8$ | 0.786 | 1.75% | 0.785 | 1.88% | 0.800 | **0.00**% | [0.782,0.790] | 2.25% | [0.762,0.809] | 4.75% | [0.794,0.804] | **0.75**% |
| $p_0^*=0.7$ | 0.689 | 1.57% | 0.709 | 1.30% | 0.699 | **0.14**% | [0.685,0.693] | 2.14% | [0.696,0.722] | 3.14% | [0.694,0.704] | **0.86**% |
| $p_0^*=0.6$ | 0.593 | 1.17% | 0.591 | 1.50% | 0.600 | **0.00**% | [0.585,0.597] | 2.50% | [0.581,0.612] | 3.17% | [594,0.605] | **1.00**% |
| $p_0^*=0.5$ | 0.497 | 0.60% | 0.481 | 3.80% | 0.502 | **0.40**% | [0.491,0.502] | 1.80% | [0.473,0.490] | 5.40% | [495,0.507] | **1.40**% |
| Avg Error | 1.42% | | 2.81% | | | **0.13**% | | 2.20% | | 5.20% | | **0.91**% |
| $\alpha$=[0.861,0.916], `BlackHair` (CelebA) | | | | | | | | | | | | |
| $p_0^*=0.9$ | 0.782 | 13.11% | 0.803 | 10.78% | 0.899 | **0.11**% | [0.777,0.787] | 13.67% | [0.791,0.815] | 9.44% | [0.893,0.900] | **0.78**% |
| $p_0^*=0.8$ | 0.705 | 11.88% | 0.699 | 12.63% | 0.800 | **0.00**% | [0.699,0.710] | 12.63% | [0.686,0.713] | 14.25% | [0.793,0.807] | **0.88**% |
| $p_0^*=0.7$ | 0.623 | 11.00% | 0.661 | 5.56% | 0.700 | **0.00**% | [0.618,0.628] | 11.71% | [0.643,0.68] | 8.14% | [0.694,0.706] | **0.86**% |
| $p_0^*=0.6$ | 0.550 | 8.33% | 0.609 | 1.50% | 0.600 | **0.00**% | [0.544,0.556] | 9.33% | [0.604,0.614] | 2.33% | [0.593,0.608] | **1.17**% |
| $p_0^*=0.5$ | 0.478 | 4.40% | 0.521 | 4.20% | 0.506 | **1.20**% | [0.472,0.484] | 5.60% | [0.506,0.536] | 7.20% | [0.498,0.514] | **2.80**% |
| Avg Error | 9.74% | | 6.93% | | | **0.70**% | | 10.59% | | 8.27% | | **1.30**% |
| $\alpha$=[0.964,0.897], `Cat/Dog` (AFHQ) | | | | | | | | | | | | |
| $p_0^*=0.9$ | 0.875 | 2.77% | 0.880 | 3.26% | 0.897 | **0.34**% | [0.872,0.878] | 3.07% | [0.871,0.890] | 3.25% | [0.894,0.900] | **0.68**% |
| $p_0^*=0.8$ | 0.784 | 2.00% | 0.770 | 3.75% | 0.791 | **1.11**% | [0.780,0.788] | 2.53% | [0.759,0.781] | 5.12% | [0.786,0.796] | **0.42**% |
| $p_0^*=0.7$ | 0.704 | 0.62% | 0.692 | 1.08% | 0.698 | **0.20**% | [0.700,0.708] | 1.19% | [0.684,0.709] | 2.40% | [0.694,0.703] | **0.86**% |
| $p_0^*=0.6$ | 0.617 | 2.78% | 0.611 | 1.83% | 0.597 | **0.54**% | [0.611,0.622] | 2.78% | [0.602,0.620] | 3.42% | [0.591,0.603] | **1.58**% |
| $p_0^*=0.5$ | 0.529 | 5.87% | 0.536 | 7.20% | 0.495 | **0.93**% | [0.523,0.536] | 7.17% | [0.524,0.548] | 9.68% | [0.488,0.503] | **2.44**% |
| Avg Error | 2.81% | | 3.42% | | | **0.62**% | | 3.35% | Avg Error | 4.77% | | **1.20**% |

**D.6 Measuring SOTA GANs and Diffusion Models with Additional Classifier**

In this section, we further explore the utilization of CLIP as a sensitive attribute classifier; more details on CLIP in Sec. E. Here, we follow the setup in Sec. 5.1 of our main manuscript to measure the bias in GenData-StyleGAN2 and GenData-StyleSwin *w.r.t.* Gender. Additionally, we evaluate a publically available pre-trained Latent Diffusion Model (LDM) [17] on FFHQ [6], where we acquire the GT $p^*$ *w.r.t.* Gender with the same procedure as GenData.

Our results in Tab. 14 and 15 shows that the Baseline is able to achieve reasonable accuracy in estimating the GT $p^*$. This is because CLIP's accuracy is very high ($\alpha_0$=0.998) on the bias class ($p_0^*$) for both StyleGAN2, StyleSwin and LDM resulting in less mis-classification from occurring. Regardless, CLEAM is still able to further improve on the already very accurate baseline, further reducing the error from $e_{\mu_{Base}} \geq 0.91\%$, on StyleGAN2, StyleSwin and LDM to $e_{\mu_{CLEAM}} \leq 0.47\%$. A similar trend can be observed in the IE, where it is able to bound the GT $p_0^*$.

Table 14: Comparing the *point estimates* and *interval estimates* of Baseline [2] our CLEAM in estimating $p^*$ of StyleGAN2 [6] and StyleSwin [11] with the GenData datasets. We utilize SA classifier CLIP to classify sensitive attribute Gender. The $p_0^*$ value of each GAN *w.r.t.* SA is determined by manually hand-labeling the generated data. We repeat this for 5 experimental runs and report the mean error rate, per Eqn. 1 of the main manuscript.

| $\alpha = \{\alpha_0, \alpha_1\}$ | Avg. $\alpha$ | Point Estimate | | | | | | Interval Estimate | | | | |
|---|---|---|---|---|---|---|---|---|---|---|---|---|
| | | Baseline | | Diversity | | CLEAM (Ours) | | Baseline | | Diversity | | CLEAM (Ours) |
| | | $\mu_{\text{Base}}$ | $e_\mu(\downarrow)$ | $\mu_{\text{Div}}$ | $e_\mu(\downarrow)$ | $\mu_{\text{CLEAM}}$ | $e_\mu(\downarrow)$ | $\rho_{\text{Base}}$ | $e_\rho(\downarrow)$ | $\rho_{\text{Div}}$ | $e_\rho(\downarrow)$ | $\rho_{\text{CLEAM}}$ | $e_\rho(\downarrow)$ |
| **(A) StyleGAN2** | | | | | | | | | | | | |
| Gender with GT class probability $p_0^*$=0.642 | | | | | | | | | | | | |
| CLIP {0.998, 0.975} | 0.987 | 0.653 | 1.71% | — | — | 0.645 | **0.47%** | [0.649, 0.657] | 2.34% | — | — | [0.641, 0.649] | **1.09%** |
| **(B) StyleSwin** | | | | | | | | | | | | |
| Gender with GT class probability $p_0^*$=0.656 | | | | | | | | | | | | |
| CLIP {0.998, 0.975} | 0.987 | 0.666 | 0.91% | — | — | 0.658 | **0.30%** | [0.663,0.669] | 1.98% | — | — | [0.655,0.662] | **0.91%** |

Table 15: Comparing the *point estimates* and *interval estimates* of Baseline [2] our CLEAM in estimating $p^*$ of a pretrained Latent Diffusion Model[17] on the FFHQ dataset. We repeat this for 5 experimental runs and report the mean error rate, per Eqn. 1 of the main manuscript.

| $\alpha = \{\alpha_0, \alpha_1\}$ | Avg. $\alpha$ | Point Estimate | | | | | | Interval Estimate | | | | |
|---|---|---|---|---|---|---|---|---|---|---|---|---|
| | | Baseline | | Diversity | | CLEAM (Ours) | | Baseline | | Diversity | | CLEAM (Ours) |
| | | $\mu_{\text{Base}}$ | $e_\mu(\downarrow)$ | $\mu_{\text{Div}}$ | $e_\mu(\downarrow)$ | $\mu_{\text{CLEAM}}$ | $e_\mu(\downarrow)$ | $\rho_{\text{Base}}$ | $e_\rho(\downarrow)$ | $\rho_{\text{Div}}$ | $e_\rho(\downarrow)$ | $\rho_{\text{CLEAM}}$ | $e_\rho(\downarrow)$ |
| **Latent Diffusion Model** | | | | | | | | | | | | |
| Gender with GT class probability $p_0^*$=0.570 | | | | | | | | | | | | |
| CLIP {0.998, 0.975} | 0.987 | 0.585 | 2.63% | — | — | 0.571 | **0.18%** | [0.578, 0.593] | 4.04% | — | — | [0.564, 0.579] | **1.58%** |

## D.7 Comparing Classifiers Accuracy on Validation Dataset vs Generated Dataset

In our proposed CLEAM, we use $\alpha$ pre-measured on the validation dataset, denoted by $\alpha_{val}$. In this section, we show that $\alpha_{val}$ is a good approximate of the $\alpha$ when measured on the generated data, denoted by $\alpha_{gen}$. Note that $\alpha_{gen}$ is not available in practice, therefore $\alpha_{val}$ is used as approximation during CLEAM measurement. We further remark that our purpose of this experiment is only done to validate $\alpha_{val}$ as a good approximation for $\alpha_{gen}$ and is not necessary in the actual deployment of CLEAM.

**Comparing $\alpha_{val}$ vs $\alpha_{gen}$ on GANs.** To validate that, we utilize our newly introduced generated dataset, with known labels and measure the $\alpha_{gen}$ for both `Gender` and `Blackhair` on StyleGAN2 and StyleSwin and compared it against the $\alpha_{val}$. The results in Tab. 16 show that $\alpha_{val}$ is a good approximation of the $\alpha_{gen}$ of the generated dataset.

In addition, in Tab. 17, we further demonstrate the difference in effect when utilizing $\alpha_{gen}$ as opposed to $\alpha_{val}$ with CLEAM for sensitive attribute `Gender` on StyleGAN2 from the GenData dataset. Overall, we observed only marginal improvements, for most cases, when utilizing the $\alpha_{gen}$. Additionally, as the improvements by CLEAM were still very significant when utilizing the $\alpha_{val}$, and as the labels for the generated dataset are not readily available to evaluate $\alpha_{gen}$, we found the $\alpha_{val}$ to be a good approximation for $\alpha_{gen}$ for fairness measurement.

**Comparing $\alpha_{val}$ vs $\alpha_{gen}$ on SDM.** Similarly when evaluating the SDM with CLEAM, we also utilize $\alpha_{val}$ in-place of $\alpha_{gen}$. However, as a validation dataset is not readily available for SDM, we explored the use of a poxy validation dataset whose domain is a close representation as our applications. More specifically, we utilize CelebA-HQ as our proxy validation dataset (with known labels *w.r.t.* `Gender`) to attan $\alpha_{val}$. Then similarly, we compare $\alpha_{val}$ to $\alpha_{gen}$ from our labelled GenData-SDM (per prompt). As shown in Tab. 18 our approximated $\alpha_{val}$ (measured on CelebA-HQ), although not perfect, is a close approximation of $\alpha_{gen}$, thereby making it appropriate to be utilized with CLEAM.

Table 16: Comparing $\alpha_{val}$ ( $\alpha$ measured during the classifier's validation stage using real data), against $\alpha_{gen}$ ( $\alpha$ measured on the generated dataset). Notice that the $\alpha_{val}$ measured during the validation dataset is a close approximation of the generated dataset's $\alpha_{gen}$.

| | StyleGAN2 | | | | StyleSwim | | | |
|---|---|---|---|---|---|---|---|---|
| | ResNet18 | ResNet34 | MobileNetv2 | VGG16 | ResNet18 | ResNet34 | MobileNetv2 | VGG16 |
| | Gender | | | | | | | |
| Validated $\alpha$ | [0.947,0.983] | [0.932,0.976] | [0.938, 0.975] | [0.801,0.919] | [0.947,0.983] | [0.932,0.976] | [0.958, 0.975] | [0.801,0.919] |
| $\alpha_{gen}$ | [0.940,0.984] | [0.928,0.982] | [0.948, 0.985] | [0.815,0.922] | [0.957,0.966] | [0.944,0.981] | [0.956, 0.977] | [0.804,0.924] |
| | Blackhair | | | | | | | |
| Validated $\alpha$ | [0.869,0.884] | [0.834,0.919] | [0.839,0.880] | [0.850,0.836] | [0.869,0.884] | [0.834,0.919] | [0.839,0.880] | [0.850,0.836] |
| $\alpha_{gen}$ | [0.870,0,885] | [0.830,0.914] | [0.845,0.886] | [0.837,0.824] | [0.874,0.892] | [0.824,0.930] | [0.837,0.891] | [0.847,0.821] |

Table 17: Comparing the *point estimates* and *interval estimates* of Baseline and our proposed CLEAM measurement framework in estimating $p^*$ of the GenData datasets sampled from (A) StyleGAN2 [6]. The $p_0^*$ value for each GAN with a certain attribute is determined by manually hand-labeling the generated data. We then utilize a Resnet-18 to classify attributes `Gender` to obtain $\hat{p}$. Then with different accuracy $\alpha$, measured from the validation split (denoted by $\alpha_{val}$) and GenData datasets (denoted by $\alpha_{gen}$), we apply CLEAM. Each $\hat{p}$ utilizes $n = 400$ samples and is evaluated for a batch-size of $s = 30$. We repeat this for 5 experimental runs and report the mean error rate, per Eqn. 1 from the main manuscript.

| | Point Estimate | | | | | | Interval Estimate | | | | | |
|---|---|---|---|---|---|---|---|---|---|---|---|---|
| Classifier | Baseline[2] | | CLEAM (Ours) with $\alpha_{val}$ | | CLEAM (Ours) with $\alpha_{gen}$ | | Baseline[2] | | CLEAM (Ours) with $\alpha_{val}$ | | CLEAM (Ours) with $\alpha_{gen}$ | |
| | $\mu_{\text{Base}}$ | $e_\mu(\downarrow)$ | $\mu_{\text{CLEAM}}$ | $e_\mu(\downarrow)$ | $\mu_{\text{CLEAM}}$ | $e_\mu(\downarrow)$ | $\rho_{\text{Base}}$ | $e_\rho(\downarrow)$ | $\rho_{\text{CLEAM}}$ | $e_\rho(\downarrow)$ | $\rho_{\text{CLEAM}}$ | $e_\rho(\downarrow)$ |
| | (A) StyleGAN2 | | | | | | | | | | | |
| | Gender with GT class probability $p_0^*$=0.642 | | | | | | | | | | | |
| R18 | 0.610 | 4.98% | 0.638 | 0.62% | 0.639 | **0.44%** | [0.602, 0.618] | 6.23% | [0.629, 0.646] | **2.02%** | [0.629, 0.648] | **2.02%** |
| R34 | 0.596 | 7.17% | 0.634 | 1.25% | 0.635 | **1.06%** | [0.589, 0.599] | 8.26% | [0.628, 0.638] | 2.18% | [0.630, 0.640] | **1.87%** |
| MN2 | 0.607 | 5.45% | 0.637 | **0.78%** | 0.636 | 0.86% | [0.602, 0.612] | 6.23% | [0.632, 0.643] | **1.56%** | [0.630, 0.642] | 1.82% |
| V16 | 0.532 | 17.13% | 0.636 | 0.93% | 0.640 | **0.36%** | [0.526, 0.538] | 18.06% | [0.628, 0.644] | 2.18% | [0.632, 0.647] | **1.53%** |
| Avg Error | | 8.68% | | 0.90% | | **0.68%** | | 9.70% | | 1.99% | | **1.81%** |

Table 18: Comparing the approximate $\boldsymbol{\alpha}_{val}$ measured on CelebA-HQ versus CLIP's $\boldsymbol{\alpha}_{gen}$ evaluated on a fair distribution of GenData-SDM for each prompt *w.r.t.* `Gender`.

| Dataset | Stable Diffusion Model | | | | |
|---|---|---|---|---|---|
| CelebA-HQ | "Somebody" | "an individual" | "a human being" | "a person" | "one person" |
| $\boldsymbol{\alpha}$   [0.998,0.975] | [1.0,0.970] | [1.0,0.980] | [1.0,0.970] | [0.990, 0.970] | [1.0, 0.980] |

## D.8    Comparing CLEAM with Classifier Correction Techniques (BBSE/BBSC)

In this section, we compare CLEAM against a few classifier correction techniques. We remark that CLEAM, unlike the classifier correction techniques, does not aim to improve the sensitive attribute classifier's accuracy but instead accounts for its errors during fairness measurement. However, given that classifier correction techniques may improve bias measurement, we found it useful to make a comparison. Specifically, we look into Black-Box shift estimator/correction (BBSE/BBSC) [18], methods previously proposed to address classifier inaccuracies due to label shift. We demonstrate that even with BBSE/BBSC, errors in bias measurement still remain significant.

**Setup.** To determine the effectiveness of BBSE/BBSC in tackling the errors of fairness measurement in generative models we evaluate it on the same setup as per Sec. 5.1 of the main manuscript on GenData-StyleGAN and GenData-StyleSwin with ResNet-18. Specifically, for BBSE we follow Lipton *et al.* [18] and first evaluate the confusion matrix for the trained ResNet-18 based on the validation dataset. Then, utilizing the confusion matrix, we calculate the weight vector which accounts for label shift of the generated data. With this weight vector, we now implement a variant of CLEAM utilizing Algo.1 (with the weighted vector in-place of $\boldsymbol{\alpha}$) in the main manuscript to evaluate the PE and IE. Similarly, for BBSC, we calculate the weight vector. However, unlike BBSE, we now utilize the weighted vector and fine-tune the classifier on the generated samples [18].

Our results in Tab. 19 shows that BBSE helps to marginally reduce $e_\mu$ and $e_\rho$ for the PE and IE, when compared against the baseline. However, these results still remain poor when compared to our original CLEAM implementation. One reason for this difference may be that, unlike CLEAM which is agnostic to the cause of the error, BBSE specifically corrects for label shifts while neglecting other sources of error *e.g.* task hardness. Meanwhile, our results in Tab. 20 show that utilizing BBSC makes no improvement in the improving the baseline fairness measurements. We hypothesize that this is due to the strong assumption of invariant conditional input distribution $p(x|y)$ used in BBSC, which may not hold in our problem. Overall we conclude that while classifier correction techniques may improve fairness measurements in some cases, they may not always generalize as they are often tailored to a specific problem.

Table 19: Comparing BBSE and CLEAM in estimating $\boldsymbol{p^*}$ on GenData-StyleGAN2 and GenData-StyleSwin *w.r.t.* {`Gender`,`BlackHair`}. Here, we utilize a ResNet-18 trained on the CelebA-HQ dataset.

| Point Estimate | | | | | | Interval Estimate | | | | | |
|---|---|---|---|---|---|---|---|---|---|---|---|
| Baseline | | BBSE | | CLEAM (Ours) | | Baseline | | BBSE | | CLEAM (Ours) | |
| $\mu_{\text{Base}}$ | $e_\mu(\downarrow)$ | $\mu_{\text{BBSE}}$ | $e_\mu(\downarrow)$ | $\mu_{\text{CLEAM}}$ | $e_\mu(\downarrow)$ | $\rho_{\text{Base}}$ | $e_\rho(\downarrow)$ | $\rho_{\text{BBSE}}$ | $e_\rho(\downarrow)$ | $\rho_{\text{CLEAM}}$ | $e_\rho(\downarrow)$ |
| **(A) StyleGAN2** | | | | | | | | | | | |
| Gender with GT class probability $p_0^*$=**0.642** | | | | | | | | | | | |
| 0.610 | 4.98% | 0.621 | 3.38% | 0.638 | **0.62%** | [0.602,0.618] | 6.23% | [0.613,0.628] | 4.52% | [0.629,0.646] | **2.02%** |
| BlackHair with GT class probability $p_0^*$=**0.643** | | | | | | | | | | | |
| 0.599 | 6.48% | 0.630 | 2.02% | 0.641 | **0.31%** | [0.591,0.607] | 8.08% | [0.622,0.638] | 3.27% | [0.631,0.652] | **1.40%** |
| **(B) StyleSwin** | | | | | | | | | | | |
| Gender with GT class probability $p_0^*$=**0.656** | | | | | | | | | | | |
| 0.620 | 5.49% | 0.628 | 4.27% | 0.648 | **1.22%** | [0.612,0.629] | 6.70% | [0.620,0.634] | 5.49% | [0.639,0.658] | **2.59%** |
| BlackHair with GT class probability $p_0^*$=**0.668** | | | | | | | | | | | |
| 0.612 | 8.38% | 0.640 | 4.20% | 0.659 | **1.35%** | [0.605,0.620] | 9.43% | [0.633,0.647] | 5.24% | [0.649,0.670] | **2.84%** |

Table 20: Comparing fairness distribution with ResNet-18 trained with and without Black-Box Shift Correction (BBSC) on the GenData dataset. Here we utilize the prior work's fairness measurement framework (Baseline) and our proposed CLEAM to evaluate the fairness distribution.

| | | | Point Estimate | | | | Interval Estimate | | | |
| | | | Baseline | | CLEAM (Ours) | | Baseline | | CLEAM (Ours) | |
| Setup | $\alpha$ | Avg $\alpha$ | $\mu_{\text{Base}}$ | $e_\mu(\downarrow)$ | $\mu_{\text{CLEAM}}$ | $e_\mu(\downarrow)$ | $\rho_{\text{Base}}$ | $e_\rho(\downarrow)$ | $\rho_{\text{CLEAM}}$ | $e_\rho(\downarrow)$ |
|---|---|---|---|---|---|---|---|---|---|---|
| | | | **(A) StyleGAN2** | | | | | | | |
| | | | Gender with GT class probability $p_0^*$=**0.642** | | | | | | | |
| Original Classifier | {0.947,0.983} | 0.97 | 0.610 | 4.98% | 0.638 | 0.62% | [0.602,0.618] | 6.23% | [0.629,0.646] | 2.02% |
| Adapted Classifier w. BSSC | {0.932,0.980} | 0.96 | 0.609 | 5.28% | 0.645 | 0.46% | [0.601,0.616] | 6.53% | [0.635,0.655] | 2.02% |
| | | | BlackHair with GT class probability $p_0^*$=**0.643** | | | | | | | |
| Original Classifier | {0.869,0.885} | 0.88 | 0.599 | 6.48% | 0.641 | 0.31% | [0.591,0.607] | 8.08% | [0.631,0.652] | 1.40% |
| Adapted Classifier w. BSSC | {0.854,0.875} | 0.86 | 0.588 | 8.55% | 0.635 | 1.24% | [0.581,0.596] | 9.64% | [0.627,0.643] | 2.49% |
| | | | **(B) StyleSwin** | | | | | | | |
| | | | Gender with GT class probability $p_0^*$=**0.656** | | | | | | | |
| Original Classifier | {0.947,0.983} | 0.97 | 0.620 | 5.49% | 0.648 | 1.22% | [0.612,0.629] | 6.70% | [0.639,0.658] | 2.59% |
| Adapted Classifier w. BSSC | {0.932,0.980} | 0.96 | 0.617 | 5.94% | 0.655 | 0.15% | [0.610,0.614] | 7.01% | [0.649,0.661] | 1.06% |
| | | | BlackHair with GT class probability $p_0^*$=**0.668** | | | | | | | |
| Original Classifier | {0.869,0.885} | 0.88 | 0.612 | 8.38% | 0.659 | 1.35% | [0.605,0.620] | 9.43% | [0.649,0.670] | 2.84% |
| Adapted Classifier w. BSSC | {0.854,0.875} | 0.86 | 0.608 | 8.98% | 0.663 | 0.75% | [0.600,0.616] | 10.18% | [0.655,0.671] | 1.95% |

### D.9 Applying CLEAM to Re-evaluate Bias Mitigation Algorithms

*Importance-weighting* [2] is a simple and effective method for bias mitigation. However, its performance in fairness improvement is measured by the Baseline, which could be erroneous. In this section, we re-evaluate the performance of importance-weighting with CLEAM, which has shown better accuracy in fairness estimation.

Following Choi *et al.* [2], we utilize the original source code to train two BIGGANs [19] on CelebA [8]: for the first GAN, without applying any bias mitigation (Unweighted), while in the second, we apply importance re-weighting (Weighted). We do this for the originally proposed sensitive attribute `Gender`, and extend the experiment to `BlackHair`. For a fair comparison, we follow [2] and similarly use a ResNet-18 with a reasonably high average accuracy of $88\%$ and $97\%$ for sensitive attributes `BlackHair` and `Gender`. Our results in Tab. 21 show that Baseline measures a $\mu_{\text{Base}}$ of 0.727 and 0.680 for Unweighted and Weighted, with SA `Gender` (similar to reported results in [2]). Meanwhile, CLEAM's results show that $\mu_{\text{CLEAM}} > \mu_{\text{Base}}$, implying that previous work could have underestimated the bias of the GANs. This could lead to an erroneous evaluation of a bias mitigation technique, or comparison across different bias mitigation techniques. Then, when analyzing bias mitigation techniques using IE of CLEAM (as per Tab. 22), since the IE of unweighted and weighted GANs do not overlap, we are provided with some statistical guarantees that the bias mitigation techniques, importance-weighting, is indeed effective.

Table 21: Re-evaluating the *point estimates* of previously proposed bias mitigation method, importance-weighting (imp-weighting) [2] with CLEAM. We first evaluate the bias of a BIGGAN [19] with and without imp-weighting *i.e.* unweighted and weighted, with the Baseline. Then, we apply CLEAM to obtain a more accurate measurement. We do this for both `Gender` and `BlackHair` sensitive attributes.

| Setup | Baseline | Diversity | CLEAM (Ours) |
|---|---|---|---|
| | $\mu_{\text{Base}}$ | $\mu_{\text{Div}}$ | $\mu_{\text{CLEAM}}$ |
| $\alpha$=[0.976,0.979], `Gender` | | | |
| Unweighted | 0.727 | 0.711 | 0.738 |
| Weighted | 0.680 | 0.671 | 0.690 |
| $\alpha$=[0.881,0.887], `BlackHair` | | | |
| Unweighted | 0.729 | 0.716 | 0.803 |
| Weighted | 0.716 | 0.706 | 0.785 |

Table 22: Re-evaluating the *interval estimates* of previously proposed bias mitigation method, importance-weighting (imp-weighting) [2] with CLEAM. To do this, we first evaluate the bias of a BIGGAN [19] with and without implementing imp-weighting *i.e.* unweighted and weighted, with the Baseline. Then, we apply CLEAM to obtain more accurate measurements, which we use to compare against the Baseline. We do this for both `Gender` and `BlackHair` sensitive attributes.

| Setup | Baseline | Diversity | CLEAM(Ours) |
|---|---|---|---|
| | $\rho_{\text{Base}}$ | $\rho_{\text{Div}}$ | $\rho_{\text{CLEAM}}$ |
| $\alpha$=[0.976,0.979], `Gender` | | | |
| Unweighted | $[0.721, 0.732]$ | $[0.697, 0.722]$ | $[0.733, 0.744]$ |
| Weighted | $[0.674, 0.685]$ | $[0.658, 0.684]$ | $[0.686, 0.693]$ |
| $\alpha$=[0.881,0.887], `BlackHair` | | | |
| Unweighted | $[0.725, 0.733]$ | $[0.704, 0.729]$ | $[0.798, 0.809]$ |
| Weighted | $[0.710, 0.722]$ | $[0.696, 0.716]$ | $[0.778, 0.792]$ |

# E  Details on Applying CLIP as a SA Classifier

**CLIP as a Sensitive Attribute classifier.** To utilize CLIP as a sensitive attribute classifier (with the VIT-B/32 architecture), we follow the best practices suggested by Radford *et al.* [20]. Here, we first input two different prompts, describing the respective classes, to the CLIP text-encoder, as seen in Tab. 23. As suggested by Radford *et al.* we utilize the prompt starting with "A photo of a" *i.e.* a scene description, followed by our sensitive attribute's classes *e.g.* female/male. Next, we also encode the generated images with the CLIP image encoder. Finally, for each encoded generated image and the two encoded text-prompt, we take the cosine similarities followed by the $\arg\max$. The $\arg\max$ output provides us with the respective hard label of the generated image.

**Generated Image pre-processing.** We remark that as the stable diffusion model produces a mixture of colored and greyscale images, for a fair comparison, we transform all images from RGB to greyscale before feeding into CLIP for classification.

Table 23: Prompts for using CLIP [20] for sensitive attribute classification .

| Sensitive Attribute | Class 0 prompt | Class 1 prompt |
|---|---|---|
| Gender | "A photo of a female" | "A photo of a male" |
| Smiling | "A photo of a face not smiling" | "A photo of a face smiling" |

# F    Ablation Study: Details of Hyper-Parameter Settings and Selection

**Sensitive Attribute Classifier $C_u$.**   In our experiments, we utilized a Resnet-18/34 [7], MobileNetv2 [12] and VGG-16. The respective datasets *i.e.* CelebA, [8] CelebA-HQ [21] and AFHQ [9] datasets are then segmented into {Train, Test, Validation} with respect to the ratio {80%,10%,10%}, where each segmentation of the dataset contains uniform distribution *w.r.t.* the queried sensitive attribute. The classifiers are then trained with the training datasets and the $\alpha$ are evaluated with the validation dataset. Each classifier is trained with an Adam optimizer[22] with a learning rate=$1e^{-3}$, Batch size=64 and input dim=64x64 from the CelebA dataset [8], dim=64x64 from the AFHQ dataset and dim=128x128 from the CelebA-HQ dataset [21]. Tab. 24 details the $\alpha$ of the ResNet-18 utilized in Sec.6 of our main manuscript.

Table 24: Accuracy of ResNet-18 trained and evaluated on CelebA-HQ.

| Sensitive Attribute | Accuracy, $\alpha$ |
|---|---|
| NoBeard | [0.968,0.898] |
| HeavyMakeup | [0.925,0.883] |
| Bald | [0.930,0.972] |
| Chubby | [0.838,933] |
| Mustache | [0.925,0.896] |
| Smiling | [0.933, 0.877] |
| Young | [0.871, 0.857] |
| BlackHair | [0.869,0.885] |
| Gender | [0.947,0.983] |

**Generator $G_\phi$ used in sec.D.9.**   As mentioned in sec. D.9, we utilized the setup in Choi *et al.* [2][2] for the training of our imp-weighted and unweighted GANs. With this, we replicate their hyperparameter selection of 64 x 64 celebA [8] images with a learning rate=$2e^{-4}$, $\beta_1 = 0$, $\beta_2 = 0.99$ and four discriminator steps per generator step. We utilize a single RTX3090 for the training of our models.

**Evaluating CLEAM with Different n.** Utilizing the same setup in Sec. 5.1 of our main manuscript – with the GenData-StyleGAN and GenData-StyleSwin datasets, we repeated the experiment with ResNet-18 and $n \in [100, 600]$. Our results in Fig.5 show that there is a marginal increase in error for both the Baseline and CLEAM as $n$ approaches 100, while the converse occurs when $n$ approaches 600. However, given the diminishing improvements for $n > 400$, we found $n = 400$ to be ideal- a balance between computational cost and measurement accuracy.

**Batch Size $s$.**   In our experiments, we utilized $s$ batches of data each of which contains $n$ images to approximate $p^*$ with the Baseline and CLEAM methods. In the previous experiment, we found $n = 400$ samples to be the ideal balance between computational time and minimizing fairness measurement error. Here, we repeat the same hyper-parameter search, utilizing the real generator setup in Sec 5.1 of the main paper with ResNet-18, but instead varied the number of batches, $s$. Our results in Fig. 6 found $s = 30$ to be the optimal value when approximating $p^*$. Increasing $s$ did not result in significant improvements by both baseline and CLEAM. However, decreasing $s$ did observe some significant degradation in performance *i.e.* increase in $e_\mu$.

**Computational Time.**   In our main paper, we note that CLEAM is a lightweight correction to the existing baseline method, that requires no additional parameter to be computed during evaluation. To support this, we evaluated the computational time for the Baseline, Diversity, and our proposed CLEAM. Our results in Tab. 25 show that there is only a small difference in computational time ($\approx 0.1$s) between the Baseline and our proposed CLEAM. This difference is solely to facilitate the computation of Algo. 1. See Tab. 26 for discussion on carbon emission.

---

[2]https://github.com/ermongroup/fairgen

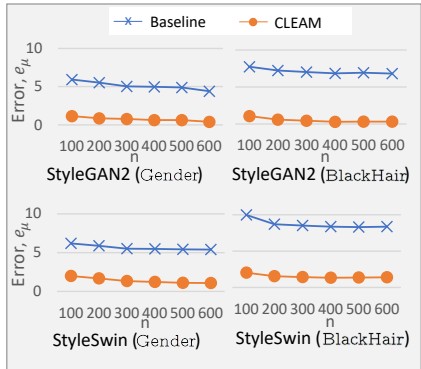

Figure 5: Comparing the point error $e_\mu$ for Baseline and CLEAM when evaluating the bias of GenData with ResNet-18, with varying sample size, $n$.

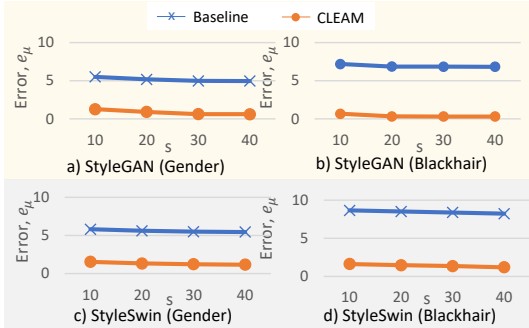

Figure 6: Comparing the point error $e_\mu$ for Baseline and CLEAM when evaluating the bias of the generated data with ResNet-18, for varying sample the number of batches, $s$.

Table 25: Average computation time for estimating $p^*$ with $s$=30 and $n$=400 for Baseline [2], Diversity [5] and our proposed CLEAM with a single RTX3090 for 5 consecutive runs.

|  | Baseline [2] | Diversity [5] | CLEAM (Ours) |
|---|---|---|---|
| **CelebA, 64x64** , $s$ | 99.9 | 600.4 | 100.0 |
| **AFHQ, 64x64** , $s$ | 99.8 | 601.2 | 99.9 |
| **CelebA-HQ, 128x128**, $s$ | 135.9 | 820.4 | 136.0 |

Table 26: **Estimated Computation time**. The carbon emission values are computed using `https://mlco2.github.io/impact`.

| Experiment | Hardware | GPU Hours | Carbon emitted (kg) |
|---|---|---|---|
| Training of SA Classifiers | RTX3090 | 2.0 | 0.39 |
| Comparing CLEAM on GANs, Main Paper Tab. 1 | RTX3090 | 4.8 | 0.94 |
| Comparing CLEAM on DGN, Main Paper Tab. 2 | RTX3090 | 0.3 | 0.1 |
| Inferring with CLEAM on DGN, Main Paper Fig. 3a | RTX3090 | 0.3 | 0.1 |
| Inferring CLEAM on GANs, Main Paper Fig. 3b | RTX3090 | 0.52 | 0.15 |
| Comparing CLEAM on PsuedoG, Supp Tab 11 | RTX3090 | 4.5 | 0.88 |
| Comparing CLEAM on PsuedoG Additional SA, Supp Tab 12 | RTX3090 | 3 | 0.59 |
| Comparing CLEAM on PsuedoG Additional classifier, Supp Tab 13 | RTX3090 | 4.5 | 0.88 |
| Comparing CLEAM on DGN with CLIP, Supp Tab. 14 | RTX3090 | 0.15 | 0.05 |
| Comparing CLEAM with BBSE/BBSC, Supp Tab. 19 | RTX3090 | 0.25 | 0.07 |
| Applying CLEAM on Bias mitigation, Subb Tab 21 | RTX3090 | 0.88 | 0.17 |
| **Total:** | | 21.2 | 4.32 |

# G Related Work

**Fairness in Generative Models.** Fairness in machine learning is mostly studied for discriminative learning, where usually the objective is to handle a classification task independent of a sensitive attribute in the input data, *e.g.* making a hiring decision independent of the applicant `Gender`. However, the definition of fairness is quite different for generative learning, where it is considered as equal representation/generation probability *w.r.t.* a sensitive attribute. Because of this difference, the conventional fairness metrics used for classification, like Equalised Odds, Equalised Opportunity [23] and Demographic Parity [24], cannot be applied to generative models. Instead, the similarity between the probability distribution of the generated sample *w.r.t.* a sensitive attribute ($p^*$) and a target distribution $\bar{p}$ (a uniform distribution) [2] is utilized as fairness metric. See sec. A.3 for details.

**Existing Works on Fair Generative Models.** Existing works focus on ***bias mitigation*** in generative models. The importance reweighting algorithm is proposed by Choi *et al.* [2] where a re-weighting algorithm favours a reference fair dataset *w.r.t.* the sensitive attribute in-place of a larger biased dataset. Frankel *et al.* [16] introduces the concept of prior modification, where an additional smaller network is added to modify the prior of a GAN to achieve a fairer output. Tan *et al.* [25] learns the latent input space *w.r.t.* the sensitive attribute, which they can later sample accordingly to achieve a fair output. MaGNET [26] demonstrates that enforcing uniformity in the latent feature space of a GAN, through a sampling process, improves fairness. Um *et al.* [4] improves fairenss through the utilization of total variation distance which quantifies the unfairness between a small reference dataset and the generated samples. Teo *et al.* [3] introduces fairTL++, which utilizes a small fair dataset to implement fairness adaptation via transfer learning. In all of these works, the focus is on improving fairness of the generative model (where the performance of the model is measured with a framework, in which the inaccuracies in the sensitive attribute classifier has been ignored). However, our proposed CLEAM method focuses on improving ***fairness measurement***, by compensating for the inaccuracies in the sensitive attribute classifier through a statistical model. Therefore, it can be used to evaluate the bias mitigation algorithms more accurately.

**Equal Representation.** Some literature also use a similar notion of equal representation (used in generative models) to address fairness. For example, fair clustering variation [27] is proposed by enforcing the clusters to represent each attribute equally, and fair data summarization [28] is used to mitigate the bias in creating a representative subset for a given dataset, while handling the trade-offs between fairness and diversity during sampling. However, unlike our setup, these works assume to have access to the attribute labels. Meanwhile, in data mining, a similar problem was recently studied. Given a large dataset of unlabelled mined data, the objective is to evaluate the disparity of the dataset *w.r.t.* an attribute. To do this, an evaluation framework called diversity [5] was introduced. To measure this, a pre-trained classifier is used as a feature extractor. The unlabelled dataset is then compared against a controlled reference dataset (with known labels) via a similarity algorithm.

**Biases in Text-Image generation.** Some literature have attempted to look into the biases in text-to-image generators [29]. Specifically, Bianchi *et al.* study existing biases in occupations-based prompts for popular text-to-image generators *e.g.* stable diffusion models. They found the biases to exasperate existing occupation stereotypes, *e.g.* nurses being over-represented as non-Caucasian females. To measure these biases, [29] has a simple approach utilizing a pre-trained feature extractor to assign the sensitive attribute labels to a small batch of generated images (100 samples). We remark that this approach is similar to Diversity, a method which we found to also demonstrate significant errors due to the lack of consideration for the classifier's error. Furthermore, we emphasize the difference between our study and Bianchi *et al.* . Specifically, in our application of CLEAM (Sec. 6 of the main manuscript), we examine the impact of using prompts with indefinite pronouns/nouns that are synonymous to each another. Our objective, unlike Bianchi *et al.* 's work, is to investigate the influence of subtle changes in the prompts on bias, which is studied on a large dataset ($\approx 2k$ samples). Our results are the first to demonstrate that even subtle changes to the prompt (which are semantically unchanged), could result in drastically different biases.

**Classifier Calibration.** The proposed CLEAM can be seen from a classifier calibration point of view as it refines the output of the classifier. However, CLEAM should not be mistaken with conventional calibration algorithms, *e.g.* temperature scaling [30], Platt Scaling [31] and Isotonic regression [32]. Unlike these algorithms that concern themselves with the confidence of prediction, CLEAM focuses on sensitive attribute distribution, thereby making these algorithms ineffective.

More specifically, conventional classifier calibration methods usually work on soft labels (probabilities). Note that in our framework, the $argMax$ is applied to the output probabilities to determine the hard label. Therefore, in our application that deals with hard labels, regular classification techniques are less effective. To investigate this, we conduct a few calibration experiment by applying some popular classifier calibration techniques; temperature scaling(T-scaling) [30], Isotonic Regression[32] and Platt Scaling[31] on a pre-trained ResNet-18[7] senstive attribute classifier. In Fig. 7, we see that T-scaling is the most effective in correcting the calibration curve to the ideal Ref line. Note that, this Ref line indicates that the classifier is perfectly calibrated *w.r.t.* the soft labels.

Next, using the pseudo-generator from Sec. D.3, we utilised the calibrated sensitive attribute classifiers earlier and compare them against CLEAM (which was applied on an uncalibrated model). **In our results,** seen in Fig. 8, we observe that these traditional calibration methods are less effective in correcting the sensitive attribute distribution error. In fact, methods like Platt scaling worsen the error, and T-scaling —which is shown in [30] and our experiment to be one of the most effective traditional calibration methods— does not change class predictions (hard labels), but merely perturb the soft labels. This demonstrates that traditional calibration technique are not direct correlation to hard label calibration, which CLEAM aims to address.

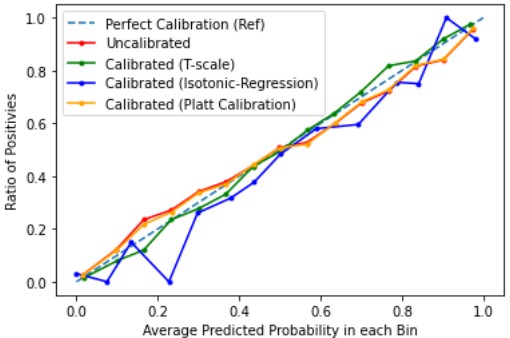

Figure 7: Calibration Curve on ResNet-18 for `Attractive` sensitive attribute. We observe that the T-scaling is the most effective technique in improving soft label calibration and Isotonic regression the worst. However, this same trend does not follow in the hard label errors of Fig 8.

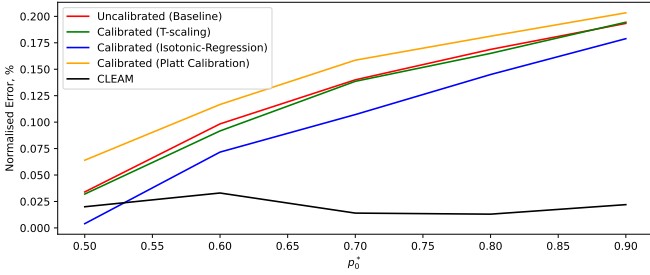

Figure 8: **Comparing Calibration Techniques**: Using the pseudo-generator, we compare CLEAM against well known calibration techniques, overall we observe that previous techniques are significantly less effective, achieving an average error of; T-Scaling: $12.4\%$, Isotonic Regression: $10.1\%$, Platt Calibration: $14.5\%$ and uncalibrated (baseline): $12.4\%$ against CLEAM: $2.0\%$

# H  Details of the GenData: A New Dataset of Labeled Generated Images

In this section, we provide more information on our new dataset, containing generated samples labeled based on sensitive attributes from StyleGAN2[3] [6] and StyleSwin [4] [11] trained on CelebA-HQ [21], and a Stable Diffuson Model(SDM)[14]. More specifically, our dataset contains $\approx$9k randomly generated samples based on the original saved weights and codes of the respective GANs, and $\approx$2k samples for four different prompts inputted in the SDM. These samples are then hand labeled *w.r.t.* the sensitive attributes. More specifically, `Gender` and `BlackHair` for both the GANs and `Gender` for the SDM. Then with these labeled datasets, we can approximate the ground-truth sensitive attribute distribution, $p^*$, of the respective GANs.

**Dataset Labeling Protocol.**  To ensure high-quality samples and labels in our dataset, we passed the dataset through Amazon Mechanical Turk, where labelers were given detailed guidelines and examples for identifying the individual sensitive attributes. In addition to the sensitive attribute option *e.g.* `Gender(Male)` or `Gender(Female)`, labelers were also given an "unidentifiable" option which they were instructed to select for low-quality samples, as per Fig, 9 and 13. We repeated this process for 4 runs *s.t.* each sample had the opinions of four independent labelers. Finally, each sample was assigned the label that the majority had selected.

Overall, the GANs and SDM received 97% and 99% unanimous agreement rates. This for example includes male, female, or unidentifiable, for the sensitive attribute `Gender`. We discard the samples that had been labeled unidentifiable and were left with a high-quality dataset as per Fig. 10, 11 and 12. We remark that the discarded samples consist only a small portion of the generated samples *i.e.* 3% of the GANs, and 1% of the SDM. Upon further evaluation, we found that the sensitive attribute classifiers appear to uniformly assign these (rejected) ambiguous samples a random class with low confidence. As a result, we can assume that the impact of disregarding these samples was insignificant to CLEAM's evaluation.

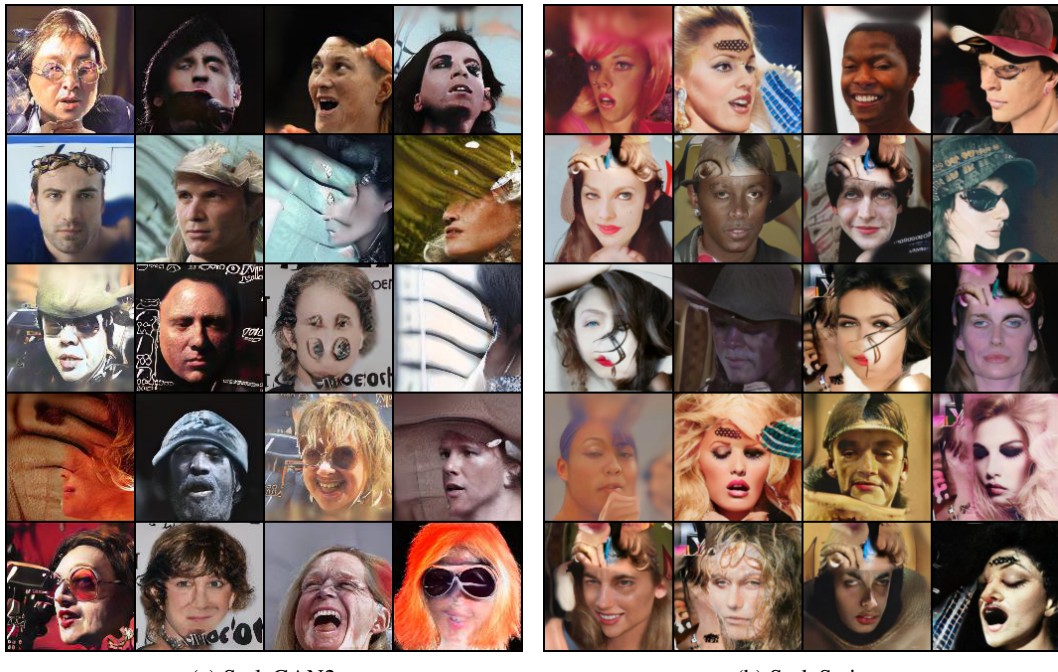

(a) StyleGAN2  (b) StyleSwin

Figure 9: Examples of rejected samples during hand-labeling due to poor quality.

---

[3]https://github.com/NVlabs/stylegan2-ada-pytorch
[4]https://github.com/microsoft/StyleSwin

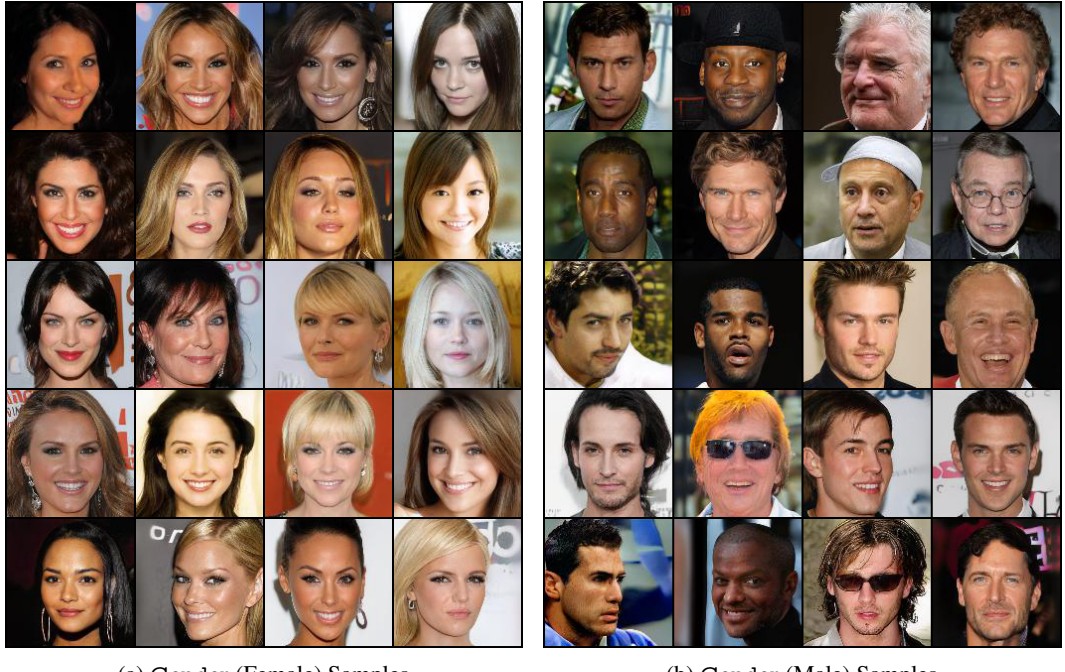

(a) Gender (Female) Samples          (b) Gender (Male) Samples

Figure 10: Examples of samples *w.r.t.* Gender sensitive attribute in our proposed GenData dataset.

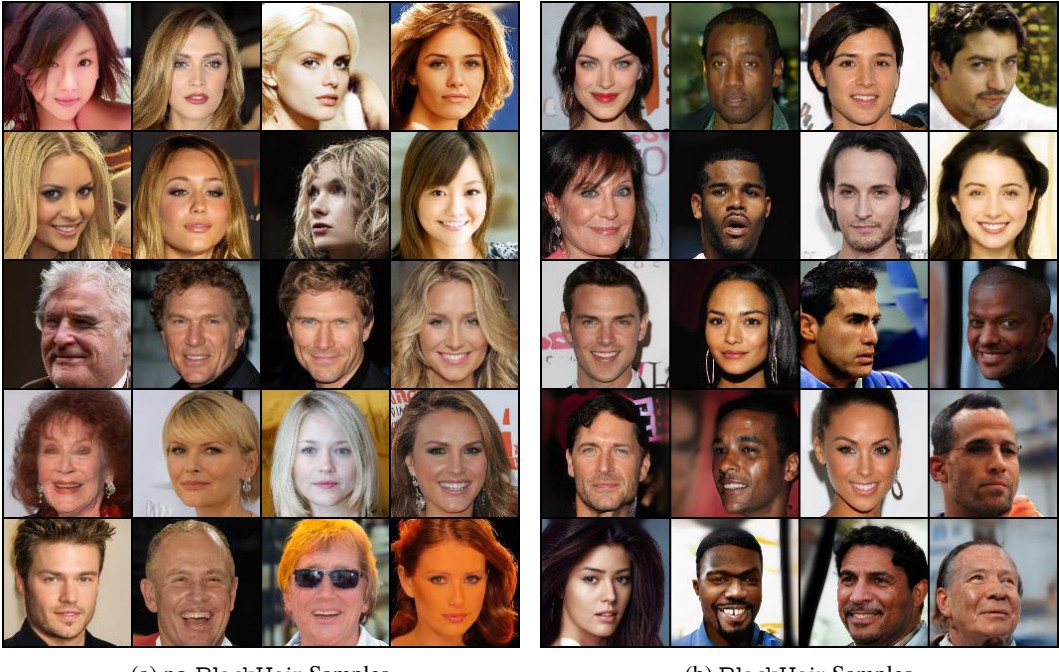

(a) no-BlackHair Samples          (b) BlackHair Samples

Figure 11: Examples of samples *w.r.t.* BlackHair sensitive attribute in our proposed GenData dataset.

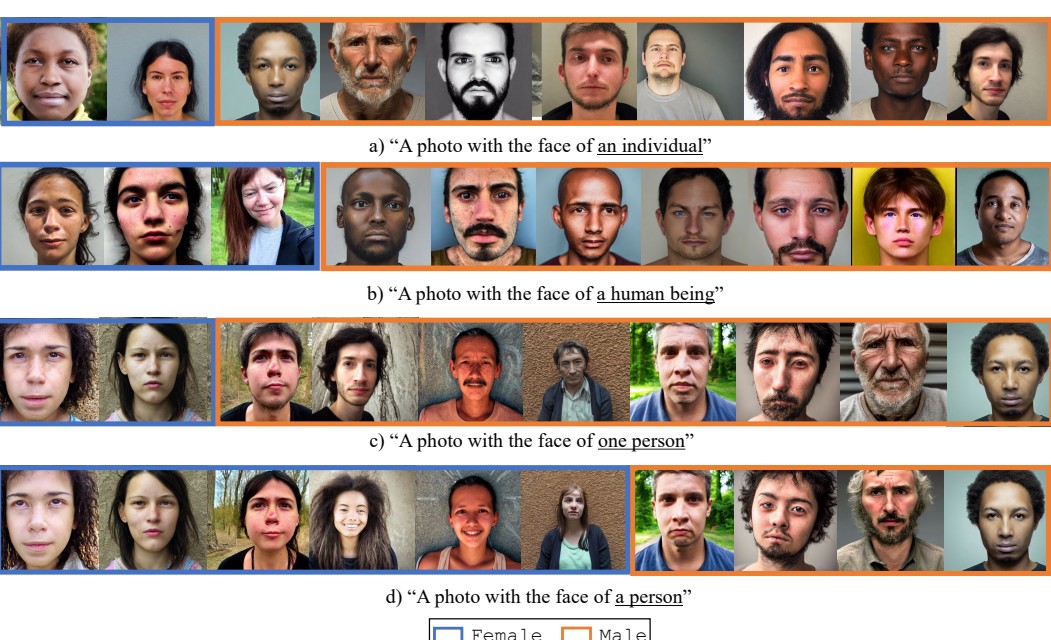

a) "A photo with the face of an individual"

b) "A photo with the face of a human being"

c) "A photo with the face of one person"

d) "A photo with the face of a person"

☐ Female ☐ Male

Figure 12: Examples of randomly generated samples based on the prompts "A photo with the face of an individual" and "A photo with the face of a human being" *w.r.t.* the sensitive attribute Gender.

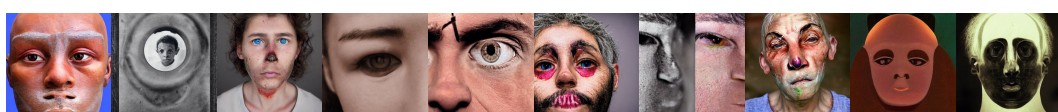

Figure 13: Examples of rejected samples from the SDM.

# I   Limitations and Considerations

**Ethical consideration.** In general, we note that our work does not introduce any social harm but instead improves on the existing fairness measurement framework to better gauge progress. However, we stress that it is important to consider the limitations of the existing fairness measurement framework, which we discuss in the following.

**Sensitive Attribute Labels.** Certain sensitive attributes may exist on a spectrum *e.g.* `Young`. However, given that this work aims to improve fairness measurement, and the current widely used definition is based on binary outcomes, we utilize the same setup in our work. Additionally, it is also important to be aware that certain sensitive attributes may be ambiguous *e.g.* `Big Nose` (which exist in popular datasets like CelebA-HQ), but definitions could differ based on different cultural expectations. In our work, we try to select less ambiguous sensitive attributes e.g., `BlackHair`.

**Human and Auto Labelling.** Labeling sensitive attributes in generative models is essential to better understand the possible biases that may exist in some proposed generative model algorithms. To do this, researchers often utilize either human labelers or machines for automated labeling. However, when utilizing such labeling procedures it is important to consider ethical implications, especially in many cases where sensitive information such as gender is involved. One particular concern is that there could be potential discrimination in the assignment of labels such as gender. For example, if only certain facial features are considered when assigning gender labels, some individuals may be inaccurately labeled due to their unique characteristics that deviate from traditional notions of male and female identity.

Human labelers may bring their own biases, subjectivity, and cultural background to the labeling process, which can lead to inaccuracies or reinforce stereotypes. Additionally, it is important to ensure that the labelers represent a diverse range of backgrounds and perspectives, particularly if the samples being labeled are from a diverse population. This can help mitigate potential discrimination against some social identities and improve the accuracy of the labeling process.

In the case when utilizing machines for labeling, it is important to be aware that labeling algorithms may be biased, depending on the data set it was trained on. If the data set is not diverse or balanced, the algorithm may produce inaccurate or biased results that reinforce stereotypes or discrimination against certain social identities.

**Utilizing Zero Shot Classifiers.** When utilizing pre-trained classifiers it is important to carefully select proxy validation dataset with a similar domain to the generated images. A significant mismatch in these two domains could result in an inaccurate approximation of $\alpha$, resulting in poor performance by CLEAM. Then similar to our previous discussion, we would also refrain from ambiguous sensitive attributes, as this may result in a mismatch between the proxy validation dataset and the pre-trained sensitive attribute classifier.