# OpenReview forum: "On Measuring Fairness in Generative Models"
_NeurIPS.cc/2023/Conference — NeurIPS 2023 poster_

### Official Review · Reviewer_6ZNP · 2023-06-27

**Soundness:** 3 good
**Presentation:** 2 fair
**Contribution:** 3 good
**Rating:** 6
**Confidence:** 3

**Summary:**

The paper considers the problem of measuring fairness in generative models. In particular, the paper has two main contributions: (1) they have produced a dataset of hand labeled (sensitive attributes, SA) dataset for various SOTA generative models; and (2) they have proposed a method for estimating the expected sensitive attribute distribution which utilizes the error rates of the SA classifier. These two contributions are used to show that SA classifiers with low error can still cause high errors in previous methods of approximating the sensitive attribute distribution.

**Strengths:**

- The paper presents a strong empirical study of measuring fairness in generative models, and outlines flaws in prior studies.
- CLEAM seems to be an intuitive correction to the naive baseline method.

**Weaknesses:**

1. One weakness of the paper is a hole in the narrative: how is $C_{u}$ obtained? In particular, it is unclear in the main-text on how one would generate a SA classifier. Furthermore, the assumption of knowing the underlying error rates should be discussed as a potential limitation. I assume that the SA classifier is trained on data which are not samples from the generative model one is trying to measure the fairness of (as otherwise we already have labels). In this case, the validity of error rates transferring might be questionable.
2. The notation in the paper is somewhat strange. In particular, as far as I can see, $ Pr(u \mid x )$ is not a probability … despite the notation. And it is further “aliased” as $ C_u(x)$ which is additionally confusing. I don’t see why one could not just define the “argmax classification” as $C_u(x)$ directly.
3. The notation of $\hat{p}$ and $p^{\*}$ seems consistent to me. In Section 2, it seems that $p^{\*}$ is the population statistic. However, in Section 3 $p^{\*}$ becomes the estimate generated from GenData. Yes, one could argue that GenData somewhat becomes the “new” population, but this change in perspective is not clear. I think it makes more sense to think of the $p^\*$ in this Section as a high quality estimate of the true statistic. It may be worth changing notation to reflect the 3 possible $p$’s: that generated by the SA-classifier, that generated by GenData, and the unknown true statistic which is being approximated by the former two.

Typos:
- Appendix Eq (3) LHS


**Questions:**

1. Relating to “Weakness 1”, how exactly are the SA classifiers typically obtained?
2. Further relating to “Weakness 1”, how well do the error rates of the SA classifier transfer? That is, what is the difference between the SA classifier’s error rates with the dataset it was trained on versus the error rates on GenData (which I am assuming is that of Table 1 & 2)?


**Limitations:**

I think the assumption of knowing the underlying error rate / accuracy needs to further discussed (see Questions)

---

> ### Author Rebuttal · Authors · 2023-08-08
>
> > **Q1**: "how is $C_u$ obtained?"
>
> **A1**: Thanks for your comment and we apologize if it was unclear. As discussed in the main paper (Sec.3), to obtain $C_u$, we strictly follow previous work (e.g., Imp-Weighting [1] and fairTL [2]). Particularly, we train the SA classifiers using the labeled datasets w.r.t. SAs based on standard training procedure (e.g., train a ResNet-18 as SA classifier on CelebA dataset considering BlackHair as SA). To follow previous work, we train various SA classifiers on different sensitive attributes. We remark that as mentioned in the main paper (Sec.3) we defer the details of training SA classifier to Supp. F, due to lack of space. In addition, as mentioned in the main paper (Sec. 3), we utilize CLIP as an additional SA classifier to explore zero-shot SA classification. The details of utilizing CLIP is included in Sec. E where after encoding images using image encoder, we define two related text prompts for our SA following the guidelines in CLIP [6], and encode these text prompts using text encoder. Then the cosine similarity between the encoded image and two encoded text prompts is used to get the output.
>
> $ $
>
> >**Q2**: “Furthermore, the assumption of knowing the underlying error rates should be discussed as a potential limitation. I assume that the SA classifier is trained on data which are not samples from the generative model one is trying to measure the fairness of (as otherwise we already have labels). In this case, the validity of error rates transferring might be questionable.”
>
> **A2**: Thank you for your comment, Reviewer’s understanding is correct: SA classifiers are trained on **real data**, and they are not samples from the generative model which fairness measurement is performed.
>
> However, in our work we have addressed the validity of error rate transferring. In Supp. D.7, we have validated that the error rate on real validation data can be transferred to the setups where we measure the attributes of the generated data. The results in Tab. 15 of Supp. D.7 shows that for two GANs used in our study (StyleGAN and StyleSwin), the obtained error rate ($\alpha$) from real validation data (real CelebA-HQ data) is similar to the error rate obtained from our labeled dataset of generated images. Results in Tab. 16 show that, in our proposed CLEAM, using error rate $\alpha$ from real validation data has similar performance as that using the error rate from generated images. Similar results are shown for Stable Diffusion in Tab. 17. We emphasize that as we mentioned in Supp. D.7, the error rate for generated data is not assumed to be available in our fairness measurement, and the analysis in Supp D.7 is solely to validate error rate transferring.
>
> We thank the Reviewer for the comment, and we will make this clear in the revised version.
>
> $ $
>
> >**Q3**: “I don’t see why one could not just define the “argmax classification” as $C_u(x)$ directly.”
>
> **A3**: We thank the Reviewer for the good suggestion and will directly define $C_u(x)$ as the argmax classification for clearer discussion.
>
> $ $
>
> >**Q4**: “It may be worth changing notation to reflect the 3 possible $p$’s: that generated by the SA-classifier, that generated by GenData, and the unknown true statistic which is being approximated by the former two.”
>
> **A4**: We appreciate the Reviewer's sharp observation. Reviewer's understanding is correct. In the paper, to avoid introducing additional symbol and with a slight abuse of notation, we use $p^*$ to denote population statistics in Sec 2, and $p^*$ to denote estimation from GenData in Sec 3. We will consider the Reviewer's suggestion seriously and update the submission.
>
> $ $
>
> >**Q5**: Relating to “Weakness 1”, how exactly are the SA classifiers typically obtained?
>
> **A5**: Please find the details to this question in A1.
>
> $ $
>
> >**Q6**:  Further relating to “Weakness 1”, how well do the error rates of the SA classifier transfer? That is, what is the difference between the SA classifier’s error rates with the dataset it was trained on versus the error rates on GenData (which I am assuming is that of Table 1 & 2)?
>
> **A6**:  Addressed earlier in previous questions.

---

> > ### Comment · Reviewer_6ZNP · 2023-08-14
> > **Re: Response**
> >
> > Thanks you for the detailed response.
> > I currently have no further questions and will keep my scores for now.

---

> > > ### Author Response · Authors · 2023-08-15
> > > **Thank you for the positive feedback and insightful comments**
> > >
> > > We sincerely thank the reviewer for the insightful comments. Thank you for the positive feedback and evaluation. We will include all additional results in the revised version. Sincerely, Authors

---

### Official Review · Reviewer_o1Z2 · 2023-07-06

**Soundness:** 3 good
**Presentation:** 3 good
**Contribution:** 2 fair
**Rating:** 6
**Confidence:** 4

**Summary:**

This paper considers the fairness measurement for generative models. The contributions of this paper are three-fold. First, the authors reveal that the existing frameworks have significant measurement errors, even using sensitive attribute classifiers. Second, the authors propose a new framework namely CLEAM that uses a statistical model to evaluate inaccuracies of SA classifiers, thus reducing the measurement errors. Finally, the authors use the proposed CLEAM to measure fairness in important text-to-image generators and GANs, which show the effectiveness of the proposed framework. Experimental results with a manually labeled dataset show that the proposed CLEAM can achieve lower error as compared with some baseline schemes.

**Strengths:**

1)Propose a novel framework to measure the fairness of generative models from both theoretical and experimental perspectives.
2)The fundamental statistic model is easy to follow, and the proposed method CLEAM is able to reduce the fairness measurement error.
3)The dataset created in this paper will benefit the research community.

**Weaknesses:**

After reading this manuscript, this reviewer finds that there are some issues that need to be addressed, as
1)On page 6, the authors say that “the probability of the counts for each output cT in Eqn. 2 (denoted by Nc) can be modeled by a multinomial distribution.” Does this assumption hold in practical systems?
2)The presentation of the statistical model should be improved. Why do the authors assume a multivariate Gaussian distribution instead of other statistical distributions?
3)On page 6, what does "M" represent in equation (3)?
4)The description of equation (8) is not clear enough. The authors are suggested to explain it in detail.
5)The authors are suggested to provide more experiments with other datasets and generative models, in order to demonstrate the effectiveness of the proposed framework.

**Questions:**

Please refer to my comments for details.

**Limitations:**

Please refer to my comments for details.

---

> ### Author Rebuttal · Authors · 2023-08-08
>
> >**Q1**: “On page 6, the authors say that “the probability of the counts for each output $c^T$ in Eqn. 2 (denoted by $N_c$) can be modeled by a multinomial distribution.” Does this assumption hold in practical systems?”
>
> **A1**: Thank you for your insightful question. In Sec. 4.1, we have considered the system very carefully (more details in Supp. A.1) and find that this model does indeed hold in most practical systems (i.e., practical generative models). Specifically, we recall that Eqn. 2 mentions that the sequence of $n$ independent experiments (image generations), each with the same success rate (for example, same probability of generating an image with value of sensitive attribute=1 in each generation), can be modeled as a multinomial distribution based on the definition of this distribution. We remark that in a practical system, as long as generation of each sample is independent, and the generative model is time invariant, the requirements are satisfied and our statistical model holds. Note that for the current image generative models like GANs and diffusion models, usually both requirements are met. We will add a statement to clarify this.
>
> $ $
>
> >**Q2**: “The presentation of the statistical model should be improved. Why do the authors assume a multivariate Gaussian distribution instead of other statistical distributions?”
>
> **A2**: We thank the Reviewer for the feedback. The assumption of multivariate Gaussian distribution is based on “normal approximation to the multinomial” [36,37], an application of the central limit theorem. Specifically, as mentioned in Sec 4.1 of the main paper (and also in our previous response), we use multinomial distribution to model the possible events of SA classifier output. Then, since $\mathbf{p}$ in Eqn. (2) is not extreme and $n$ is reasonably large, this multinomial distribution can be approximated by a multivariate Gaussian distribution [36,37] (more details in Supp. A.1). We additionally remark that, to the best of our knowledge, multivariate Gaussian distribution should be the most appropriate approximation, and this enables us to later estimate the distribution of the $\hat{p}$ (Eqn. (4) and (5)) with more ease. Note that the accuracy of this prediction is validated in Supp. C. We will shift some details from Supp., and add some explanations to this part to make it clear.
>
> $ $
>
> >**Q3**: “On page 6, what does "M" represent in equation (3)?”
>
> **A3**: We apologize if it was unclear. In Sec 4.1, the matrix $\mathbf{M}$ is a component of the covariance matrix of the multivariate gaussian distribution i.e., $\mathbf{\Sigma}=n\mathbf{M}$, which is determined following the literature [36,37] and $\mathbf{M}$ characterizes the interaction of elements of the probability vector $\mathbf{p}$. As mentioned, the expanded form of this term can be found in Supp. A.1. We will make this part more clear in the final version.
>
> $ $
>
> >**Q4**: The description of equation (8) is not clear enough. The authors are suggested to explain it in detail.
>
> **A4**: We apologize if it was unclear. As discussed in Sec 4.2. , Eqn. 8 is the maximum likelihood approximation of $p^*$, taking into account error in the sensitive attribute classifier ($\alpha$) and therefore could achieve better estimation and measurement of fairness. We refer to Eqn. 8 as the CLEAM’s point estimate. Due to space limitation, we provide a compact derivation of this equation in the main paper. However, in Supp. A.2, we provide a step-by-step derivation (as mentioned in the main manuscript) and additional in-depth mathematical intuition –statistical requirements and assumption– on how this maximum likelihood approximation is derived and what it entails. We will make this part clearer.
>
> $ $
>
> >**Q5**: Authors are suggested to provide more experiments with other datasets and generative models, in order to demonstrate the effectiveness of the proposed framework.
>
> **A5**: Thank you. We would like to respectfully clarify that our experiments already consider multiple different generative models and datasets. Specifically, in Tab.1 we consider two state-of-the-art (SOTA) generative models StyleGAN2 (Conv-based) and StyleSwin (transformer-based) which are both based on the CelebA-HQ dataset. Then in Tab.2, we consider Stable Diffusion Model (SDM; as SOTA text-to-image generative model) which is based on the LAION-5B [a] dataset. Finally, in the ablation studies (Sec 5.2) we provide further assessment on the AFHQ dataset, which due to space constraint we briefly discuss the results in the main manuscript. More details can be found in Supp. D.3 and D.5. We will include an additional remark, to make these points more explicitly in the final manuscript.
>
> Based on reviewer’s suggestion, as an additional experiment, we further study our proposed CLEAM on an additional generative model and dataset. Specifically, we carried out a similar procedure to analyze the bias w.r.t. $\texttt{Gender}$ of another pre-trained diffusion model [b] on the FFHQ dataset [c]. We utilized CLIP as the SA classifier. For evaluation, we similarly find the ground-truth (GT) $p^*$ by the same procedures in Supp. H to hand-label the generated samples. Note that the GT is used solely for evaluation. Our results show that CLEAM is similarly able to reduce the errors ($e_\mu$ and $e_\rho$) when compared against the baseline. We will include these results in the revised version of the paper.
>
>
> | Model | GT | $\mu_{Base}$ | $e_{\mu}$ | $\mu_{CLEAM}$ | $e_{\mu}$ | $\rho_{Base}$ | $e_\rho$ | $\rho_{CLEAM}$ | $e_{\rho}$ |
> |---|---|---|---|---|---|---|---|---|---|
> | Diffusion model [b] | 0.57 | 0.585 | 2.63% | 0.571 | 0.18% | [0.578,0.593] | 4.04% | [0.564,0.579] | 1.58% |
>
>
> [a] Laion-5b: An open large-scale dataset for training next generation image-text models. NeurIPS’22.
>
> [b] High-resolution image synthesis with latent diffusion models. CVPR’22.
>
> [c] A style-based generator architecture for generative adversarial networks. CVPR’19.

---

> > ### Comment · Reviewer_o1Z2 · 2023-08-15
> > **Re: Response**
> >
> > Thank you for the detailed response. I don’t have any further questions at the moment and will keep my scores for now.

---

> > > ### Author Response · Authors · 2023-08-15
> > > **Thank you for the positive feedback and insightful comments**
> > >
> > > We extend our appreciation to the Reviewer for their insightful comments. We are grateful for the positive feedback, and evaluation. As previously mentioned, the revised manuscript will incorporate all the necessary clarifications and additional results discussed in this rebuttal.
> > >
> > > $ $
> > >
> > > Sincerely,
> > >
> > > Authors

---

### Official Review · Reviewer_rFFs · 2023-07-07

**Soundness:** 3 good
**Presentation:** 2 fair
**Contribution:** 3 good
**Rating:** 6
**Confidence:** 4

**Summary:**

The authors conduct a study on the fairness of generative models.  They propose a CLassifier Error-Aware Measurement (CLEAM) framework which accounts for inaccuracies in classifiers involving sensitive attributes.  The authors also create a new dataset of generated images from a text-to-image generator which are then used to evaluate the accuracy of existing fairness frameworks.

**Strengths:**

Better measurement and quantification of bias is a very critical topic both in classification as well as generation tasks.

The proposed CLEAM framework appears to produce results which are more balanced than the baseline methods.

Comparisons are made to other approaches.

**Weaknesses:**

The text/language/grammar could be improved in many places, especially the abstract.  Even the opening of the intro is confusing- "fairness is defined as equal generative quality and equal representation  w.r.t  some sensitive attributes (SA). In this work, we focus on the more widely utilized definition – equal representation."  Isn't this the same, or actually less demanding definition?  Does generating samples from both classes, but one at a much lower quality level, actually constitute fairness?  Doesn't there need to be some demand on the quality of generation?

The jumping around between the language of SA classifiers and generative models is confusing throughout.

The description/introduction of CLEAM (lines 59-68) is wordy and a bit confusing.  Other than the fact that it is "a statistical model", it's unclear what it is after reading this section. Since this is the focus of the paper, it should be crystal clear what the core of this method is after reading this section.

The authors state "we observe an intriguing property in Stable Diffusion Model"- however what they are observing is instability due to noise.  Again, I believe this is more related to instability/adversarial attacks than "bias" as it is traditionally defined.  I think disentangling these (related) concepts is important.

The paper is very dense with many results as well as mathematics.  It's clear the authors have much to tell with limited space.  However, they need to walk the reader through it a bit more as it is hard to read through the tables and understand what are the key take-aways from each.

**Questions:**

The abstract would benefit form fairly substantial rewriting IMO.  "A ResNet18 for Gender with accuracy 97%" is both an awkward phrase, but also unclear what/why it is so specifically being referenced in the abstract.  The abstract should seek to make a general observation or summary as to the goals and abstract of the work.  Additionally generative models are mentioned first, but then the focus is on classifiers, before moving back to generative models.  Cleaning up the abstract would be very beneficial.

Are the results of Figure 2 "bias" or instability to adversaries?  Traditionally bias is usually thought of when a difference in classes is introduced due to something like "doctor" being associated with men and "nurse" being associated with women.  Here there is literally no difference between "a" and "one", yet the distribution shifts.  This feels more similar to an adversarial attack where random noise is added as opposed to a biased classifier.   Usually adversarial instability in image generation is addressed by adding noise either to the input or latent space.  Here the issue is being caused by noise within the text prompt that is impacting the generation.

It would be helpful to put the SE in the tables and not the supplemental only (makes it tough on the reader jumping back and forth).

**Limitations:**

This work is explicitly designed to overcome limitations of other frameworks which may have generative bias

---

> ### Author Rebuttal · Authors · 2023-08-08
>
> We thank the Reviewer for the valuable suggestions. They are very helpful. We will clean up the abstract and introduction, shorten the discussion of CLEAM in the introduction, shift some results/mathematics to Supp., and better arrange the discussion of SA classifiers/generator as suggested by the Reviewer.
>
> $ $
>
> >**Q1**: "fairness is defined as equal generative quality and equal representation w.r.t some sensitive attributes (SA). In this work, we focus on the more widely utilized definition – equal representation."  Isn't this the same, or actually less demanding definition? Does generating samples from both classes, but one at a much lower quality level, actually constitute fairness? Doesn't there need to be some demand on the quality of generation?
>
> **A1**: We apologize for a typo here. The sentence should be:
>
> "fairness is defined as equal generative quality **or** equal representation w.r.t some sensitive attributes (SA). In this work, we focus on the more widely utilized definition – equal representation."
>
> We would like to clarify that i) equal generative quality and ii) equal representation are two different fairness definitions. In this work, we focus on equal representation. We remark that to the best of our knowledge, majority of works in fair generative modeling focus on equal representation, e.g. to learn a fair generative model with more equal representation [1,2]. Meanwhile, equal generative quality is not the focus of our work. We will clarify this.
>
> $ $
>
> >**Q2**: Are the results of Figure 2 "bias" or instability to adversaries? Traditionally bias is usually thought of when a difference in classes is introduced due to something like "doctor" being associated with men and "nurse" being associated with women. Here there is literally no difference between "a" and "one", yet the distribution shifts. This feels more similar to an adversarial attack where random noise is added as opposed to a biased classifier. Usually adversarial instability in image generation is addressed by adding noise either to the input or latent space. Here the issue is being caused by noise within the text prompt that is impacting the generation.
>
> **A2**: Reviewer's comment is very insightful. The intention of this experiment is to apply our proposed fairness measurement framework to reliably measure biases in popular generative models. When we perform this measurement on Stable Diffusion Model (SDM), we follow best practices for input prompts [24, 29-31] and use indefinite (gender-neutral) pronouns or nouns [32, 33], see Sec. 3 for details. Our careful design of input prompts is to avoid gender stereotypes as mentioned by Reviewer. However, in our experiments, we observe markedly different biased outputs for SDM as shown in Fig. 2. To the best of our knowledge, such observation has not been reported previously for SDM.
>
> We note that in our work, for generative models' fairness/bias, we follow the definition of equal representation as discussed in introduction (also discussed above) instead of the traditionally definition of stereotyping (i.e. generalization about a group of people based on certain traits).
>
>
> Reviewer's viewpoint on instability to adversaries is very insightful, and we believe this could be the **root cause** of our observation of different biased outputs (unequal representation). However, understanding and validating the root cause of such based outputs would require substantial investigation, and is beyond the scope of this work. In our work, the focus is to study fairness measurement and to report biases in popular generative models using our improved framework.
>
> We believe it will be a very interesting future study to understand the root cause of our observed bias, and Reviewer's insight on instability to adversaries is a promising direction to uncover underlying reason.
>
> $ $
>
> >**Q3**: It would be helpful to put the SE in the tables and not the supplemental only (makes it tough on the reader jumping back and forth).
>
> **A3**: We appreciate the Reviewer’s feedback. Given the limited space, including the SE in the tables may make it a bit difficult to read. However, we would attempt to reformat the table to include this following the suggestion by the Reviewer.

---

> > ### Comment · Reviewer_rFFs · 2023-08-21
> > **Response to rebuttal**
> >
> > I have read the other reviews and authors' responses.  I am willing to update my recommendation to weak accept.  I think the question around bias vs. adversarial stability is an important point which is worth addressing.  If the authors make the improvements to the text which they have indicated throughout the review, I think this paper will offer contribution to the community.

---

> > > ### Author Response · Authors · 2023-08-22
> > > **Thank you for the very constructive feedback and increasing the rating**
> > >
> > > We sincerely thank the reviewer for the insightful comments and for increasing their rating.
> > >
> > > We will indeed include the above discussion in the revised manuscript.
> > >
> > > Sincerely, Authors

---

### Official Review · Reviewer_uk5v · 2023-07-24

**Soundness:** 2 fair
**Presentation:** 2 fair
**Contribution:** 2 fair
**Rating:** 3
**Confidence:** 3

**Summary:**

The paper studies measuring fairness in generative models, which is defined as equal number of samples generated from different groups. The measurement needs a sensitive attribute (SA) classifier to predict group attribute to compute the fairness. The paper emprically finds out the error in SA classifier would largely impact the measurement performance of the fairness. They test it by manually labeling samples and comparing fairness measure. The paper then proposes a calibration trick to reduce the fairness measurement error from the SA classifer's error,

**Strengths:**

1. Fairness in generative models is an important problem and open problem

**Weaknesses:**

1. The paper has no insights on why the error in SA classifier would propagate to fairness measurement. The finding is entirely empirical, and seems independent of what kind of fairness definitions used. Would the SA classifier error impact all fairness definitions equally? How much would it impact? What determines the fairness error? The paper is mostly emprical in this finding without providing good insights.
2. The mitigation method seems to be simply computing the metric on different data subsets and then taking average and computing confidence interval. The technical contribution is a little too elementary.
3. The paper largely ignore the vast literature of noisy label, which I think is directly relevant to the problem, i.e. studying the label noise (i.e. imperfect SA prediction in this case) on the impact of fairness and models. For example:

[1] Natarajan, Nagarajan, et al. "Learning with noisy labels." Advances in neural information processing systems 26 (2013).
[2] Lukasik, Michal, et al. "Does label smoothing mitigate label noise?." International Conference on Machine Learning. PMLR, 2020.

There also seems to be some connection between label smoothing and the mitigation. Can you point out any if it exists?

**Questions:**

See Weakness 1 and 3.

**Limitations:**

See Weakness.

---

> ### Author Rebuttal · Authors · 2023-08-09
>
> Regarding Summary: Thank you. Our apologies if unclear, but Reviewer’s Summary is not very accurate:
>
>
> - Our statistical model is overlooked: We develop a statistical model to understand how errors in SA classifier ($\alpha$) affect the fairness measurement ($\hat{p}$). See Fig 1.b., entire Sec 4.1 and Supp A.1 which discuss the statistical model. Therefore, Reviewer’s summary is not accurate: "The paper emprically finds out the error in SA classifier …". Instead, our finding is supported by statistical analysis.
> - Our proposed method is mis-understood: Based on our statistical analysis, we propose a new measurement framework to mitigate such error. See entire Sec 4.2, Algo 1, and Supp A.2. Our statistical modeling of SA classifier error (Sec 4.1) and our new estimators taking into account such error (Eqn. 8, Eqn. 10) have been validated in Supp C. Therefore, Reviewer's summary is not accurate: "The paper then proposes a calibration trick …". Instead, we propose a new measurement framework with statistical grounding.
> - This contribution is overlooked: Using our proposed framework, we evaluate biases in SOTA GAN and diffusion models.
>
> $ $
>
> >**Q1**: The paper has no insights on why the error in SA classifier would propagate to fairness measurement. The finding is entirely empirical, and seems independent of what kind of fairness definitions used.
>
> **A1**: We apologize if unclear, but we have developed a statistical model to provide insights on how errors in SA classifier ($\alpha$) propagate to the fairness measurement ($\hat{p}$). See Fig 1.b., entire Sec 4.1 and Supp A.1. **Importantly, based on our model, Eqn. 4, 5 directly relate the error in SA classifier to fairness measurement, explaining how error in SA classifier impacts fairness measurement statistically.**
>
> $ $
>
> >**Q2**: Would the SA classifier error impact all fairness definitions equally? How much would it impact? What determines the fairness error? The paper is mostly emprical in this finding without providing good insights.
>
> **A2**: Based on our statistical model, Eqn 4, 5 clearly illustrate SA classifier error $\alpha$ determines error in fairness $\hat{p}$ and its impact, see Sec 4.1.
>
> As explained in Fig 1.a, 1.b and entire Sec 4.1, our statistical model is specific to equal representation, the most popular fairness definition for generative model (see Sec 1).
>
> $ $
>
> >**Q3**: The mitigation method seems to be simply computing the metric on different data subsets and then taking average and computing confidence interval. The technical contribution is a little too elementary.
>
>
> **A3**: Our apologies if unclear, but our mitigation method is misunderstood by Reviewer. **Our method is grounded on our statistical model which relates error in SA classifier ($\alpha$) to statistical distribution of fairness measurement ($\hat{p}$)**. Based on this model, we derive our framework (CLEAM) and new estimators, taking into account SA classifier error $\alpha$ to achieve improved fairness measurement. See Fig 1.b, Sec 4.1, Sec 4.2, Algo 1, Supp A.1 and A.2. Supp Sec C validates our proposed framework and our proposed estimators statistically.
>
> By taking into account SA classifier error, our proposed framework could significantly reduce fairness measurement error. **It is important to note that our improved accuracy cannot be achieved by the simple approach mentioned by Reviewer, which we refer to as Baseline and has been extensively compared with our framework CLEAM in Sec 5.**
>
> $ $
>
> >**Q4**: The paper largely ignore the vast literature of noisy label, which I think is directly relevant to the problem, i.e. studying the label noise (i.e. imperfect SA prediction in this case) on the impact of fairness and models.
>
> **A4**: Our apologies if unclear, but our paper does not ignore label noise. Instead, our statistical model  (Fig 1.b) encompasses and takes into account broadly different causes of imperfect SA prediction, e.g. task hardness (ln 208), label noise, and other causes. Specifically, in our statistical model, $\alpha$ captures SA error arising from different causes, including label noise. Meanwhile, our proposed CLEAM takes into account $\alpha$ to achieve accurate fairness measurement (Sec 4.2).
>
> Note that, instead of concerning itself over a specific cause of imperfect SA prediction (e.g., label noise), and attempting to mitigate such a specific issue, our statistical model broadly encompasses different causes of SA classifier error (captured in $\alpha$), and achieves improved fairness measurement without the need to update the SA classifier.
>
> In the next response, we demonstrate effectiveness of our approach vs. an alternative approach to focus on label noise narrowly.
>
> $ $
>
> >**Q5**: There also seems to be some connection between label smoothing and the mitigation. Can you point out any if it exists?
>
> **A5**: Our proposed mitigation is fundamentally different from label smoothing. As discussed, our statistical model broadly encompasses different types of SA classifier error (captured in $\alpha$) and takes them into account under one unified method, without the need to update the SA classifier. This is in sharp contrast to previous work which applies label smoothing to mitigate label noise to train an improved classifier.
>
> To compare our approach vs. label smoothing, we conduct this experiment. We implement label smoothing in the training of a ResNet-18 based SA classifier with the same setup as Tab.1(A) w.r.t. SA $\texttt{Gender}$. In this setup, label smoothing does not significantly improve accuracy of the SA classifier (see $\mathbf{\alpha}$ in the Table), as the SA classifier is already very accurate without label smoothing. However, even with these accurate SA classifiers, fairness errors are significant using Baseline, consistent with our findings in other setups. However, with our proposed CLEAM, fairness errors can be reduced considerably (with or without label smoothing).
> Please see global response for Tab.

---

### Official Review · Reviewer_uNfr · 2023-07-26

**Soundness:** 3 good
**Presentation:** 2 fair
**Contribution:** 3 good
**Rating:** 5
**Confidence:** 2

**Summary:**

This paper proposes a framework for fairness measurement. It first shows that existing framework has considerable measurement errors even when highly accurate sensitive attribute classifiers are used, then propose CLassifier Error-Aware Measurement (CLEAM), a new framework which uses a statistical model to account for inaccuracies in SA classifiers.

**Strengths:**

1. Shows the significant measurement errors of existing frameworks by experiments.
2. Propose new datasets based on generated samples with manual labeling w.r.t. SA.
3. Proposes a simple statistical approximation method is proposed to obtain a stable and accurate estimation of the GT probabilities.

**Weaknesses:**

1. The organization of the paper is kind of hard to follow. The first contribution takes too long and may confuse the readers.
2. The introduction to the proposed method is too short and not very solid. Some theoretical support may be better. Maybe you can talk about cases when using distributions other than Gaussian to approximate the distributions.
3. Only a public dataset CelebA-HQ is used. It's better to test methods on various datasets.

**Questions:**

plz refer to the "weaknesses" part.

**Limitations:**

The structure of paper is hard to follow and somehow boring for readers.

---

> ### Author Rebuttal · Authors · 2023-08-09
>
> Thank you for the valuable suggestion. We will shorten the first contribution and discuss more on our proposed method in the introduction following the Reviewer's suggestion.
>
> $ $
>
> >**Q1**: The introduction to the proposed method is too short and not very solid. Some theoretical support may be better.
>
> **A1**: We apologize if this was unclear and would like to respectfully clarify that the entire proposed solution (Sec.4) is based on theoretical statistical modeling. As mentioned, the proposed method (Sec.4.1) first derives a statistical model to link the observed errors in the fairness measures to the SA classifier’s inaccuracy. Only then are we able to utilize the statistical model to systematically account for the classifier’s error for fairness measurement (Sec.4.2). Furthermore, we remark that our intention was to highlight the essential information to understand CLEAM in the main manuscript, while Supp. A.1 and A.2 provide extensive step-by-step details and derivation of the theoretical model, as indicated in the paper. Nevertheless, we will follow the Reviewer's suggestion and would shift some of these details back into the manuscript.
>
> $ $
>
> >**Q2**: Maybe you can talk about cases when using distributions other than Gaussian to approximate the distributions.
>
> **A2**: Thanks, we would like to clarify that we model our system as a Gaussian Distribution based on “normal approximation to the multinomial” [36,37], an application of the central limit theorem. To the best of our knowledge, this is the most appropriate distribution [36,37] that enables us to derive the distribution of $\hat{p}$. We will clarify this in the paper.
>
> $ $
>
> >**Q3**: Only a public dataset CelebA-HQ is used. It's better to test methods on various datasets.
>
> **A3**: We apologize if this was unclear and would like to respectfully clarify that **our experiments were carried out on three different datasets. This is more than the number of datasets used in previous fair generative modeling work [1,2].**
>
> Specifically, as mentioned by the reviewer, In Tab.1 StyleGAN2 and StyleSwin are used and are based on the CelebA-HQ dataset. Then, in Tab.2, the pre-trained Stable Diffusion Model is based on the LAION-5B[a] dataset. Finally, in the ablation studies (Supp D.3/5) we provide further assessment on the AFHQ dataset (mentioned in Sec.5.2). Overall, our proposed CLEAM demonstrates improved performance compared to other approaches when evaluated over all three datasets.
>
> As an additional experiment in this rebuttal, we include a fourth dataset to assess our proposed CLEAM. Specifically, we carry out a similar procedure to analyze the bias w.r.t. $\texttt{Gender}$ of a pre-trained diffusion model [b] on the FFHQ dataset [c]. We utilize CLIP as the SA classifier. Here, we similarly find the GT $p^*$ by utilizing the same procedures discussed in Supp.H – to hand-label the generated samples. Note that the GT is for evaluation and is not used in our proposed method. Similar to other datasets, our results show that our proposed CLEAM can significantly reduce the errors for this dataset (see $e_\mu$ and $e_\rho$). We will include these results in the final manuscript.
>
> | Model | GT | $\mu_{Base}$ | $e_{\mu}$ | $\mu_{CLEAM}$ | $e_{\mu}$ | $\rho_{Base}$ | $e_\rho$ | $\rho_{CLEAM}$ | $e_{\rho}$ |
> |---|---|---|---|---|---|---|---|---|---|
> | Diffusion model [b] | 0.57 | 0.585 | 2.63% | 0.571 | 0.18% | [0.578,0.593] | 4.04% | [0.564,0.579] | 1.58% |
>
> $ $
>
> [a] "Laion-5b: An open large-scale dataset for training next generation image-text models." NeurIPS’22.
>
> [b] "High-resolution image synthesis with latent diffusion models." CVPR’22.
>
> [c] "A style-based generator architecture for generative adversarial networks." CVPR’19.

---

> > ### Comment · Reviewer_uNfr · 2023-08-13
> >
> > Thanks for your rebuttal. I think basically it solves my confusions and I raise my score.

---

> > > ### Author Response · Authors · 2023-08-15
> > > **Thank you for the very constructive feedback and increasing the rating**
> > >
> > > We sincerely thank the reviewer for the insightful comments. Thank you for the positive feedback and for increasing the rating. We will include all additional results in the revised version. Sincerely, Authors

---

### Official Review · Reviewer_2H4p · 2023-07-27

**Soundness:** 2 fair
**Presentation:** 3 good
**Contribution:** 2 fair
**Rating:** 6
**Confidence:** 1

**Summary:**

The objective of this paper is on measuring the fairness in generative models. There are three contributions. (i) Consideration of measurement errors of sensitive attribute (SA) classifiers in fairness measurement of generative models. (ii) A classification error aware measurement framework, called CLEAM, which based on statistical model accounts for the inaccuracies of SA classifier to reduce measurement error in generative models. (iii) As an application, the authors demonstrate that CLEAM can be applied to measure fairness in text-to-image generator and GANs.



**Strengths:**

The paper is well written, with clear intuitions, illustrations, and experimental results.

**Weaknesses:**

- Putting the application of gender bias in the introduction seems to be out of place, possibly undermining the main framework CLEAM. Similarly, table 1 is also out of place, throwing numbers to the readers without explaining the setup.

- The authors make an assumption of ground-truth labels of sensitive attributes, which may not be available in practice.

- Some of the human faces in Figure 2 and figure 3(a) are different, revealing the uncertainty of the generative models. I understand the intuition of the authors to provide a demonstration of the paper at an early part of the paper. Please clarify if I misunderstood anything.



**Questions:**

- The presented method considers binary sensitive attributes. However, modern generative models easily generates images with multi-sensitive features. How do the authors extend to this scenario? Is naive enumeration the only possible way?

- In Equation (2), are $p_i$ and  $\alpha_i$ known?

- In line 222, is $ s = 30 $ large enough?

- Is there any future work based on this paper?

Please also address the points in "Weakness" above.

**Limitations:**

NA.

---

> ### Author Rebuttal · Authors · 2023-08-09
>
> >**Q1**: Putting the application of gender bias in the introduction seems to be out of place, possibly undermining the main framework CLEAM. Similarly, table 1 is also out of place, throwing numbers to the readers without explaining the setup
>
> **A1**: Thank you for your suggestion. We will fix the placement of Fig. 2 and Tab. 1
>
> $ $
>
> >**Q2**:The authors make an assumption of ground-truth labels of sensitive attributes, which may not be available in practice.
>
> **A2**: We thank the reviewer for the comment. Importantly, we would like to respectively clarify that ground-truth (GT) labels are not used in our proposed method.
>
> Meanwhile, the GT labels for the generated data (GenData introduced in our work) is used **solely for evaluation** of different methods. Again, the GT labels for the generated data is not used in our proposed method.
>
> Our method assumes availability of a SA classifier and its validation accuracy ($\alpha$). In practice, $\alpha$ is computed during the validation stage of a SA classifier. Note that a SA classifier is needed in all existing measurement methods.
>
> $ $
>
> >**Q3**: Some of the human faces in Figure 2 and figure 3(a) are different, revealing the uncertainty of the generative models. I understand the intuition of the authors to provide a demonstration of the paper at an early part of the paper. Please clarify if I misunderstood anything.
>
> **A3**: We apologize for any confusion. To clarify, the images displayed across Fig. 2 and Fig. 3(a) contain some different seed values, while in between each individual figure we utilize the same seed value. However, for improved clarity, we have included a revised Fig.3(a) (in the attached pdf, Fig. 1(rebuttal) ) with the same seed values for the displayed images as Fig 2.
>
> $ $
>
> >**Q4**: The presented method considers binary sensitive attributes. However, modern generative models easily generates images with multi-sensitive features. How do the authors extend to this scenario? Is naive enumeration the only possible way?
>
> **A4**: Thank you for your comment. Considering the multi-sensitive attributes is indeed an interesting idea. However, we remark that in current literature, fairness of generative models has been studied for binary sensitive attributes mainly due to lack of an available large labeled dataset needed for systematic experimentation. As a result, CLEAM similarly focuses on binary SA to address a common flaw in the evaluation process of the many proposed State-of-the-Art methods.
>
> Assuming that constraint of dataset is addressed, our same CLEAM approach can be easily extended to a multi-label setting. For example, given a 3 label sensitive attribute where $p^*_j$ is the probability of generating a sample with label $j$ and ${\alpha}\_{i|j}$ denotes the probability (“accuracy”) of the SA classifier in classifying a sample with GT label $j$ as $i$ for $i,j \in \\{0,1,2\\}$, Fig. 2 (Rebuttal) in the attached Pdf shows our statistical model for this setting. We can then similarly solve for the $p^*$ point estimate by solving the matrix:
> $$ \begin{bmatrix}
> \alpha\_{0|0} & \alpha\_{0|1} & \alpha\_{0|2}\\\\
> \alpha\_{1|0} & \alpha\_{1|1} & \alpha\_{1|2}\\\\
> \alpha\_{2|0} & \alpha\_{2|1} & \alpha\_{2|2}\\\\
> \end{bmatrix}
> \begin{bmatrix}
> p^*\_0 \\\\
> p^*\_1 \\\\
> p^*\_2
> \end{bmatrix}
> = \begin{bmatrix}
> \ddot{\mu}\_\hat{p_0} \\\\
> \ddot{\mu}\_\hat{p_1} \\\\
> \ddot{\mu}\_\hat{p_2}
> \end{bmatrix}$$
>
> We will include the detailed procedure and full solution in the final version of the paper.
>
> $ $
>
> >**Q5**: In Equation (2),  are $p_i$  and $ \alpha_i$  known?
>
> **A5**: We apologize if it was unclear, as discussed in Sec.3 of the main paper, $p^*_i$ is the unknown GT distribution that we are trying to find and $\alpha_i$ is a known accuracy of the SA classifier. In practice, accuracy of the classifier ($\alpha_i$) is evaluated during the validation stage, a common practice when training discriminative models (line 113-114). On the other hand, as first introduced in Sec.2, the ground truth SA distribution ($p^*_i$) is unknown as the SA classifier only outputs some approximation $\hat{p}$. Hence, in Sec. 4.1., we provide the setup to model the theoretical distribution, and then utilize it in Sec.4.2 to propose CLEAM for a better estimation of $p^*_i$ with eqn. (8) and eqn. (10).
>
> $ $
>
> >**Q6**: In line 222, is s=30 large enough?
>
> **A6**: Thanks, our experimental results in Supp. F-Fig 5 demonstrates that $s$=30 is sufficiently large enough to significantly reduce measurement error, where increasing $s$ does not result in considerable performance improvement. Supp. C.1 then further substantiates these findings by showing that $s$=30 provides a close approximation between the sample based estimate and the model based estimates.
>
> $ $
>
> >**Q7**: Is there any future work based on this paper?
>
> **A7**: Thanks for your question. As future work, now with a more reliable fairness measurement framework, firstly, we consider utilizing this more accurate measurement as a bias mitigation technique during fair generative model training. Secondly, inspired by the findings in Sec.6, we propose to further look into the biases that exist in stable diffusion models (discussed in Sec 6 of our manuscript) and identify their possible sources.

---

> > ### Comment · Reviewer_2H4p · 2023-08-18
> > **After rebuttal**
> >
> > Thanks for the response. It clarifies my questions. Best of luck.

---

> > > ### Author Response · Authors · 2023-08-18
> > > **Thank you for the positive feedback and constructive comments**
> > >
> > > We express our gratitude to the Reviewer for their constructive remarks, as well as the positive assessment.
> > > As discussed, we will carefully address the Reviewer's comments in the final manuscript.
> > >
> > > Sincerely,
> > >
> > > Authors

---

### Author Rebuttal · Authors · 2023-08-09

We thank all the reviewers for their valuable time and effort to review our work. We appreciate the Reviewers' kind comments and recognition, such as:

- "The paper is well written, with clear intuitions, illustrations, and experimental results." (Reviewer 2H4p)
- "Fairness in generative models is an important problem and open problem" (Reviewer uk5v)
- "The fundamental statistic model is easy to follow, and the proposed method CLEAM is able to reduce the fairness measurement error." (Reviewer o1Z2)
- "The proposed CLEAM framework appears to produce results which are more balanced than the baseline methods." (Reviewer rFFs)
- "CLEAM seems to be an intuitive correction to the naive baseline method" (Reviewer 6ZNP)
- "Propose new datasets based on generated samples with manual labeling w.r.t. SA." ( Reviewer uNfr)



We would also like to express our appreciation to all the Reviewers for giving us the opportunity to clarify our work, as well as the constructive comments.  We will consider all suggestions seriously.

To briefly recap our work:

1. Our work is the first to **challenge the accuracy of the existing fairness measurement framework** in generative models. Our major contribution is a detailed **statistical model** to model fairness measurement  (Fig 1.b, Sec 4.1 and more details in Supp. A.1). Importantly, our statistical model relates error in sensitive attribute (SA) classifier to statistical distribution of fairness measurement (Eqn. 4, 5), providing insights on how SA classifier error impacts accuracy of fairness measurement.

2. Based on our statistical model, we derive our new fairness measurement framework (CLEAM) and new estimators, taking into account SA classifier error to achieve improved fairness measurement. See Sec 4.2, Algo 1, Supp A.2. Supp Sec C validates our proposed framework and our proposed estimators statistically.

3. To make it possible to study fairness measurement, we develop and make available a new dataset (GenData) consisting of labeled generated samples from three State-of-the-Art (SOTA) generative models. We remark that in generative modeling, fairness is popularly defined as **equal representation** [1, 2, 7, 9, 12, 16, 17] and should not be mistaken with the classifier’s fairness definitions (more detailed comparison in Supp. Sec. G)


4. We have performed comprehensive experiments to validate our proposed measurement framework and demonstrate consistent and substantial improvement in accuracy compared to other approaches. Our experiments include:

   - **3 different generative models**: StyleGAN2, StyleSwin and Stable Diffusion model
   - **3 different datasets**: CelebA-HQ, LAION-5B [a] and AFHQ
   - **5 different SA classifier**: ResNet-18/34, MobileNet2,VGG-16 and CLIP .

Due to space limitation, additional experiments (e.g., with different sensitive attributes) have been diverted to the Supp. Sec.D

5. Finally, utilizing CLEAM, we carry out **reliably** measurement of biases in existing SOTA generative models.

$ $

In this rebuttal, we have also included few additional experiment, as requested by the reviewers. All results support our measurement framework and is superior compared to previous work.

1. Additional generative model and dataset: a Diffusion Model [b] pre-trained on FFHQ [c]

2. Additional comparison with label smoothing: our proposed framework outperforms label smoothing significantly. It should be noted that in our original submission we have already compared with other classifier correction methods: i) mitigate label shift in SA classifier and ii) calibrate SA classifier. All experiments show our proposed framework is superior. More details can be found in Supp. D.8 and G.

In what follows, we provide comprehensive responses to all questions. We have provided anonymized link to Area Chair for the code of all additional experiments. We could provide more details if there are further questions. We hope that our responses can address the concerns and we sincerely hope that reviewers could consider increasing the ratings if our responses have addressed all the questions.

$ $

### Additional experiments on Diffusion Model [b] on FFHQ dataset
| Model | GT | $\mu_{Base}$ | $e_{\mu}$ | $\mu_{CLEAM}$ | $e_{\mu}$ | $\rho_{Base}$ | $e_\rho$ | $\rho_{CLEAM}$ | $e_{\rho}$ |
|---|---|---|---|---|---|---|---|---|---|
| Diffusion model [b] | 0.57 | 0.585 | 2.63% | 0.571 | 0.18% | [0.578,0.593] | 4.04% | [0.564,0.579] | 1.58% |

$ $

### Additional comparison with Label Smoothing

| Model | $\alpha$ | GT | $\mu_{Base}$ | $e_{\mu}$ | $\mu_{CLEAM}$ | $e_{\mu}$ | $\rho_{Base}$ | $e_\rho$ | $\rho_{CLEAM}$ | $e_{\rho}$ |
|---|---|---|---|---|---|---|---|---|---|---|
| R18 w/o smooting | \{0.947,0.982\} | 0.642 | 0.610 | 4.98% | 0.638 | 0.62% | [0.602,0.618] | 6.23% | [0.629,0.646] | 2.02% |
| R18 w    smooting | \{0.935,0.985\} |0.642 | 0.605 | 5.76% | 0.641 | 0.16% | [0.595,0.615] | 7.32% | [0.632,0.650] | 1.56% |



$ $

[a] "Laion-5b: An open large-scale dataset for training next generation image-text models." NeurIPS’22.

[b] "High-resolution image synthesis with latent diffusion models." CVPR’22.

[c] "A style-based generator architecture for generative adversarial networks." CVPR’19.

---

### Author Response · Authors · 2023-08-18
**Thanking the Reviewers for their valuable feedback**

We thank the Reviewers for all of their valuable feedback and for taking the time to thoroughly review our paper.


Our proposed fairness measurement framework is grounded on a careful statistical analysis and can improve accuracy substantially. The framework can be used to study biases in SOTA generative models as demonstrated in our paper. We believe our work would make important contributions to advance fair generative modeling research.

We have submitted all the codes of our experiments for reproducible research.

Thank you for your valuable time in reviewing our submission. We hope Reviewers could consider increasing the ratings from initial decisions if our responses can resolve Reviewers' concerns.

$ $

Sincerely,

Authors

---

### Decision · Program_Chairs · 2023-09-21

**Decision:**

Accept (poster)

**Comment:**

The committee recognizes the paper's novel approach to assessing fairness in generative models, noting the potential contributions of the CLEAM method and the newly introduced dataset. While there is merit in the presented framework, the meta-reviewer echoes concerns regarding language clarity and the delineation of concepts within the paper. The meta-reviewer recommends acceptance, urging the authors to address the identified issues to further enhance the quality and impact of the work.